# Structures of tmRNA and SmpB as they transit through the ribosome

Charlotte Guyomar[1,2], Gaetano D'Urso [1,2], Sophie Chat[1], Emmanuel Giudice [1✉] & Reynald Gillet[1✉]

In bacteria, *trans*-translation is the main rescue system, freeing ribosomes stalled on defective messenger RNAs. This mechanism is driven by small protein B (SmpB) and transfer-messenger RNA (tmRNA), a hybrid RNA known to have both a tRNA-like and an mRNA-like domain. Here we present four cryo-EM structures of the ribosome during *trans*-translation at resolutions from 3.0 to 3.4 Å. These include the high-resolution structure of the whole pre-accommodated state, as well as structures of the accommodated state, the translocated state, and a translocation intermediate. Together, they shed light on the movements of the tmRNA-SmpB complex in the ribosome, from its delivery by the elongation factor EF-Tu to its passage through the ribosomal A and P sites after the opening of the B1 bridges. Additionally, we describe the interactions between the tmRNA-SmpB complex and the ribosome. These explain why the process does not interfere with canonical translation.

---

[1] Univ. Rennes, CNRS, Institut de Génétique et Développement de Rennes (IGDR) UMR 6290, Rennes, France. [2] These authors contributed equally: Charlotte Guyomar, Gaetano d'Urso. ✉email: emmanuel.giudice@univ-rennes1.fr; reynald.gillet@univ-rennes1.fr

In bacteria, ribosomes frequently stall on messenger RNAs (mRNAs) that lack a termination codon, leading to the accumulation of non-productive translation complexes. These complexes are composed of chains of polysomes that are in turn made up of stalled ribosomes, incomplete peptidyl-tRNAs and problematic mRNAs. Depending on where the first ribosome gets stuck, these defective events are defined either as 'non-stop' or 'no-go': non-stop if occurring at the 3′-end of a problematic mRNA lacking a stop codon, or no-go if it occurs before encountering a stop codon (for a review, see Giudice and Gillet[1]). Because stalling results in the build-up of non-productive polysomes as well as a potential synthesis of toxic polypeptides, cell viability and survival depends on rescuing the trapped ribosomes[2].

While *trans*-translation is the main rescue system for freeing stalled ribosomes in bacteria, the mechanism is absent in eukaryotes, making it a particularly attractive target when developing new antibiotics[2,3]. The process is driven by two principal players, transfer-messenger RNA (tmRNA) and small protein B (SmpB). tmRNA is a hybrid RNA molecule that is highly structured. It folds into several domains, including a tRNA-like domain (TLD), which is associated with SmpB, and an mRNA-like domain (MLD), which has an internal open reading frame. The TLD resembles the upper part of tRNA, and SmpB mimics tRNA's anticodon stem-loop[4]. Because the TLD's third base pair is a G·U wobble specifically recognized by alanyl-tRNA synthetase (AlaRS), tmRNA is always charged with an alanine[5]. Aminoacylation allows the elongation factor Tu (EF-Tu·GTP) to bring the alanyl-tmRNA-SmpB complex into the stalled ribosome, similarly to how a canonical tRNA is delivered. The second key domain of tmRNA is the MLD, whose internal open reading frame encodes for a tag, which is added to the stuck incomplete peptide so that it is specifically recognized by proteases[6]. The rest of the tmRNA is composed of four pseudoknots (PKs) (PK1 to PK4) and several RNA helices (Supplementary Fig. 1). When tmRNA travels through the ribosome, canonical translation resumes on the MLD, and the ribosomal subunits are finally released when the stop codon is reached. The non-stop mRNAs are degraded by RNase R[7], and several proteases degrade the incomplete peptides[8,9].

The early steps of *trans*-translation can be subdivided into three main processes. The first is the pre-accommodation step (PRE-ACC), when the quaternary complex made by alanylated tmRNA, SmpB, EF-Tu and GTP binds to the A site of stalled ribosomes. In the accommodation step (ACC), EF-Tu hydrolyses GTP and disassociates from the complex, causing the aminoacyl-tRNA end to swing into the peptidyl transferase centre (PTC). The stalled peptide is then transferred to the alanine residue on tmRNA, and in the third translocation step (TRANS), EF-G catalyses the shifting of the tmRNA-SmpB complex from the A to the P site. The problematic mRNA is released, and the tmRNA resume codon enters the A site to be decoded.

Various structural analyses have been done that have greatly contributed to understanding how tmRNA-SmpB moves through the ribosome to perform *trans*-translation. Among these, the first cryo-electron microscopy (cryo-EM) studies described the PRE-ACC, ACC and TRANS steps, but unfortunately these were poorly resolved (with resolutions not better than 14 Å)[10–14]. The crystal structure of a truncated tmRNA associated with SmpB and EF-Tu and bound to the ribosome was then resolved at 3.2 Å. This showed how, during the PRE-ACC step, SmpB recognizes stalled ribosomes and facilitates decoding in the absence of an mRNA codon in the A site. However that structure does not include the PK ring, which should contain the MLD, the PKs and the H5 stem-loop[15]. At 8.3 Å, a cryo-EM structure of the *Escherichia coli* ribosome in complex with tmRNA-SmpB and EF-

G allowed us to better understand re-registration on the MLD[16]. More recently, three high-resolution structures of *trans*-translation intermediates were also published[17], and these provide more details on how the circularized tmRNA-SmpB complex moves through the ribosome. However, no high-resolution structure of the PRE-ACC step in full-length tmRNA has been published, so there has been no detailed molecular description of how the tmRNA[ala]-SmpB-EF-Tu complex loads on the ribosome. This step is particularly interesting since, unlike in canonical translation, ribosome recognition only occurs in the absence of codon–anticodon base pairing. The major role played by SmpB and tmRNA interactions during pre-accommodation and how this affects the subsequent transition to the ACC and TRANS states was unclear.

Here we show four cryo-EM structures of *trans*-translating ribosomes at resolutions from 3.0 to 3.4 Å (Fig. 1, Supplementary Figs. 2–4 and Supplementary Table 1). The ribosomes from *E. coli*, stalled on a small non-stop mRNA with Phe-tRNA[Phe] in the P site, are mixed with aminoacylated tmRNA, EF-Tu·GTP and SmpB, and the complexes are examined alone or in the presence of kirromycin, an antibiotic that prevents EF-Tu·GDP release, to identify the PRE-ACC (Fig. 1a) and ACC (Fig. 1b) states. In a third experiment, in the presence of elongation factor G (EF-G), we identify two different translocation states. The first 'TRANS' state is just after the translocation of tmRNA-SmpB from the A to the P site (Fig. 1c), while the 'TRANS*' state (Fig. 1d) occurs after TRANS but just before the tmRNA-SmpB complex exits the P site. Our structures provide insight into the mechanism by which tmRNA-SmpB navigates into the ribosome to perform *trans*-translation.

## Results

**Pre-accommodation step**. The first structure we present is of the PRE-ACC state. It features a stalled ribosome, a truncated mRNA, and the tmRNA-SmpB-EF-Tu·GDP quaternary complex bound to the A site (Fig. 1a). This is the first high-resolution structure that shows the entire quaternary complex, including tmRNA's PK ring, H2 helix and H5 stem-loop. The complex is quite dynamic and the PK ring is flexible, resulting in a local resolution that fluctuates between 3.5 and 10 Å. However, the tips of the H5 stem-loop and SmpB C-terminal tail are seen ~3.5 Å, PK2 is ~4.5 Å and the interfaces between tmRNA, SmpB and EF-Tu are ~5.5 Å, which allows for the molecular description of specific interactions.

The TLD/EF-Tu interactions are similar to those observed in the previously published crystal structure of a *Thermus thermophilus* ribosome bound to a complex containing a tmRNA fragment, SmpB, EF-Tu·GDP and kirromycin[15] (Supplementary Fig. 5), although tmRNA-SmpB binds the ribosome slightly differently in this structure (see below). It also confirms that only one SmpB protein is bound to the TLD during pre-translocation[18]. While the EF-Tu switch 1 loop is too flexible to be modelled, densities are observed for the GDP and kirromycin (Supplementary Fig. 6), so it is clear that EF-Tu is in its GDP-bound state after GTP[19,20]. The TLD is partly positioned in what is known as the 'A/T' state, which allows the simultaneous interactions of tmRNA-SmpB with the decoding centre (DC) and EF-Tu with the 50S subunit[21]. As expected[22,23], we confirm that the large open L-shaped TLD forms an angle of ~120°, allowing the SmpB-bound tmRNA to mimic the functioning of a canonical tRNA. The conformations of the 3′-CCA end, the acceptor arm, and the T-arm portion also resemble those of the PRE-ACC state of aminoacyl-tRNA (Supplementary Fig. 7). While the acceptor and T-arms of the TLD are in close contact with EF-Tu (Supplementary Fig. 6), the TLD elbow region (formed by the D

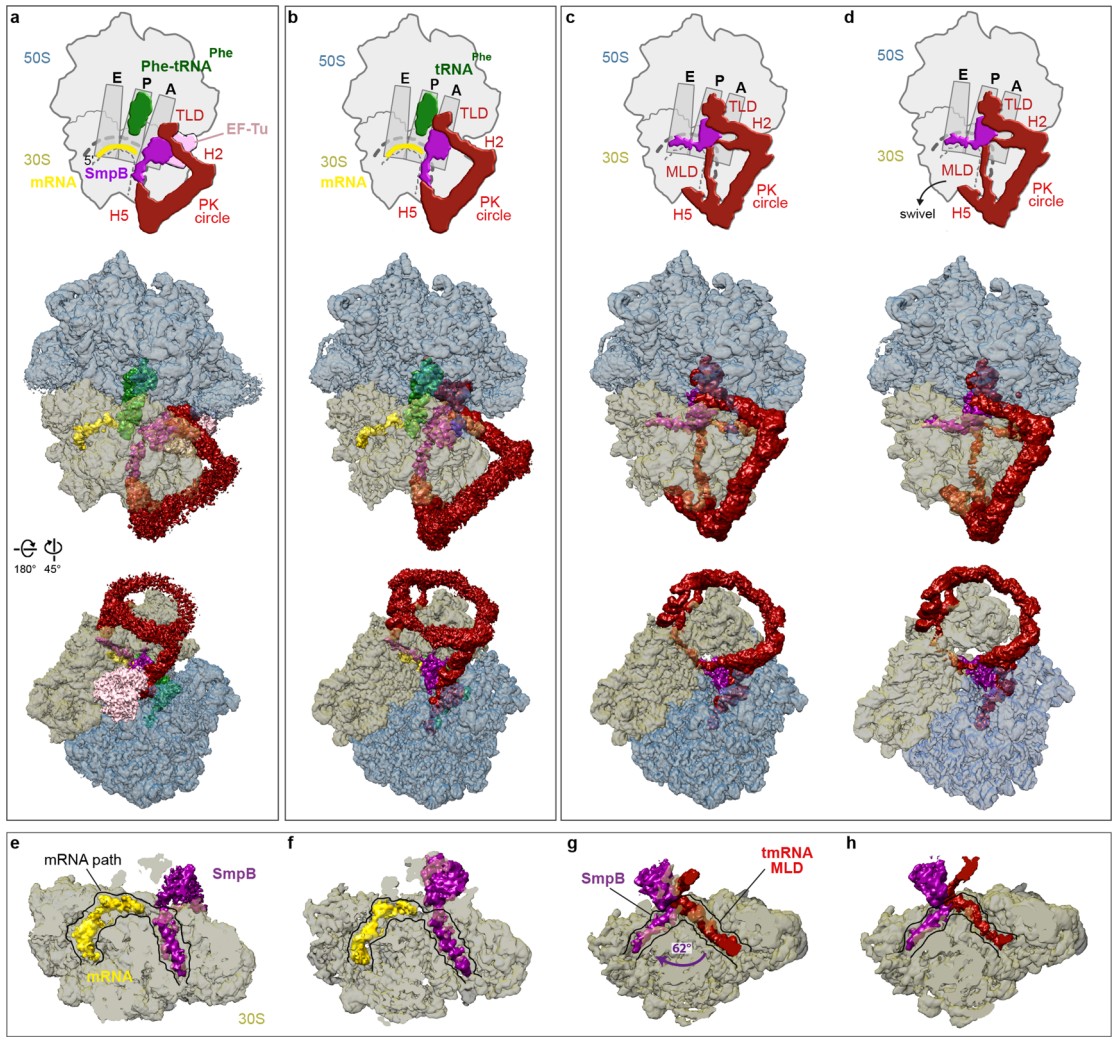

**Fig. 1 High-resolution structures of four consecutive *trans*-translation states.** Shown are the **a** pre-accommodation (PRE-ACC), **b** accommodation (ACC), **c** translocation (TRANS) and **d** intermediate post-translocation (TRANS*) complexes. Top, schematic representations of the complexes showing the ribosomal 50S (blue) and 30S (khaki) subunits, non-stop mRNA (yellow), P-site tRNA$^{phe}$ (green), elongation factor EF-Tu (pink), SmpB (purple) and tmRNA (red). Also indicated are the ribosome's A, P and E sites, and the tmRNA structural domains: the H2 helix, H5 stem-loop, pseudoknot (PK) ring, mRNA-like domain (MLD) and tRNA-like domain (TLD). Middle, electron density maps contoured at 2.5 σ, where sigma refers to the variance in the map. Bottom, the same density maps rotated by 180° around the *Y*-axis and 45° around the *X*-axis. **e–h** Cross-sections of the 30S subunits in the same states as (**a–d**), respectively, showing the SmpB C-terminal tail, non-stop mRNA and the MLD in the mRNA channel.

and variable loops) is in close contact with the SmpB body (Supplementary Fig. 8a). As a result, the Trp122 residue of SmpB is inserted into the hydrophobic pocket formed by A15, U17 and A334 (Supplementary Fig. 9a, left). The highly conserved nucleotide G19, known to prevent tmRNA-SmpB binding when mutated to C[24], is tightly packed at the surface of one of SmpB's hydrophobic patches (Leu91 and Leu92) of SmpB, and it is maintained there by its stacking with C18 (Supplementary Fig. 9b, left). The H2 helix points out of the ribosome between the two subunits, while the MLD, H5 stem-loop and the four PKs are tightly packed into an elliptic ring of 81 × 97 Å around the beak of the 30S small subunit (Fig. 1a). The PK ring is quite flexible, resulting in a lower resolution than everywhere else in the ribosome except for the H5 stem-loop and PK2, which is the only PK in close contact with the ribosome (Fig. 2). PK2's nucleotides C183 to A185 interact with residues Arg72, Pro73 and Ile77, which belong to the type II K-homology (KH2) RNA-binding domain of uS3 (Fig. 2b). The H5 stem-loop is well-defined and interacts with the uS3, uS4 and uS5 proteins involved in the helicase activity of the ribosome[25,26] (Fig. 2a). Its nucleotides

G114 and C115 are stabilized by residues Arg132 and Arg136 of the uS3 C-terminal domain. U119 interacts with uS4 residue Arg47. U120, the stop codon's first nucleotide, lies on top of uS5 residue Ile60, while its second, A121, is at the interface between uS3 and uS5 (Fig. 2c). Interestingly, most of these interactions are similar to those previously described for structured mRNAs[27,28]. However, compared to those mRNAs, the H5 stem-loop mostly differs in how it interacts with uS5. Indeed, as the tmRNA is not yet engaged in the mRNA channel, there are no interactions with Arg20, Phe33, or Val56, and instead the tip of the stem-loop rests on top of the α1 helix residues Ile60 and Gln61.

Despite its high flexibility, the MLD is folded and quite dense, resting between the shoulder and beak of the 30S subunit, parallel to the mRNA path. SmpB lies on the DC, and its core mimics the cognate codon–anticodon base pairs[15]. In the DC, A1492 is stacked on the 16S rRNA h44 helix, while A1493 is in an intermediate state, shifted toward the major groove because of its interaction with SmpB's His22 (Fig. 3). This conformation is different from that of canonical decoding[29,30], but similar to the published high-resolution structure of an empty *E. coli*

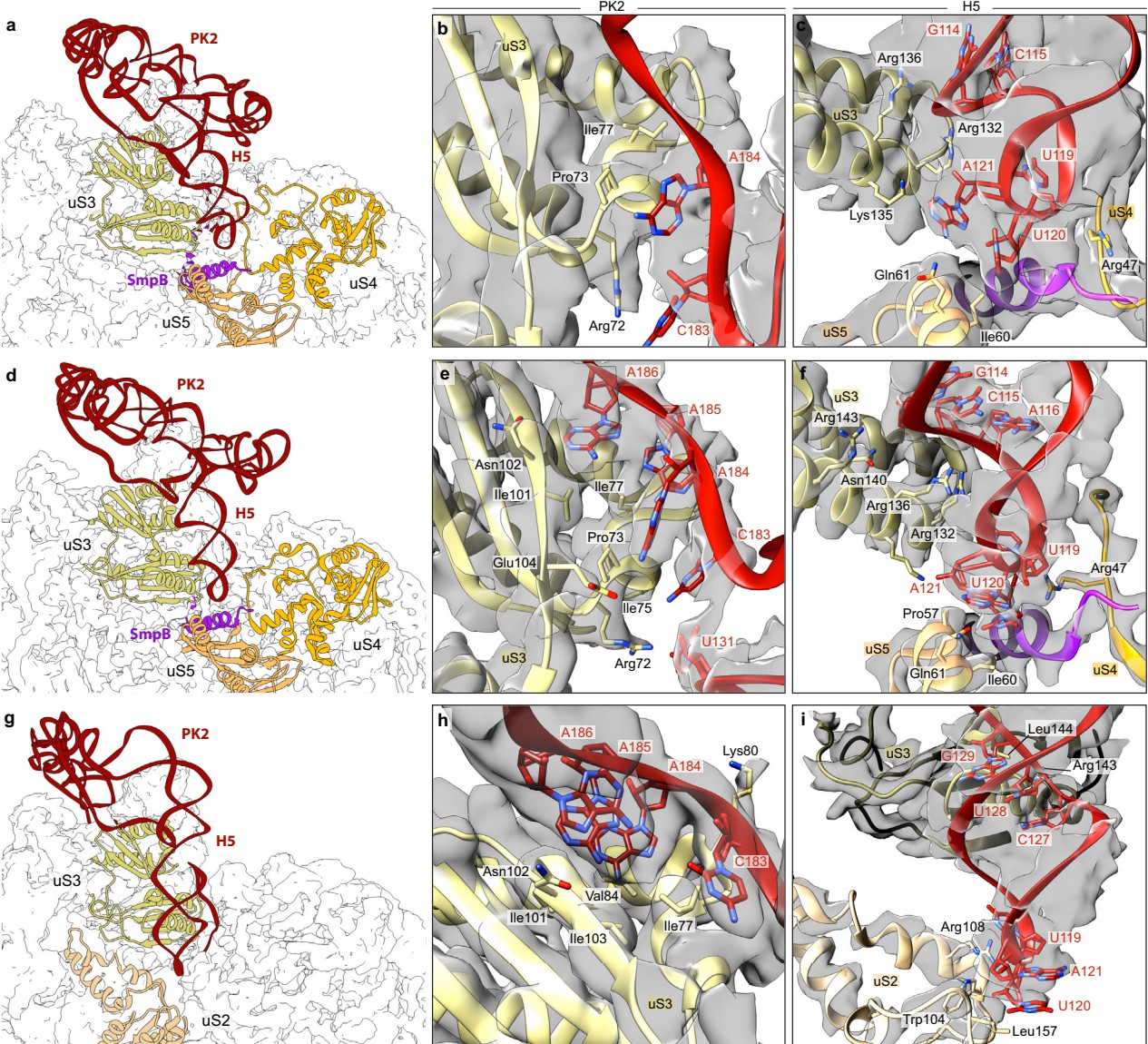

**Fig. 2 The H5 and PK2 domains of tmRNA interact with the uS2, uS3, uS4 and uS5 proteins on the small ribosomal subunit.** Shown are the **a–c** pre-accommodation state, **d–f** accommodation state, and **g–i** translocation state. Left column, overview of the interactions between tmRNA (red), SmpB (purple), and the ribosomal proteins uS2 (tan), uS3 (khaki) uS4 (gold) and uS5 (sandy brown). To highlight the motion of the H5 stem-loop during translocation, all structures are aligned on uS3. Middle, close-up of the interactions between the PK2 pseudoknot and uS3's KH2 RNA-binding domain. Right, close-up of the interactions between the H5 stem-loop and SmpB, uS3, uS4 and uS5. Residues and nucleotides within 4 Å of each other are indicated, and the cryo-electron density map is displayed.

ribosome[31] (Supplementary Fig. 10). Our structure also slightly differs from the crystal structure of a tmRNA fragment, SmpB and EF-Tu of *Thermus Thermophilus* bound to a ribosome[15] (Supplementary Fig. 10a, g). Since the main difference between the two complexes comes from our inclusion of the PK ring, we hypothesize that H5 stalling at the entrance channel slows down the movement of the complex. The interaction between H5 and the SmpB C-terminal tail (see below) would therefore result in the stabilization of an earlier stage of pre-accommodation, explaining the difference in SmpB position/conformation (Supplementary Fig. 5) and why A1493 is not yet flipped out. The beginning of the SmpB C-terminal tail is mostly unstructured, but the Gln135, His136 and Lys138 residues tightly interact with the conserved 16S rRNA G530 residue (Fig. 3a and Supplementary Fig. 11).

Interactions between tmRNA-SmpB and the 30S subunit lead to the closure of 30S, similarly to what occurs with cognate tRNA binding[32] (Supplementary Fig. 12). This closed conformation increases GTP hydrolysis, explaining why tmRNA-SmpB can be efficiently accommodated in the PTC even without cognate codon–anticodon pairing. The rest of the SmpB C-terminal tail is folded into an α-helix that occupies the mRNA path downstream from the stalled mRNA (Fig. 1e), and it is stabilized there by its interaction with 16S rRNA. Specifically, SmpB's Arg139 interacts with the C1195 nucleotide of the h34 helix, while Lys138 and Arg145 interact with the tip of the h18 helix at C528, G529 and A535 (Fig. 4). The tail is also further maintained by its interaction with the first two β-sheets of uS5, in particular, the residues Arg20 and Phe33 (which stack with SmpB's Trp147) and Phe31 (which interacts with Lys143).

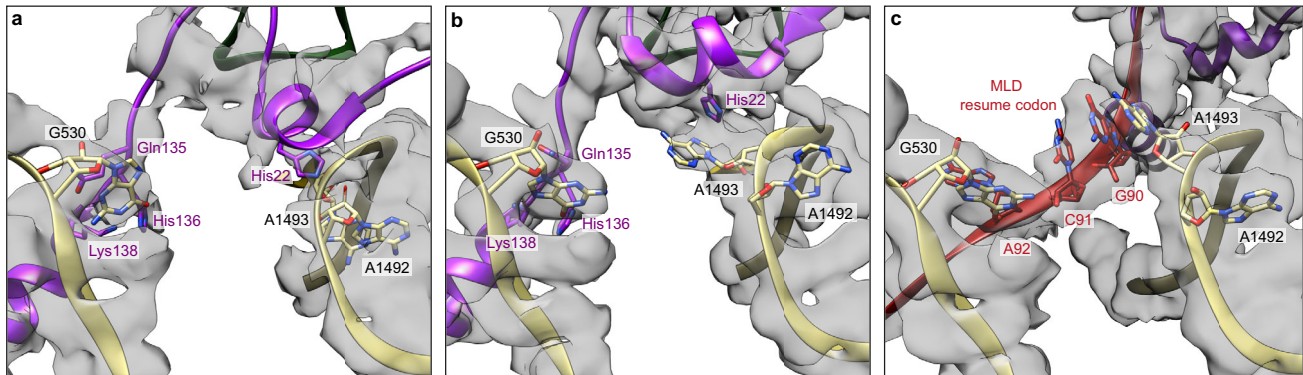

**Fig. 3 Detailed view of the decoding centre during *trans*-translation.** Close-up of the interactions between SmpB (purple) and the conserved nucleotides G530, A1492 and A1493 of the 16S rRNA (khaki) in the **a** pre-accommodation and **b** accommodation states. **c** Interactions between tmRNA's resume codon (red) and the same nucleotides during the translocation state. Residues and nucleotides within 4 Å of each other are indicated, and the cryo-electron density map is displayed.

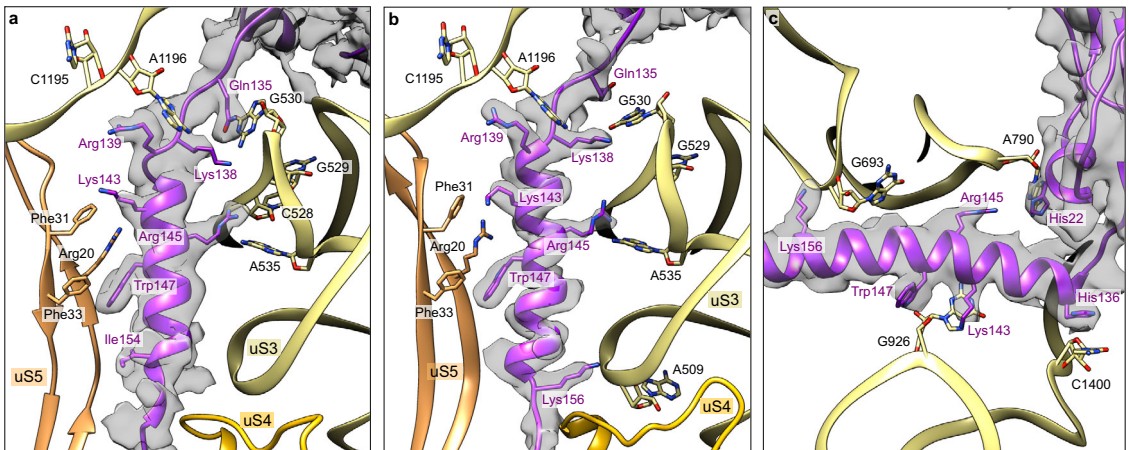

**Fig. 4 Interactions between SmpB's C-terminal tail and the 30S small ribosomal subunit.** Each panel details the contacts that stabilize the C-terminal tail of SmpB in the mRNA channel during the **a** pre-accommodation, **b** accommodation, and **c** translocation states. SmpB is purple, 16S rRNA is khaki, and the ribosomal proteins uS4 and uS5 are gold and sandy brown, respectively. Residues and nucleotides within 4 Å of each other are indicated, and the cryo-electron density map of SmpB is displayed.

The SmpB C-terminal tail also engages in a previously unseen interaction with the tmRNA H5 stem-loop. Indeed, the tmRNA nucleotides U119, U120 and U121 interact with SmpB residues Lys156 and Asn157 (Fig. 5).

**Accommodation step.** The second structure we present here is of the ACC state, which occurs after the release of the EF-Tu·GDP complex and the accommodation of the TLD in the PTC (Fig. 1b). The structure resolution is 3.1 Å (Supplementary Fig. 2), with local resolutions fluctuating between 3 and 10 Å. Unsurprisingly, the PK ring is the least well-resolved part of the complex. The H5 stem-loop, TLD, SmpB and some parts of PK2 are observed at resolutions better than 3.5 Å, allowing for detailed molecular description of the interactions between tmRNA, SmpB and the ribosome. In contrast with the structure recently described by Rae et al.[17], our structure also includes a tRNA[phe] in the P site, and no tRNA in the E site.

When compared to the PRE-ACC state, the PK ring is in the same position around the beak, but is now larger (96 × 128 Å). As the H5 stem-loop moves towards the ribosome (Fig. 2d and Supplementary Fig. 13), PK2 interacts more closely with the KH2 RNA-binding domain of uS3 (Fig. 2e), confirming previous data by Rae et al.[17]. Indeed, the hydrophobic patch formed between uS3's third α-helix (residues Pro73 to Ile77) and its third β-sheet (residues Ile101 to Glu104) stabilizes PK2 nucleotides C183 to

A186 (Fig. 2e), while uS3's Arg72 forms an ionic interaction with U131. The PKs are better outlined here than in the PRE-ACC state (Supplementary Fig. 3), which suggests that at this point they are more relaxed and stable. The MLD is mostly stretched, and presents a single dense region at its centre that is compatible with the presence of a previously described hairpin between nucleotides U88 and A100[33], a pairing which may protect the resume codon until it is used (Supplementary Fig. 14). The TLD acceptor arm swings into the PTC and interacts with the 50S subunit (Fig. 1b). After analysing the PTC density, we concluded that the transfer of the P-site tRNA phenylalanine to the incoming tmRNA alanine had already occurred (Supplementary Fig. 15). The N-terminal arm of bL27, known to play a critical role in tRNA substrate stabilization during the peptidyl transfer reaction[34–37], is well-resolved, which allowed us to build a complete atomic model. It extends into the PTC, where it contacts both the P-site tRNA[phe] and the TLD (Supplementary Fig. 15). Interestingly, it was recently suggested that a rotated conformation of bL27 is preferred when non-stop ribosomes are bound by KKL-2098, a newly proposed *trans*-translation inhibitor[38]. The N-terminal arm is usually quite flexible but it is particularly well-resolved in the current ACC conformation. This may imply that *trans*-translation specifically requires the N terminus of bL27 to be oriented towards the PTC. It would

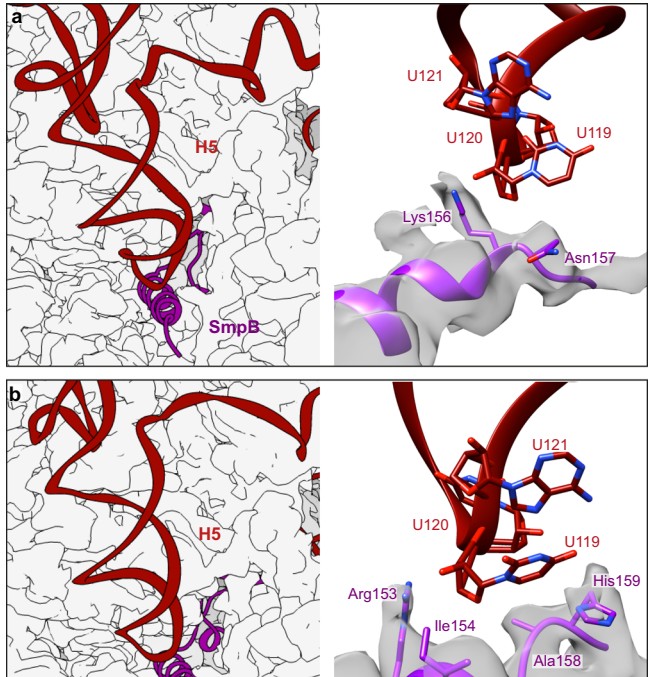

**Fig. 5 Detailed view of the interactions between the C-terminal tail of SmpB and the H5 stem-loop of tmRNA.** Left: position of the H5 stem-loop at the entrance of the mRNA channel. Right: focus on the residues involved in the interactions between SmpB and H5 during the **a** pre-accommodation and **b** accommodation states. SmpB is purple, tmRNA is red and the surface of the ribosomal small subunit is light grey. Residues and nucleotides within 4 Å of each other are indicated, and the cryo-electron density map of SmpB is displayed.

explain why KKL-2098 stabilisation of the rotated orientation of bL27 impedes *trans*-translation but not canonical translation.

The TLD T-arm mimics tRNA and interacts with uL16 and the 23S rRNA helices H38 and H89 (Supplementary Fig. 16). The contacts between the TLD D-loop and SmpB are maintained during this state, but the body is rotated to allow for TLD accommodation on the A site (Supplementary Fig. 8b). This results in a further insertion of Trp122 into the previously discussed hydrophobic pocket, where it interacts strongly with tmRNA nucleotides A334 and U17 (Supplementary Fig. 9a, middle). The tmRNA conserved nucleotide G19 stays at the surface of SmpB (Supplementary Fig. 9b, middle). In the DC, interactions with the conserved 16S rRNA nucleotides G530 and C1054 are also maintained and even reinforced, with Lys138 binding with G530, and His136 interacting with both C1397 and G530 (Fig. 3b and Supplementary Fig. 11b). In addition, the conserved nucleotide A1492 is now in an intermediate state inside the 16S rRNA h44 helix, whereas A1493 is outside and completely flipped, stacking on the SmpB's residue His22 (Fig. 3b). This conformation is very similar to the structure of a translating *E. coli* ribosome with no tRNA in the A site[39] (Supplementary Fig. 10b, e, as well as what is seen in the previous structure from Rae et al.[17], although the His22 position differs (Supplementary Fig. 10b, h). The tmRNA H5 stem-loop still interacts with helicases uS4 and uS5, and it is even more tightly packed on the C-terminal domain of uS3 (Fig. 2f). The interactions with uS3 residues Arg132, Lys135 and Arg136 and uS4 residue Arg47 are maintained, while new interactions are observed with uS3 Arg72, Asn140 and Arg143. However, both uS3 Arg131 and uS4 Arg44,

which were previously shown to be critical for helicase activity[25], point away from the helix. SmpB's C-terminal tail has the same position and folding as are observed during the PRE-ACC state (Fig. 1f), and maintains the same interactions with uS5 ribosomal protein and 16S rRNA, with the addition of an interaction between Lys156 and A509 (Fig. 5b). However, tmRNA's H5 loop interacts even more tightly with the extremity of the C-terminal tail of SmpB. Arg153 and His159 interact with the phosphate groups between U119, U120 and A121, while Ile154 and Ala158 join with uS5 to form a hydrophobic pocket in which U120 resides (Fig. 5b).

**Translocation step.** A third experiment in the presence of EF-G permitted us to visualize two different states, which occur after the translocation of tmRNA-SmpB to the P site. The first of the resulting structures shows the translocation state (TRANS) resolved at a resolution of 3.2 Å (Fig. 1c and Supplementary Fig. 2), and is consistent with the previously described TRANS conformation[12,14,17]. The local resolution of the TLD and SmpB C-terminal tail is ~3.25 Å, the rest of SmpB and the MLD are ~3.5 Å, and the resolution of the H5 stem-loop and PK2 fluctuates between 3.75 and 5 Å. The map's resolution allowed us to build a robust and detailed atomic model that includes crucial portions of the MLD. The tmRNA PK ring is now fully distorted but remains well-outlined (Supplementary Fig. 3), with a distance of about 130 Å between PK1 and the top of the H5 stem-loop. PK1 is pulled between the two subunits and lies on the tip of the H38 helix of the 23S rRNA, while the tmRNA H2 helix interacts with uS19. PK2 still strongly interacts with uS3's KH2 RNA-binding domain (Fig. 2g) while the pulling on the MLD moves the H5 stem-loop away from uS4 and uS5 (Supplementary Fig. 13). The tip of the helix (nucleotides U119, U120 and A121) now interacts with uS2's Trp104, Arg108 and Leu157, while C127, U128 and G129 interact with uS3's Arg143 and Leu144 (Fig. 2i). The MLD is stretched and inserted into the mRNA channel (Fig. 1g). It interacts with the helicases uS3, uS4 and uS5 in a way that resembles the binding of structured mRNAs (Supplementary Fig. 17)[27,28]. The contacts observed during the PRE-ACC and ACC steps are maintained. However, the MLD also interacts with the uS3 α5 helix (Gln123, Arg126 and Arg127) and β5 sheet (Ile162, Arg164, Glu166) and lies on the surface of uS5 (Phe31, Glu55 and Val56). The tmRNA nucleotide A97 interacts with Arg131 and is stacked on uS5 Val56, which puts it right at the centre of the proximal helicase active site[40]. The first resume codon's G90 nucleotide is stacked on the conserved 16S rRNA nucleotide A1493, while A1492 remains inside the 16S rRNA's helix h44 (Fig. 3c and Supplementary Fig. 10c, i). G530 interacts with the third resume codon's nucleotide A92 (Fig. 3c), which is, however, not yet in a conformation compatible with the codon–anticodon interaction observed during canonical translation[30] (Supplementary Fig. 10c, f). This precise positioning is made possible by direct interactions between the four nucleotides just upstream from the MLD resume codon and SmpB, as previously suggested by biochemical and structural studies[17,41–43] (Fig. 6). Indeed, SmpB's Tyr55 and Tyr24 residues are instrumental in correctly positioning the tmRNA resume codon in the A site. Tyr55 stacks with tmRNA's A86 and serves as the foundation for the stacking of the four bases upstream from the resume codon. Tyr24 lies between tmRNA's A86 and G87, and by forming weak hydrogen bonds with both, it might serve as a secondary checkpoint, ensuring a finer control of the sequence and limiting the risk of frameshifting.

The TLD has moved to the P site and continues to interact strongly with the SmpB body (Supplementary Fig. 8c). SmpB's Trp122 residue is still deeply embedded in its hydrophobic pocket

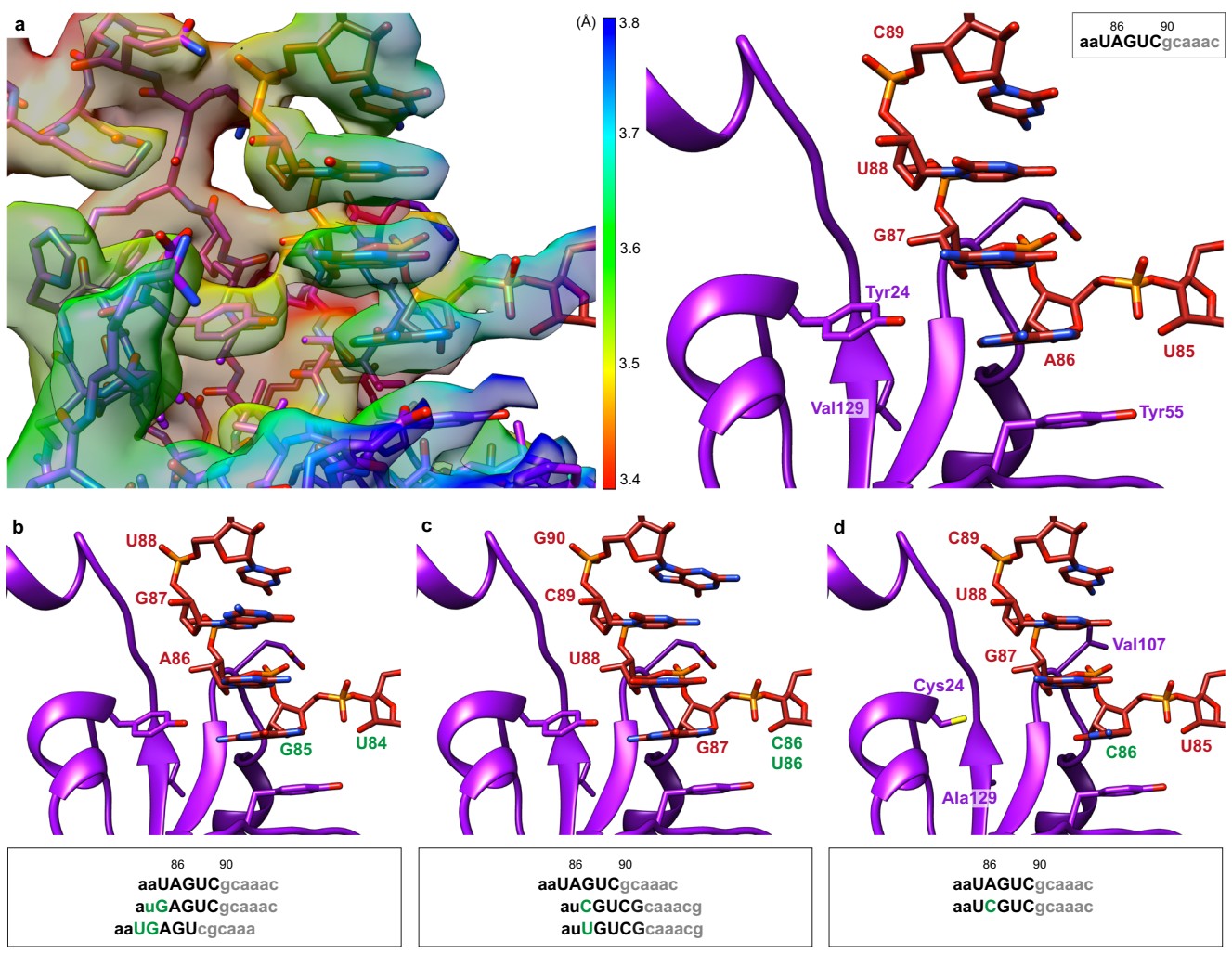

**Fig. 6 The structure of translocated tmRNA reveals how the right resume codon is selected. a** Left, focus on the interactions between SmpB (purple) and the four nucleotides just upstream of the tmRNA (red) resume codon. The cryo-electron density map is displayed and coloured according to the local resolutions as computed with ResMap[74]. Right, same but without the map. For clarity, cartoon representation is used for SmpB and only the four residues involved in the codon selection are shown. **b** An A84U/U85G double mutation in tmRNA maintains a high level of *trans*-translation but also promotes -1 frameshifting[41]. **c** Mutation of tmRNA's highly conserved A86 nucleotide into a pyrimidine nucleotide lowers the stacking interaction with SmpB Tyr55 and results in a +1 frameshift[41,42]. **d** An SmpB triple mutant (Y24C, E107V and V129A) can partially reverse the effect of the tmRNA A86C mutation, allowing for both +1 frameshifting and in-frame re-registration[43]. The boxes show part of the MLD sequence, with the wild-type sequence repeated at the top for reference. The first two codons are grey, the mutated nucleotides are green and the five nucleotides presented in the figure are in capital letters. Where appropriate, the sequences of the mutants are shifted to highlight the frameshifts caused by the mutations.

(Supplementary Fig. 9a, right), while the tmRNA G19 nucleotide is tightly packed at the surface of the first β-sheet of SmpB (Supplementary Fig. 9b, right). During translocation, the SmpB C-terminal tail rotates by 62° to make the DC available to the MLD resume codon. It retains its helical structure and binds the mRNA channel in the E site (Fig. 1g), where it is stabilized through different interactions with 16S rRNA (Fig. 4c). It is noteworthy that the two conserved histidines His22 and His136, which were involved in the DC during the PRE-ACC and ACC states, have in fact a dual purpose, as they now also help position the tmRNA-SmpB complex in the P site. Indeed, His22 now stacks with 16S rRNA's nucleotide A790, while SmpB's His136 is instrumental in positioning the tail through new stacking interactions with 16S rRNA C1400 (Fig. 4c). In doing so, SmpB mimics the way in which a P-site tRNA anticodon loop interacts with both mRNA and 16S rRNA (Fig. 7).

The SmpB C-terminal tail is further stabilized in the mRNA path by a series of interactions that occur between the charged residues Lys143, Arg145 and Lys156 and the 16S rRNA phosphates G926, A790 and G693, respectively (Fig. 4c). Arg139 is also instrumental, as it forms hydrogen bonds with both C1399 and G1401 (Fig. 4c and Supplementary Fig. 11c). This strong anchoring of the SmpB C-terminal tail into the mRNA exit tunnel is essential as it enables the truncated mRNA to be ejected from the ribosome. As previously described[16], the translocation process is assisted by the swivel of the 30S subunit's head (Supplementary Movie 1), which allows the tmRNA H2 helix to cross the B1a bridge. However, there is no EF-G in our structure, and we observed only a moderate swivel, with a head-to-body rotation of 5.7° and a head tilt of 3.7° (Supplementary Table 2).

**New intermediate translocation step yields insights into tmRNA-SmpB movement within the ribosome.** Another population obtained from the same dataset allowed us to identify

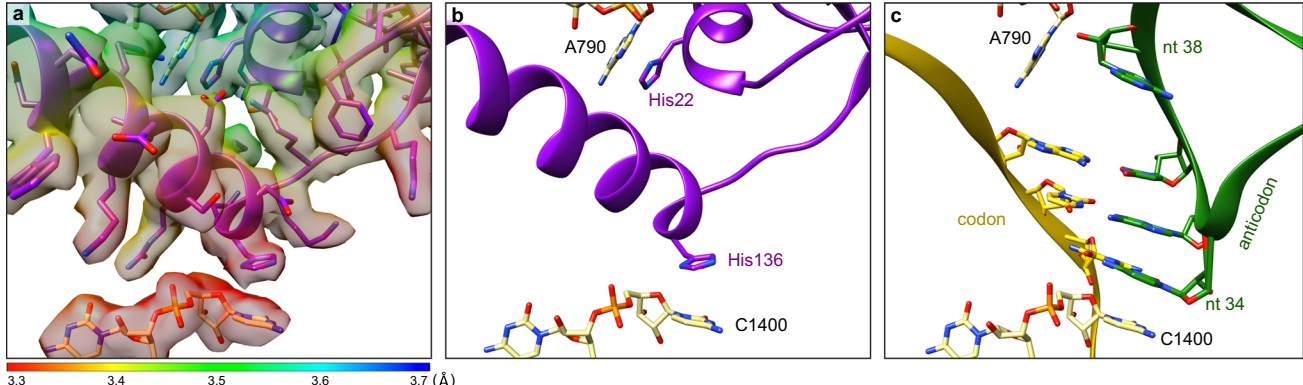

**Fig. 7 SmpB mimics the interactions of a tRNA anticodon loop in the P site. a** Focus on the interactions between SmpB and the 16S rRNA P-site in the translocation state. The cryo-electron density map is displayed and coloured according to the local resolutions as computed with ResMap[74]. **b** Same as (**a**), but without the map. For clarity, cartoon representation is used for SmpB and only His22 and His126 are shown. **c** Comparison with a P-site tRNA[30] (PDB code 7K00). The SmpB C-terminal tail occupies the mRNA path, replacing the codon–anticodon interaction. SmpB also reproduces the anticodon loop interactions with the 16S rRNA. The stacking interaction between His22 and A790 replaces the interaction usually observed with the sugar of tRNA's nucleotide (nt) 38. Meanwhile, His136 replaces the anticodon's nucleotide 34, stacking on the C1400 rRNA nucleotide. SmpB is purple, tRNA is green, mRNA is yellow and 16S rRNA is khaki.

a new intermediate state, which we named 'TRANS*' (Fig. 1d). Seen at a resolution of 3.4 Å (Supplementary Fig. 2), it occurs after TRANS and just before tmRNA-SmpB exits the P site.

In canonical translation, before translocation, the two ribosomal subunits spontaneously rotate by 8–10° with respect to each other, in a ratchet-like motion. This moves the tRNAs into the hybrid A/P and P/E states (in these states, the first letter refers to the position of the tRNA on the 30S subunit and the second to its position on the 50S)[44]. The 30S head domain also undergoes a partial (5–7°) forward rotation with respect to its body[45]. EF-G binding then triggers a swivel, a larger ~21° rotation and moderate ~3° tilt of the 30S head[16,46]. This clears the way for the translocation of tRNA between the P and E sites on the small subunit, a path that would otherwise be constricted by rRNA nucleotides in the head and platform regions[47]. In the TRANS* structure, the rotation of the 30S head is rather large (~14°), but still less than that observed in the presence of EF-G·GDP and tRNAs[48,49], EF-G·GDP and tmRNA[16], or with a frameshift-prone tRNA[50] (Supplementary Table 2). This surely reflects a back-swivelling motion of the 30S head after the release of EF-G. The presence of the TLD in the P/P state clearly shows that TRANS* is a post-translocational intermediate state. Interestingly, the ~12° head tilt is much larger than in any other structure except that of the tmRNA-SmpB-EF-G complex on the ribosome[16]. This demonstrates that the presence of tmRNA-SmpB in the P site is sufficient to strongly tilt the head, and this prompted us to revisit how tmRNA-SmpB passes the ribosomal bridges[51] (Fig. 8).

In the PRE-ACC state, the tip of the 50S helix H38 is unstructured. The B1a bridge is wide open, but the B1b/c bridges are closed (Fig. 8a). In the ACC state, the tmRNA H2 helix follows the movement of the TLD and is inserted between the two ribosomal subunits. The H38 helix of the 50S is well-ordered, and all three B1 bridges are closed[13] (Fig. 8b). Next, in TRANS, the tmRNA H2 helix moves towards the other side of B1a as B1b begins to open (Fig. 8c). The TRANS* state displays a much larger 30S head rotation and tilt (+8° each) (Fig. 8d). While the TLD-SmpB complex is mostly unaffected by this movement, the tmRNA H2 helix is slightly bent, and the entire PK ring rotates along with the head. This results in a destabilisation of the MLD at the entrance of the mRNA channel (Supplementary Fig. 18). Furthermore, the B1b and B1c bridges are now wide open, unlike during canonical translation (Supplementary Movie 1). We hypothesize that the head's large tilt is due to its interaction

with the tmRNA H2 and PK1 conserved domains, and that this movement is instrumental in helping tmRNA to pass the intersubunit bridges on its way towards the E site. However we could not detect structures showing SmpB and/or the TLD in the E site. In fact, Rae et al.[17] recently suggested that a stable E-site intermediate was unlikely, due to induced clashes with the ribosome. We can therefore assume that tmRNA-SmpB just transits through quickly. Indeed, in our TRANS* structure the SmpB C-terminal tail is still tightly interacting with the ribosome in the mRNA path (Fig. 1h). As the tmRNA-SmpB complex continues to travel towards the E site, the C-terminal tail will either have to move along the mRNA path (between uS7, uS11 and bS21 and the 16S rRNA helices H28 and H45), tightly anchoring SmpB in the E site, or else the tail will have to stay in place, forcing SmpB to fold on itself. Together with the observation of a large opening between the two ribosomal subunits in the TRANS* conformation, this suggests that the TLD and SmpB are promptly ejected from the E site as translation continues on the MLD. The complex would then reach the post-E conformation described by Rae et al.[17].

### Discussion

In this study, we present four cryo-EM structures of the *trans*-translation machinery, confirming previous structural data but also providing details of the interactions between tmRNA, SmpB and the ribosome throughout the process of *trans*-translation. These shed light on how the tmRNA-SmpB complex recognizes stalled ribosomes, how it selects the codon needed to resume translation of the tag and how it crosses the various ribosomal bridges without interfering with canonical translation.

One significant finding is that the SmpB C-terminal tail is not the only sensor used by *trans*-translation for stalled ribosomes detection. Indeed, in both the pre-accommodation and accommodation structures, the tmRNA H5 stem-loop plays a crucial role, interacting with the end of the SmpB C-terminal tail and closing off access to the mRNA channel. The positions of both the C-terminal tail and the stem-loop are incompatible with a ribosome undergoing canonical translation, because a steric clash would occur with a non-truncated mRNA (Fig. 9). This conclusion is further supported by the observation that although the deletion of the last five SmpB residues has no effect on tmRNA-mediated tagging, the tagging is severely reduced with the deletion of two additional residues (Met155 and Ile154) or by their

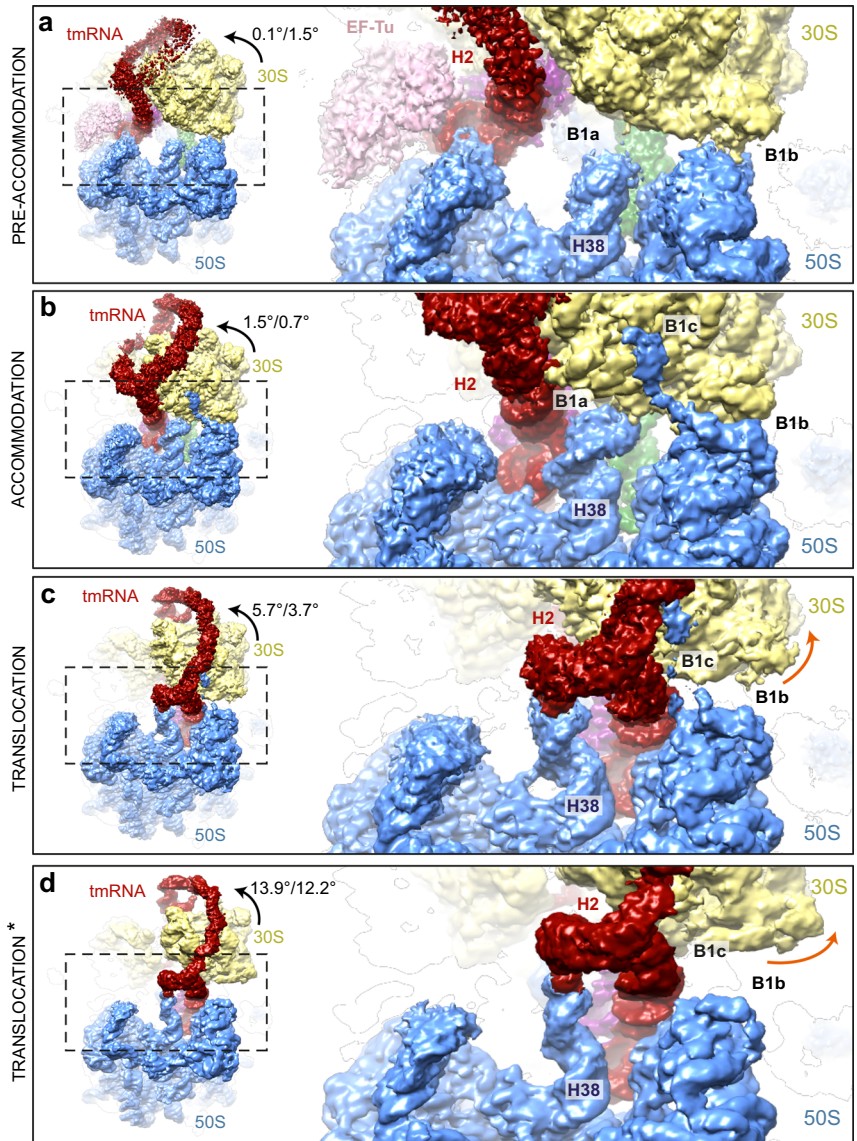

**Fig. 8 Transit of tmRNA through the B1a, B1b, and B1c bridges during *trans*-translation.** Close-up view of the opening and closing of bridges B1a, B1b and B1c during **a** pre-accommodation, **b** accommodation, **c** translocation and **d** intermediate translocation (TRANS*) states. The 50S ribosomal subunit is blue, 30S is khaki, elongation factor EF-Tu is pink, tmRNA is red, SmpB is purple and tRNA[phe] is green. The tmRNA helix H2 and the 50S helix H38 are also labelled. Black arrows indicate the degree of rotation and tilt of the 30S head measured with respect to its body, as per Nguyen and Whitford[87]. Orange arrows highlight the opening of bridges B1b and B1c. All maps are contoured at 2.5 $\sigma$, where sigma represents variance in the map.

simultaneous mutation into negatively charged residues[52]. The SmpB residue Ile154 seems to be of particular importance, as its mutation into proline is sufficient to impede tmRNA tagging[53]. The fact that Met155 and Ile154 mutations impair tagging but not binding suggest that these residues are mandatory for the correct positioning of the resume codon within the DC. As the H5 stem-loop interacts with SmpB exactly in this region (Fig. 5), we hypothesize that the contact between H5 and the C-terminal tail provide a second anchoring point (the first being the SmpB–TLD interaction) and this facilitates the correct positioning of the resume codon during translocation. We also confirmed that the tmRNA H5, PK2 and MLD domains make numerous contacts with the uS3, uS4 and uS5 proteins at the entrance of the mRNA channel. These three proteins are known to be instrumental in ribosome helicase activity[25]. In the PRE-ACC and ACC states, the H5 stem-loop strongly interacts with residues in the uS3 α6 helix, uS4 α1–α2 linker and uS5 α1 helix (Supplementary Fig. 17a, b).

Interestingly, this mostly involves the same positively charged residues that have been shown to interact with structured mRNAs[27,28]. These notably include uS3 residues Arg132 and Lys135, and uS4 Arg47, all of which are known to be critical for helicase activity[25]. While H5 lies on the distal helical active site, it is not inserted deep enough in the mRNA channel to reach the proximal active site[40]. This may explain why the H5 stem-loop remains highly structured at these early stages. However, because of its strong interaction with the 30S, H5 could play the role of a fulcrum, helping with the mechanical unfolding and correct positioning of the MLD in the mRNA channel during translocation. In the TRANS state, H5 is flipped towards uS2 and replaced by the single-stranded portion of the MLD in a manner resembling the binding of structured mRNAs[27,28] (Supplementary Fig. 17c, d). The backbone of the single-stranded portion of the MLD is extended and straightened, and SmpB helps properly place the resume codon in the A site. This brings the tmRNA

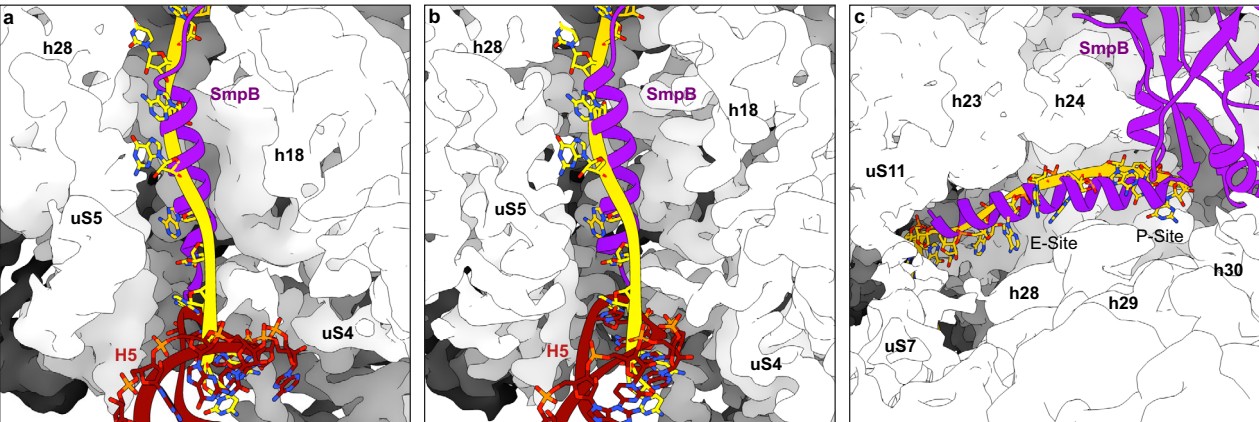

**Fig. 9 Close-up of the mRNA path during *trans*-translation.** In the **a** pre-accommodation and **b** accommodation states, the C-terminal tail of SmpB (purple) occupies the mRNA channel, while the H5 stem-loop of tmRNA (red) closes the entrance. Both domains would clash with a canonical mRNA (yellow, PDB code 6ZTJ[88]), forcing the selection of stalled ribosomes. **c** After translocation, the C-terminal tail is rotated by 62° and deeply inserted into the mRNA exit tunnel. There again, it clashes with the mRNA (PDB code 7JT1[89]), resulting in the ejection of the non-stop mRNA from the ribosome. The cryo-electron density map of the 30S small subunit is shown as a white surface.

nucleotide A97 (the 11th nucleotide when counting from the P site) to the centre of the helicase proximal active site, while part of the MLD interacts downstream with the distal active site. This confirms that the uS3 protein favors binding of extended single-strand mRNAs at the entrance of the tunnel, and is consistent with the hypothesis that product stabilization plays a role in the unwinding of structured mRNAs by the ribosomal helicase[28,40]. While the H5 helix is yet to be unfolded, the comparison between the TRANS and TRANS* states also provides some information on the helicase mechanism (Supplementary Fig. 18). Indeed, the most apparent difference between these two states is an extra 8° of both rotation and tilt of the 30S head in TRANS* (Supplementary Table 2). Although this state is not as well-resolved as TRANS, the PK2, H5 and MLD domains remain quite well-defined, except for a small portion of the single-stranded MLD right in the proximal helicase active site. This suggests that the forward head rotation and tilt lengthen the mRNA channel and destabilize the MLD. The canonical translation could then resume, allowing for the unfolding of H5 (which contains a large portion of the tagging sequence, see Supplementary Fig. 1) during the ensuing steps. Together, our structures confirm the involvement of the positively charged residues of uS3 and uS4 in both the basic mRNA-binding activity of the ribosome as well as its helicase activity, and they are consistent with the recent tandem model for the ribosomal helicase[40].

During translocation, SmpB also plays a major role in ensuring that the resume codon is correctly positioned within the DC. Our TRANS structure unambiguously shows how SmpB binds to the four nucleotides just upstream of the MLD resume codon (Fig. 6). The two residues, Tyr24 and Tyr55, are of particular importance in this interaction. First, the residue at position 55 is always an aromatic amino acid[54], and it serves as the foundation for the stacking of the four bases upstream of the resume codon. Because the stacking of amino acids is more stable on purines than on pyrimidines[55], tmRNA has evolved to strongly favour purine at position 86 and pyrimidine at position 85 to avoid −1 frameshifting[56]. Second, the highly conserved Tyr24 lies between the two tmRNA nucleotides A86 and G87. By forming weak hydrogen bonds with both, it certainly serves as a secondary checkpoint to ensure a finer control of the sequence and to limit the risks of frameshifting. Indeed, the steric repulsion between

tyrosine's hydroxyl group and guanine's amine group explains the clear preference for A over G at position 86, while the possibility of forming a hydrogen bond between the same hydroxyl group and the N3 explains the slight preference for purine instead of pyrimidine at position 87[56]. Taken together, these observations explain why the A84U/U85G double mutation not only maintains a high level of *trans*-translation, but also promotes −1 frameshifting[41] (Fig. 6b), as well as why the mutation of A86C and A86U in tmRNA impedes *trans*-translation and promotes a +1 frameshifting of the MLD[41,42] (Fig. 6c). Our TRANS structure also help us to understand how the triple mutation of Y24C, E107V and V129A in SmpB can partially reverse the effects of the A86C mutation of tmRNA (Fig. 6d). Indeed, by simultaneously altering the sequence checkpoint (mutation Y24C), the shape of the binding pocket (mutation V129A) and the positioning of the tmRNA backbone (mutation E107V), the triple mutation facilitates the shifting of the MLD at the surface of SmpB, allowing for both in-frame and +1 frame re-registration[43].

While passing through the ribosome, the SmpB C-terminal tail remains folded in an α-helix. However, it is always tightly fixed inside the mRNA channel thanks to its positively charged residues (Fig. 4). Among these, the conserved DKR motif (Asp137, Lys138, Arg139) at the base of the C-terminal tail is of particular interest. Indeed, while single mutations in this motif have only a marginal effect, the substitution of alanine for all three residues completely abolishes *trans*-translation[52,53]. Interestingly, we observed that these residues work together to stabilize SmpB in the DC, with Asp137 binding to C1397, Lys138 to G530 and Arg139 to A1196 (Supplementary Fig. 11a, b). These interactions are completed by Lys134 (which binds to C1054), Gln135 (which stacks with G530) and His136 (which plays a central role, binding both G530 and C1397). Contrary to what was described by Rae et al.[17], we only observed a moderate stacking interaction between His22 and A1493, and no stacking of His136 and G530. This certainly explains why *trans*-translation is much less sensitive than canonical translation to DC mutations[53]. After translocation to the P site, the SmpB C-terminal tail rotates by 62° and occupies both the P- an E-site parts of the mRNA path. In this way, SmpB can forcefully eject the non-stop mRNA from the ribosome. While Rae et al.[17] emphasized the role of SmpB's Gly132 conserved residue in this rotation, the nature of the interaction that

correctly positions SmpB in the P site was not clear. Our structure shows that this is made possible not only by the charged residues of the C-terminal tail, but also by the two histidines His22 and His136, which mimic the interactions of a tRNA stem-loop with the 16 rRNA.

Our structures also allow for a better understanding of the molecular basis of tmRNA-SmpB interactions throughout *trans*-translation. Among the different contacts formed between the two partners (Supplementary Fig 8), it is worth noting the continuous stacking of tmRNA nucleotide G19 on the conserved hydrophobic patch (Leu90, Leu91, Leu92) on the surface of SmpB, as it has been reported that a G19C mutation in tmRNA abolishes *trans*-translation[24]. Decreasing the stacking interaction between C18, G19 and the SmpB surface also affects the binding of Trp122 in its hydrophobic pocket and destabilizes the entire tmRNA-SmpB complex, effectively preventing *trans*-translation.

To conclude, *trans*-translation is an appealing antibiotic target since it is essential for fitness and vital to many pathogens, yet absent in eukaryotes. Our structures give a better understanding of how *trans*-translation occurs, but they also showcase its differences and similarities with canonical translation, and confirm the fact that it does not interfere with that process. We hope that our detailed description of the interactions between the accommodated tmRNA-SmpB and the ribosome, as well as the newly described interface between the MLD and SmpB in the TRANS conformation, might be useful to develop the recently proposed anti-*trans*-translation strategies[38,57–59].

## Methods

**Ribosome purification**. Ribosomes were purified from the *E. coli* strain MG1655. When the culture reached an OD$_{600}$ of 0.8, the cells were pelleted, resuspended in FP buffer (20 mM Tris-HCl pH 7.5, 50 mM MgOAc, 100 mM NH$_4$Cl, 0.5 mM EDTA and 1 mM DTT), then lysed in a French press. The lysate was then clarified by centrifugation at 20,000 × *g* for 45 min at 4 °C. Next, the supernatant was layered 1:1 (v:v) over a high-salt sucrose cushion buffer (10 mM Tris-HCl pH 7.5, 10 mM MgOAc, 500 mM NH$_4$Cl, 0.5 mM EDTA, 1.1 M sucrose and 1 mM DTT). After ultracentrifugation at 92,000 × *g* for 20 h at 4 °C, the resulting ribosome pellets were resuspended in 1 mL of 'Ribo_A' buffer (10 mM Tris-HCl pH 7.5, 10 mM MgCl$_2$, 50 mM NH$_4$Cl, 0.5 mM EDTA and 1 mM DTT). To isolate the 70S ribosomes from 30S and 50S ribosomal subunits, the ribosomes were centrifuged at 95,000 × *g* for 18 h at 4 °C through a 10–45% (w/w) linear sucrose gradient in Ribo_A buffer. The different gradients were fractionated before determining the A$_{260}$ absorbance profiles. Fractions corresponding to the 70S peak were mixed and diluted in Ribo_A buffer for a final ultracentrifugation at 92,000 × *g* for 20 h at 4 °C. The ribosomal pellets were resuspended in Ribo_A buffer, then stored at −80 °C.

**RNA purification**. The tmRNAs were purified in native conditions as previously described[60]. We used a synthetic non-stop mRNA (Thermo Fisher Scientific) with the sequence 5′-AGGAGGUGAGGUUUU-3′ (the Shine–Dalgarno sequence is underlined and the phenylalanine codon is bold). Phenylalanine-specific *E. coli* tRNA was purchased from Sigma. The *E. coli* tRNA$^{Ala}$ was transcribed in vitro from the plasmid pUC19-ala-tRNA using a MEGAscript T7 transcription kit (Thermo Fisher Scientific)[61]. Before transcription, the plasmid was linearized by B*st*NI restriction enzymes, purified with purified phenol/alcohol/chloroform, then precipitated using 3 M sodium acetate.

**Protein purification**. To generate an SmpB with a His-tagged C-terminus, the *E. coli smpB* gene was cloned between the N*de*I and X*ho*I restriction sites in pET22b (+) vector. The protein was expressed in *E. coli* BL21(DE3)Δ*ssrA* cells[62] and purified as previously described[58]. T7 expression system was used to overexpress his-tagged AlaRS, his-tagged phenylalanyl-tRNA synthetase (PheRS) and his-tagged elongation factor EF-Tu in *E. coli* BL21(DE3) strain. These were then isolated using a HisTrap HP column (GE Healthcare) as previously described[63]. His-tagged EF-G was overexpressed from pQE60 vector in *E. coli* BL21/pREP4. This overexpression was induced by 1 mM IPTG at an OD$_{600}$ of 0.6 over 3 h at 37 °C. The cells were lysed in a French press and the lysate was applied to a HisTrap HP column equilibrated with 'EF-G-I buffer' (50 mM NaH$_2$PO$_4$/Na$_2$HPO$_4$ pH 8, 300 mM NaCl and 20 mM imidazole). The column was washed in that buffer and the protein eluted with a linear gradient of imidazole going from 20 to 300 mM. The elution fractions were concentrated with an Amicon Ultra-15 filter (Sigma) in 'EF-G-II buffer' (10 mM HEPES-KOH pH 7.5, 500 mM KCl, 0.5 mM EDTA and 10% glycerol) then stored at −20 °C. To avoid ribosomal degradation in vitro, the plasmid pABA-RNR_D280N was used to express the his-tagged RNase R mutated

on its catalytic site[64] in *E. coli* BL21(DE3) cells. After lysis in a French press in 'RNR-FP buffer' (50 mM NaH$_2$PO$_4$/Na$_2$HPO$_4$ pH 8, 300 mM NaCl and 10 mM imidazole), the lysate was filtered and applied to a HisTrap HP equilibrated with the same buffer. The column was then washed with an 'RNR-A buffer' (50 mM NaH$_2$PO$_4$/Na$_2$HPO$_4$ pH 8, 300 mM NaCl and 20 mM imidazole) and an 'RNR-B buffer' (50 mM NaH$_2$PO$_4$/Na$_2$HPO$_4$ pH 8, 1 M NaCl and 20 mM imidazole). The mutated RNase R was eluted in RNR-A buffer with a linear gradient of imidazole from 20 to 500 mM. The fractions corresponding to the mutated RNase R were concentrated using an Amicon Ultra-15 in 'RNR-C buffer' (10 mM HEPES-KOH pH 7.5, 500 mM KCl, 0.5 mM EDTA and 50% glycerol), then stored at −20 °C.

**Preparation of ribosomal complexes**. In order to obtain the most *trans*-translation states from the A to P sites in the ribosome, three different complexes were prepared. These were all obtained using *E. coli* components, to which we added RNase R (a known partner of *trans*-translation) in order to reproduce native conditions as closely as possible.

For the first complex, we incubated 50 pmol phenylalanine-tRNA (phe-tRNA) for 2 min at 80 °C. This was followed by a second incubation at room temperature for 30 min in 'Buffer I' (10 mM HEPES-KOH pH 7.5, 5 mM MgCl$_2$ and 20 mM NH$_4$Cl). Next, 250 pmol EF-Tu·GDP were activated into EF-Tu·GTP immediately before it was incubated with 2 mM GTP in 'Buffer II' (62.5 mM HEPES-KOH pH 7.5, 9 mM MgCl$_2$ and 75 mM NH$_4$Cl) for 15 min at 37 °C. Aminoacylation of tmRNA was performed by mixing 125 pmol tmRNA, 250 pmol EF-Tu·GTP, 125 pmol SmpB, 190 pmol AlaRS, 2 mM ATP, 30 μM alanine and Buffer II. We then blocked the ribosome on non-stop mRNA by mixing 25 pmol *E. coli* 70S, 50 pmol non-stop mRNA and 100 pmol folded phe-tRNA in 'Buffer III' (5 mM HEPES-KOH pH 7.5, 9 mM MgOAc, 50 mM KCl, 10 mM NH$_4$Cl and 1 mM DTT) for 15 min at 37 °C. Finally, the complex was created by mixing the blocked ribosomes with the aminoacylated tmRNA supplemented by 100 pmol SmpB and 250 μM kirromycin to trap EF-Tu in its GDP-bound conformation. This prevents the release of tmRNA, and was done as previously described[15] for 15 min at 37 °C. We then added 250 pmol RNase R for an additional 10 min.

The second complex was obtained exactly like the first, but without EF-Tu. For the third complex, phe-tRNA and alanine-tRNA (ala-tRNA) were folded as described above. After the folding step, either 50 pmol ala-tRNA or 20 pmol phe-tRNA was aminoacylated by mixing 2 mM ATP, 30 μM of the corresponding amino acid, 80 pmol of the corresponding aminoacyl-tRNA synthetase, and Buffer II. In parallel, tmRNAs were also aminoacylated and ribosomes blocked as described above, this time using 50 pmol phe-tRNA. The complex was completed by incubating the stalled ribosomes, the aminoacylated tmRNA and EF-G (at a 1:3 ratio between the ribosome and EF-G) for 10 min at 37 °C, after which ala-tRNA$^{ala}$ was added and the mix was left to incubate for 15 min in order to translocate the tmRNA in the ribosome.

**Electron microscopy**. After adjusting concentrations to 100 nM in Buffer III, samples were directly applied to glow-discharged holey carbon films (Quantifoil 2/ 2 μm). These grids were then flash-frozen in liquid ethane using a Vitrobot Mark III (FEI), then transferred to a Cs-corrected Titan Krios electron microscope (FEI) operating at 300 kV and equipped with an FEG electron source. Images were automatically recorded using SerialEM[65] under low-dose conditions on a K2 direct electron detector (Gatan) using a defocus range of −1 to −3 μm, a nominal magnification of ×105,000, and a final pixel size of 1.1 Å (complex 1), or a defocus range of −0.7 to −2 μm, a nominal magnification of ×130,000, and a final pixel size of 1.04 Å (complexes 2 and 3). Movies were corrected for the effects of drift and beam-induced motion using MotionCor2 software[66]. Contrast transfer function (CTF) parameters were estimated using Gctf software[67]. Electron micrographs showing signs of drift or astigmatism were discarded, and for complexes 1 through 3, this resulted in respective datasets of 3143, 10,484 and 11,433 images. Particles were semi-automatically selected in Cryosparc[68] and subjected to two rounds of 2D classification in order to discard defective particles. This resulted in 59,016 (dataset 1), 373,247 (dataset 2) and 207,135 particles (dataset 3). All subsequent data processing was performed in RELION[69,70]. Further 3-D autorefinements with a soft circular mask (diameter 380 Å) produced initial cryo-EM reconstructions for the three datasets at overall resolutions of 4.27, 3.84 and 3.24 Å, respectively. To improve the homogeneity, datasets 2 and 3 were then sorted into ten subsets using the 3D classification function. The classes that were clearly homogenous 70S ribosomes were selected, resulting in the retention of 238,808 (dataset 2) and 174,863 particles (dataset 3) for the next round of analysis.

After CTF refinement and particle polishing, the 'shiny' particles from all datasets were subjected to a second round of 3D auto-refinement. We then subtracted the signal of the ribosome from the datasets, using a soft mask (voxel values of 0 inside, 1 outside, extended by 6 pixels with a soft edge of 6 pixels) generated from the previous refinement run. The subtracted datasets were then sorted by a 3D classification without alignment[71]. Various numbers of classes were tested in order to split the datasets into as many 3D classes as possible while still keeping the groups homogeneous. This allowed for the separation of particles containing the tmRNA-SmpB complexes while limiting the bias induced by masking each factor separately. As a result, dataset 1 was separated into three classes: pre-accommodated ternary complexes (18,452 particles); poorly resolved accommodated tmRNA-SmpB complexes without EF-Tu and stalled ribosomes

without the tmRNA-SmpB complex. Dataset 2 was separated into five classes, with only one containing the tmRNA-SmpB complex accommodated in the A site (86,622 particles), and the rest having different conformations of stalled ribosomes and junk particles. Dataset 3 was also processed into five classes, separated between the tmRNA-SmpB complex accommodated in the A site (62,156 particles); two distinctive conformations of the tmRNA-SmpB complex translocated into the P site (with 14,192 particles in TRANS and 11,059 particles in TRANS*); stalled ribosomes (1 class) and junk (1 class). The particles from datasets 2 and 3 corresponding to the accommodated state were merged together and subjected to another round of 3D classification with signal subtraction of the empty ribosomes. After sorting the particles into four classes, one class of 36,069 particles was finally selected. We finally reverted to the original non-subtracted particles, and each step (PRE-ACC, ACC, TRANS, TRANS*) was reconstructed separately. The pixel size was then accessed by comparison with the atomic model of the *E. coli* mature 70S subunit[31] (PDB 4YBB) and adjusted to 1.074 (instead of 1.1 Å/pixel) for dataset 1. Further 3D auto-refinement and post-processing (with the adjusted pixel size) yielded maps with overall resolutions of 3.2 Å (PRE-ACC), 3.1 Å (ACC), 3.2 Å (TRANS) and 3.4 Å (TRANS*) based on RELION's gold-standard Fourier shell correlation calculation[72,73]. Local resolutions were estimated using Resmap[74], and map quality was analysed using Phenix mtriage software[75]. These consensus maps were then used as a basis for multi-body refinement[76,77] into three separate maps of the ribosomal large subunit, the body of the small subunit and the head of the small subunit, while including overlapping parts of the tmRNA-SmpB complex in each. The corresponding masks were made using a 30 Å low-pass filtered version of the consensus map and 12 Å soft-edges to define the solvent region boundary and to ensure that all the bodies overlapped. The resulting density maps were then sharpened using Phenix[78], and used with the consensus maps for building and refinement of the models.

**Model building and refinement**. UCSF-Chimera[79] was used to rigid-body fit the crystal structure (2.1 Å) of the *E. coli* (PDB 4YBB) mature 70S subunit[31] into the cryo-EM maps, with each protein and RNA treated separately. Atomic coordinates for ET-Tu were taken from PDB code 5AFI[80], while tmRNA and SmpB coordinates were modelled from 4V8Q[15] (PRE-ACC), 6Q97[17] (ACC) and 6Q98[17] (TRANS and TRANS*). All models were manually adjusted in the multi-body maps using COOT[81], with the exception of the MLD and PKs, which were first adjusted with MDFF[82] as follows. The system was set up in vacuo and subjected to energy minimization for 5000 steps (50 ps) to relax any steric clashes. To fit the atoms into the EM density, a production run of 100,000 steps was performed. The forces applied to the atoms were scaled by a factor of 0.3, and to prevent overfitting, harmonic restraints were applied so that the secondary structure using a force constant of 200 kcal mol$^{-1}$ rad$^{-2}$. Default values were used to restrain the hydrogen bonds, cis-peptide bonds and chiral centres. All steps were performed using VMD[83], NAMD2[84] and the CHARMM36[85] force fields. Once the structure was complete, Mg$^{2+}$ ions were manually added in COOT using the 'unmodelled blobs' function and a threshold of 5.5 RMSD. The atomic models were further improved by real space refinement against their respective consensus maps using Phenix. After a first refinement, outliers were manually corrected in COOT and the structure underwent a final refinement in Phenix, which produced the structures presented here. Models were evaluated with MolProbity[86–89] and the remaining analysis and the illustrations were done using UCSF-Chimera[79].

**Reporting summary**. Further information on research design is available in the Nature Research Reporting Summary linked to this article.

## Data availability

All of the data supporting the findings of this study are available within the paper and in the Supplementary Materials. The atomic coordinates and electron density maps have been deposited in the EMDB and PDB under the following accession codes, respectively: EMD-11710 and 7ABZ (pre-accommodated state); EMD-11713 and 7AC7 (accommodated state); EMD-11717 and 7ACJ (translocated state) and EMD-11718 and 7ACR (post-translocated intermediate state).

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

## Acknowledgements

This article is in memory of Professor Brice Felden. The authors gratefully acknowledge aid from the Agence Nationale pour la Recherche as part of the RIBOTARGET 18-JAM2-0005-03 project under the JPI AMR framework. Funding was also received from the French Direction Générale de l'Armement (C.G.), the Université de Rennes 1 (C.G.) and the European Union's ERASMUS+ program (G.U.), as well as from the Région Bretagne (G.U.). Thanks are also due for electron microscope use at the Centre de Microscopie et d'Imagerie de Rennes (MRic), the EMBL Heidelberg Cryo-Electron Microscopy Service Platform and the Integrated Structural Biology platform of the Strasbourg Instruct-ERIC center IGBMC-CBI supported by FRISBI (ANR-10-INBS-0005-001). We thank Juliana Berland and Felix Weis for their comments on the manuscript.

## Author contributions

C.G. performed the biochemical experiments and analysis. S.C. and C.G. performed the cryo-EM experiments. G.U. and E.G. performed the image analysis and model building. R.G. and E.G. supervised the project. All authors participated in writing the manuscript and approved its final version.

## Competing interests

The authors declare no competing interests.
