## [Peer Review File · Nature Communications]

Editorial Note: Parts of this Peer Review File have been redacted as indicated to remove third party material where no permission to publish were obtained

REVIEWER COMMENTS

Reviewer #1 (Remarks to the Author):

Summary

Although the movement of tmRNA-SmpB during trans-translation has been described in previous structural studies, the strength of this manuscript comes from the addition of two new high resolution structures, as well as the improvement in resolution of previously solved structures. This work provides important details about key events in trans-translation that were either previously unseen or impossible to describe due to insufficient local resolution. The experimentation was conducted expertly and the methodology used is sound.

While the work certainly deserves to be published, the manuscript could be improved considerably. It is thorough in its description of the structures, but its sequential format does not effectively tell a story. Major novel contributions: focuses on the most important new findings. Useful comparisons to canonical structures. -High-resolution full length PRE structure completes the story of the entrance of tmRNA-SmpB onto the ribosome and text, as suggested in the comments below.

-Improved resolution of TRANS structure for the first time shows detailed interactions of the MLD with SmpB to set the reading frame, an aspect critical to the fidelity of trans-translation.

Minor novel contributions:

-Visualization of a previously poorly resolved contact between the tail of SmpB and H5. This interaction could be important for initial positioning of the tail of SmpB during pre-accommodation, and for the transition from the PRE to ACC state.

-Higher resolution ACC structure gives a better atomic model for E. coli, particularly improving the P-site tRNA and peptidyl transferase center (PTC), which may be important for understanding recent trans-translation inhibitor binding data.

-A new TRANS* structure shows a 30S head conformation that further suggests h2 of tmRNA passes through the subunit interface during subsequent translocation out of the P site.

Comments

General note. By convention A-site with hyphen is used when saying, for instance, "the A-site tRNA", whereas A site without hyphen should be used when referring directly to the location in the ribosome, as in, "the tRNA bound in the A site".

General note. The author claims that previously unseen interactions between tRNA-SmpB and the ribosome explain why the process does not interfere with canonical translation however the figures completely lack illustration of this idea as no direct comparisons between trans-translation and translation are presented. Analysis as suggested below should be completed to illustrate this point.

INTRO

Line 49. In extended data figure 1 the MLD is the mRNA-like domain, not messenger-like domain. Please correct.

Line 65-66. Although a high-resolution structure of full-length tmRNA in the PRE state is required for a complete molecular description of tmRNA-SmpB loading onto the ribosome and its transition into the ACC state, how it is required for an understanding of the movement into the TRANS state is unclear. The claim should be edited to accurately reflect this.

RESULTS

PRE

Line 98. Extended data Fig 3 is unclear. To support the claim that EF-Tu is in the GDP bound state with kirromycin bound, please show the density of GDP and kirromycin alone without EF-Tu.

Line 107-108. The structure answers the open question of how during pre-accommodation, PK2 initially interacts with the small subunit as seen in all subsequent steps. This early interaction has never been seen at this resolution and provides conclusive data that not only the tail of SmpB, but also tmRNA is important for the initial detection of non-stop ribosomes, as PK2 and H5 binding in this position would be incompatible with long mRNA. The authors should highlight this finding and compare the position of H5 directly to that of the accommodated state as well as to mRNA during canonical translation.

Line 100. The authors claim that tmRNA resembles aminoacyl-tRNA in the PRE state. Please support this claim with data visualization, possibly by showing the reader an overlay of tmRNA-SmpB and tRNA in the PRE states.

Line 102. Extended data figure 4 has a labeling inconsistency. Middle panel Hing1 should be labeled Switch 1 as in extended data figure 3.

Line 110. The MLD has never been visualized outside of the ribosome and a putative hairpin may be present. If indeed the MLD is partly folded and density can be seen, this should be shown in a figure and described in more detail. This could confirm previous secondary structure analysis (as in Felden et al. 1997 RNA).

Line 112. Typo, stacked

Line 113. In extended data fig 7c, A1493 doesn't appear to be "flipped" out in the same sense as in extended data figure 7f for the accommodated state, although a similar description is used. In 7c A1493 appears to be in some sort of intermediate shifted state but not flipped out as in the field's traditional understanding of that description. Either the language should be edited to reflect this nuanced difference or the figure should be aligned to visually clarify the similarity.

A comparison between decoding bases of the PRE tmRNA-SmpB versus PRE canonical tRNA would benefit the reader as well as a comparison to the previous crystal structure of PRE tmRNA-SmpB. It is remarkable that a protein can satisfy the demands usually required of an RNA codon-anticodon interaction. Is it possible that tmRNA binding to the small subunit has any effect on 30S subunit closure? In that context, can the structure provide any insight into how GTP hydrolysis is triggered in EF-Tu even without a cognate codon-anticodon? If so this would constitute a major advancement in our understanding of the initial stage of tmRNA-SmpB loading.

Line 121. As the residues of the previously unseen interaction between the SmpB tail and H5 are described (U120, Lys156, Asn157), the density for this interaction should be shown in extended data

figure 8 to convince the reader.

Line 125-126. Based on this structure, the contact between the tail of SmpB and H5 in the PRE state could be instrumental for initial positioning of the tail of SmpB during pre-accommodation, and possibly for the transition from the PRE to ACC state. This is an interesting finding that was not well resolved in previous structures. However, there is no evidence provided that this interaction would have any influence on the later positioning of the MLD for re-registration as claimed. Either additional data is needed to support the claim, or this statement must be edited to accurately reflect only the implications of the H5-SmpB tail interaction as can be suggested by this new structure.

Lines 126 – 130. It is unclear what point is trying to be made here. The authors should clarify why the flexibility of L27 is relevant with respect to the referred structure of a non-stop ribosome bound by a trans-translation inhibitor. In extended data figure 9, as a separate panel, the authors could compare protein L27 in the PRE and ACC states for both tmRNA and canonical tRNA bound ribosomes and reference that here to illustrate the differences.

ACC

Line 152, suggestion. The MLD is mentioned to be better resolved outside the ribosome, which again could be interesting, as this has never been seen. This could be shown in a figure and compared to the MLD of PRE.

Line 163. Again the description of the decoding bases is unclear. Whereas, in extended data 7c A1492 was described as stacking within 16S rRNA, the same is said about A1492 in 7f yet the figure does not appear to show this similarity. If they are indeed the same the figures should be edited to reflect this, otherwise the text must be revised.

Line 168-170. A direct overlay showing the changes or similarities between the position of H5 and the tail of SmpB in the PRE vs ACC state is needed to illustrate this point. By comparing extended data figure 8a and 8b there seem to be numerous differences and similarities but in its current representation it is difficult for the reader to confirm the authors descriptions.

TRANS

Line 202. A longstanding structural question is how the nucleotides upstream of the resume codon of the MLD interact with SmpB to set the reading frame. This newly visualized interaction, which was not sufficiently resolved in Rae et al. 2019, is an important advancement in our structural understanding of trans-translation. Extended data fig 10 should show the density for this crucial interaction to convince the reader, and this figure more likely deserves a place in the main manuscript. Additionally, this interaction deserves a more detailed rationalization in the text. For instance, please elaborate on how the mutations in SmpB can overcome the A86C mutation in tmRNA. Early work (Lee et al. 2001 RNA) describing the importance of the nucleotides upstream of the resume codon of tmRNA in setting the reading frame should also be rationalized in the context of this high-resolution structure.

Line 213. In the text His136 is said to stack with C1400 but in extended data figure 8c/d the nucleotide is labeled G1400. This discrepancy should be corrected. It is interesting that the conserved His residues important for interaction between SmpB and the decoding center show a dual purpose and make interactions again with the 16S rRNA, but this time to position tmRNA-SmpB in the P site. These interactions were not previously described in detail and this elegant mechanism should be highlighted as such to make clear to the reader.

TRANS*

Line 261. Use of the term refolding is unclear. During this speculation it should be clarified whether

the authors think tmRNA must unfold to transit the subunit interface of the ribosome or whether conformational changes between the domains are sufficient to permit this movement.

Suggestion. In general the main value of this new structure seems to come from its novel head movement. The authors may consider visualizing, possibly with a structural heat map, the range of known head movements for tmRNA-bound ribosomes and show where this new structure lies on that spectrum.

CONCLUSION

Line 267. Like in the abstract, in its current form this manuscript has not showcased the differences of trans-translation from canonical translation because the authors have not given the reader a comparison to canonical structures. If this is to be a major conclusion of this work, then that analysis must be done and represented, as suggested above.

Lines 268-279. Structures of trans-translation intermediates may indeed be important for one day understanding the mechanism of action of trans-translation inhibitors. However, as this study did not analyze or characterize antibiotics in the context of trans-translation it does not seem appropriate that the majority of the conclusion is dedicated to this pondering. The conclusion should be written to recapitulate the important new findings of this work.

METHODS

Line 372. Typo. complexe

Line 393. The method for focused classification with signal subtraction must be clarified so as to convince the reader that homogeneous classes were indeed obtained. For instance, was focused classification run on every individual factor to prevent a reconstruction of a mixture of ribosomes? Please clarify how this was done and explain how this resulted in four distinct conformational states as described.

Reviewer #2 (Remarks to the Author):

Even under optimal growth conditions, but particularly under various stress conditions, mRNAs become damaged which can lead to the loss of the stop codon. Translation of these non-stop mRNAs by ribosomes leads to stalling at the 3' end of the mRNA and eventually sequesters ribosomes from the active translating pool. Rescue of these stalled ribosomes is essential for survival and bacteria have developed many different rescue systems to deal with these stalled ribosomes, the most conserved and important of which is trans-translation mediated by the tmRNA and SmpB. Understanding structurally how tmRNA moves through the ribosome has been a long sought-after goal with the most recent tour-de-force being a Science paper in 2019 from the Ramakrishnan group presenting multiple structures of different states of elongation. Here the Gillet group take a similar approach to catch tmRNA in action and present four cryo-EM structures, two of which appear more or less identical to those reported by Ramakrishnan and coworkers, namely, the accommodated state and the translocated state. The novel states include the EF-Tu bound state, which is in effect similar to the previous X-ray structure of the SmpB-tmRNA fragment, and a translocated state that the authors refer to as an intermediate because bridge B1a is further shifted away from the 50S subunit but in the end is very similar to the translocated state. Overall, this is a lot of work, however, lacks novelty because of the previous work of the Ramakrishnan group published one and half years ago (Feb 2019). Moreover, I find many of the statements in this paper with respect to the resolution to be very misleading and one wonders whether the authors really have the local resolution to support their many sidechain interaction claims.

Major comments

1. There are endless claims of high resolution and reports that their structures are at 3.0 to 3.4Å (abstract) or “four new high resolution cryo-EM structures” line 67, or “the very first high resolution 3.2Å” line 92 etc etc...While these statements are presumably correct with respect to the average resolution, one immediately sees in the Figure S2 that this resolution is basically for the large subunit and that the regions of interest and relevance for this study, namely, the small subunit, tmRNA etc appear to be more in the resolution range of 4Å or worse (up to 16Å!). I write appear because the only indication of local resolution is from small overviews (no transverse sections) of the structures. No effort has been made to show the local resolution of the individual factors to strengthen their interpretation. This needs to be done at a minimum for each factor separately and from each state.
2. The overall quality of many of the images is poor. For the main figures, it appears that the densities were isolated using color zone and not cleaned up to distinguish which density is really belonging to the factor and which comes from the ribosome. This does not inspire confidence in the interpretation of the interactions. Many of the maps have additional density/noise flying around, perhaps the authors should consider to filter the maps based on the local resolution, particularly since the local resolution appears to be so different for different parts of the structure, not just the tmRNA molecule but also the subunits. In fact in this regard, I am surprised that no multi body refinement was performed on the small subunit with and without the tmRNA as well as the tmRNA, tRNAs, SmpB alone. This could improve a lot the local resolution for these regions of the different states.
3. Figure S3 is simply not acceptable. Nothing can be validated with respect to the Kir or GDP density with this poor presentation. Many of the other sup figures (S4-S8) don't show density, just endless interactions using the model...one wonders whether the authors have the local resolution to describe these interactions unambiguously. The one exception is 7a and b where overviews are shown of the density, which does not look particularly resolved and then little boxes are put on regions that one cannot see properly because they are too small. The density images need to be provided for the corresponding panels b-f at the same size and orientation to be convincing. Figure S9 is not convincing to illustrate that the nascent chain is attached to the A-site and not the P-site.
4. The conclusions section I find totally speculative and should be rewritten. The idea that having “high” resolution structures of tmRNA will help drug design is a fantasy. How due the authors envisage being able to design compounds to bind at selective sites on the ribosomes to prevent tmRNA binding or changing conformations?

Minor comments

1. The use of PRE for the EF-Tu delivery state is somehow confusing since actually the accommodated state is a classical PRE state. Maybe the authors want to reconsider their nomenclature.

Reviewer #3 (Remarks to the Author):

This manuscript describes four structures of tmRNA-SmpB bound to the bacterial ribosome. To my knowledge, three of these structures of tmRNA-SmpB have been solved bound to the ribosome at varying resolutions. I would agree with the authors that one of the most important states- that is, the pre-accommodation state (PRE) of tmRNA-SmpB-EF-Tu - has not been solved to high resolution. This structure is clearly the most important result of this manuscript. Despite the biological importance of this structure, overall the manuscript lacks sufficient information required to understand its significance and to make these results broadly accessible to a diverse scientific audience as outlined in more detail below. Also most, if not all, the figures are not presented to a high standard or high resolution, further preventing the ability to properly evaluate the work. These two items are major weaknesses.

In the future, please include line numbers to allow reviewers the ability to easily refer to different

sections of the manuscript.

MAJOR ISSUES

1. Manuscript clarity. There are a number of details that are not explained in sufficient detail to understand the work presented. Outlined below are some examples but overall, the entire manuscript is difficult to read if one is outside the field.

-The first paragraph explains broadly what is known about trans-translation. Paragraph 2 goes into more detail regarding which structures are known of the ribosome bound to tmRNA-SmpB. When the authors describe the PRE, ACC and TRANS steps, to a non-expert, these states are unclear. Before describing these structures, a better description of what these steps are should be provided.

-What is the significance of the 3.2A structure not including the PK ring? (Intro)

-The authors previously published a PRE state in 2006 with Frank. How does this fit into the structures presented here?

-The authors should explain in the Intro why the PRE structure is important and why tmRNA pre-accommodation is likely to be very different from normal tRNA accommodation (if that is their point). Related to this, why is this structure important to understand the mechanism of the transition of the TRANS state?

-To my knowledge, a recent paper (ref 15) did show the ACC and TRANS states, correct? This needs to be described (end of third Intro paragraph).

-When describing the structures the authors have trapped at the end of the Intro, they state they have trapped a TRANS* state but this is the first time this terminology has been used. It is strongly suggested that the authors clearly describe the different states in the second Intro paragraph to help place the importance of these structures in the context of different functional states.

2. PRE state structure. I agree with the authors that the lack of a PRE state to high resolution prevents a complete description of the molecular details of this mechanism. The text describes some new interactions seen that may be important but Figure 2 is messy and unclear so none of these details can be seen. Since this is the most important new information in this manuscript, without clear figures, I cannot evaluate the work. More details are below:

-EFTu switch 1 is flexible and the authors are unable to model it in the PRE state. Both GDP and kirromycin are bound- how does this compare to other EF-Tu states already solved? (likely this analysis should be discussed in the Discussion section). A comparison should be provided to allow the reader a comparison to how tRNAs are delivered to the ribosome.

-In Figures 1a and 2, the structure that was captured was EF-Tu bound in a GDP state so after GTP hydrolysis but before released. It is mentioned that the "The TLD is partly positioned in the A-site in a pseudo-A/T state, except that it is in an open L-shape^{17,18}". It is not clear what a pseudo-A/T state or an open L-shape are and why these are important. This needs a better introduction.

-In this new PRE state structure, the authors describe how conserved SmpB residue Trp122 is inserted into a hydrophobic pocket. Does A8, A15, U17, A334, and C335 refer to 16S rRNA and if so, which region/ helix? Figure 2c does not show these interactions in any detail.

-The interactions between SmpB and EF-Tu shown in Figure 2b-d are not of sufficient quality to ascertain the major points that the authors attempt to make in the text.

-It is described that PK2 interacts with uS3 and that H7 interacts with uS3/4/5- any functional significance to this considering that these are helicase proteins? (also- are all these proteins universal ("u")?)

-None of the interactions between SmpB and the decoding can be observed in Figure 2d. Why not have a close up of an updated Extended Figure 7 in the main text?

-The authors state that "In a previously unseen interaction, the SmpB C-terminal tail engages with the tmRNA H5 helix (Fig. 2e), with U120 from the first tmRNA stop codon interacting with SmpB residues Lys156 and Asn157 (Extended Data Fig. 8a)."- Figure 2e does not allow one to see these interactions while Extended Data Fig. 8a shows too many details that obscures what the main interactions the reader needs to view. If this is the first time this interaction has been observed, it should be shown in detail in the main text. However, Extended Data Fig. 8 needs extensive updating for this point to be

taken. It is messy and unclear.

-The authors makes the point that the SmpB C-terminal tail interactions with H5 may be species specific because some bacteria have a truncated C terminal tail and H5. How then can they make the argument that these elements are important for positioning the MLD? Have any biochemical studies been performed that could provide these insights? As written, it is unclear and doesn't really make biological sense.

-Immediately after the statement about the C-terminal SmpB tail and H5 interaction, the following statement is made: "It was recently suggested that the region encompassing the first seven residues of bL27 is rotated in non-stop ribosomes, and that this plays an important role at the start of trans-translation¹⁹". While I agree that this new structure could be described in this manuscript, at this point in the paragraph is completely out of place and there is no connection to the surrounding text. Also from my quick read of the bioRxiv manuscript, the main conclusion is not that bL27 is rotated in non-stop ribosomes otherwise wouldn't all the other non-stop ribosomes contain a bL27 that is rotated? To my knowledge, these other non-stop structures do not contain a rotated bL27. The reason for the bL27 rotation appears to be both a non-stop ribosome and the presence of an inhibitor that specifically recognizes trans-translation. Regardless, this analysis and comparison needs expanding which would be more appropriate for the Discussion section.

-The authors state that "As is frequently the case for structures lacking A-site tRNA, we could not unambiguously reconstruct the first five residues, which confirms that this region is highly flexible." Which structures? Ones containing normal tRNA or a non-stop ribosome? This statement is ambiguous and needs a further explanation.

3. ACC state. The authors also capture a accommodated state where EFTu has been released and the TLD moves into the PTC. Although the data presented here are emphasized by the authors to be higher resolution than the Ramakrishnan paper published in Science in 2019 (3.1 vs 3.9A), it is hard to see how this increase in resolution has improved our understanding mainly because the figures are not clear.

-The authors state that "After analyzing the PTC density, we concluded that the transfer of the P-site tRNA phenylalanine to the incoming tmRNA alanine had already occurred." Which figure shows this? These details cannot be seen. Later on in the text, Figure 3c is called out but none of these details can be seen in Figure 3.

-The authors state that "The tmRNA T-arm interacts with uL16 and the 23S rRNA helices H38, and H89 (Fig. 3c)." I can't see these details. Are they important?

-The authors state that "A1493 is now completely flipped outside and interacting with the SmpB His22 residue (Extended Data Fig. 7f)". This stacking is hard to see in this figure and maybe more importantly, the question is- is this a new interaction?

4. Translocation step and intermediate translation state structure.

- the authors needs to describe what a TRANS and TRANS* conformations are.

-The authors state "The map's high-resolution allowed us to build a robust and detailed atomic model, including crucial portions of the MLD (Fig. 4a)." This figure does not show this.

-The authors state that "In the PRE state, in accordance with the open conformation of bridge B1a, the tip of the 50S H38 helix is unstructured, while the B1b/c bridges are closed (Fig. 1e)." Figure 1e does not show this. What am I supposed to be looking at?

- The authors state that "... the tmRNA H2 helix is slightly bent, and the entire PK ring has shifted (RMSD = 6.5 Å²)." RMSD of what? And how aligned?

-The authors state that "Compared to the post-translocational intermediate state observed by Ramrath et al.¹⁴, here the swivel of the 30S head is somewhat less noticeable (17.8° vs 20.7°), but EF-G has already been released, the TLD-SmpB module is clearly in a P/P state, and there are no tRNAs in the E-site." When EF-G is bound in intermediate states (the authors need to cite the Noller papers here), the 30S head both swivels and tilts and both movements should be described. Further there are some structures that also have the 30S head swivel/tilting in the absence of EFG (Noller papers and Dunham papers of tRNAs that frameshift). How does the structures presented here compare?

-Although the authors state both they and the Ramakrishnan group could not see a tmRNA in the E site, I think this might be incorrect. Didn't the Ramakrishnan group also visualize a state past the E site? In this case, isn't the authors' suggestion then that the tmRNA transits quickly through the E site without merit? (or without any biochemical data that supports this statement?)

5. Figures. Most if not all the figures needs a substantial overhaul in order to see the points that authors assert in the text. In addition to new figures, better organization rather than including all four structures together in a single figure (Figure 1).

-Overall the details described in the text and shown in Figure 1e-h are not observable given the current figures. For example, it is mentioned that Helix 2 points out of the ribosome but this cannot be seen in any detail.

-Extended Data Figure 3 needs to be updated as it is not clear what the point of this figure is. If the point is to show the map quality, this depiction does not provide any confidence with the high resolution map. If the point is to show the details of GDP, switch 1 and KIR are located on EFTu, this does not come across. These two important points needs to be separately evaluated.

-Extended Data Figure 4- again what is the purpose of this figure? These molecular interactions are difficult to discern.

-The same is true for Extended Data Figure 5, 6, 8

-Although the authors include the map resolutions of the four structures in Extended Data Figure 2, the authors should also include SmpB and tmRNA with their corresponding resolutions as this part of the manuscript is the most important.

MINOR-SmpB should be labeled in Figures 1a-d.

6. The Conclusion/Discussion section is too brief. Although the authors state that these structures "showcase its differences from canonical translation and the fact that it does not interfere with that process", they never describe in detail what these differences are. This needs to be done in the Discussion section. Comparison with EFTu structures and with structures showing altered head domains should be performed.

Some of the phrasing is odd and should be updated (MINOR):

Introduction

- "...thanks to its third base pair," "thanks to the stop codon", "...understanding of re-registration on the MLD..."

- "The problematic mRNA is taken up by RNase R6,...." - a better way to state this is- The non-stop mRNA is degraded by RNase R

-To my knowledge, uL27 is incorrect and should be relabeled bL27 (it is specific to bacteria, correct?).

-Second sentence of the second paragraph of Introduction - the authors should cite a review that describes how trans-translation is an attractive target for new antibiotics. A quick search brought up one review by the Keiler lab.

- "As for SmpB, it lies on the decoding site (DS),..."

- "Thanks to another population of 11,059 particles obtained from the EF-G experiment,...."

- "For instance, the binding of tmRNA to a stalled ribosome is based on SmpB recognition of the problematic situation,..."

REVIEWER COMMENTS

Reviewer #1 (Remarks to the Author):

Comments

General note. By convention A-site with hyphen is used when saying, for instance, “the A-site tRNA”, whereas A site without hyphen should be used when referring directly to the location in the ribosome, as in, “the tRNA bound in the A site”.

Changed everywhere.

General note. The author claims that previously unseen interactions between tRNA-SmpB and the ribosome explain why the process does not interfere with canonical translation however the figures completely lack illustration of this idea as no direct comparisons between trans-translation and translation are presented. Analysis as suggested below should be completed to illustrate this point.

We totally agree with this remark. All of the figures were modified, and comparisons with canonical translation illustrated there and added in the appropriate places in the text (see below).

INTRO

Line 49. In extended data figure 1 the MLD is the mRNA-like domain, not messenger-like domain. Please correct.

Fixed. (Note that we also improved the figure by annotating the resume codon as well as the stop codon, and by highlighting the residues involved in the formation of the putative central hairpin in the MLD).

Line 65-66. Although a high-resolution structure of full-length tmRNA in the PRE state is required for a complete molecular description of tmRNA-SmpB loading onto the ribosome and its transition into the ACC state, how it is required for an understanding of the movement into the TRANS state is unclear. The claim should be edited to accurately reflect this.

OK, the following sentence was added (new lines 76-79):

“This step would be particularly interesting to elucidate since, unlike in canonical translation, ribosome recognition only occurs in the absence of codon-anticodon base pairing. The major role played by SmpB and tmRNA interactions during this first step and how this affects the subsequent transition to the ACC and TRANS states has remained unclear.”

RESULTS

PRE

Line 98. Extended data Fig 3 is unclear. To support the claim that EF-Tu is in the GDP bound state with kirromycin bound, please show the density of GDP and kirromycin alone without EF-Tu.

The figure was improved and the densities corresponding to GDP and kirromycin are now shown. See the new Supplementary Figure 6.

Line 107-108. The structure answers the open question of how during pre-accommodation, PK2 initially interacts with the small subunit as seen in all subsequent steps. This early interaction has never been seen at this resolution and provides conclusive data that not only the tail of SmpB, but also tmRNA is important for the initial detection of non-stop ribosomes, as PK2 and H5 binding in this position would be incompatible with long mRNA. The authors should highlight this finding and compare the position of H5 directly to that of the accommodated state as well as to mRNA during canonical translation.

Right, thanks for this observation. To highlight the interactions of PK2 and H5 with the 30S small subunit, we improved the supplementary figure 6 which became the new figure 2. We also added a new Figure 9 that shows the mRNA path with an overlay of tmRNA-SmpB and a long canonical mRNA in the PRE-ACC, ACC and, TRANS states. In addition, the following text was added in the discussion (new lines 399-404):

“One significant finding is that the SmpB C-terminal tail is not the only sensor used by *trans*-translation in detecting stalled ribosomes. Indeed, in both the pre-accommodation and accommodation structures, the tmRNA H5 stem-loop plays a crucial role by interacting with the end of the SmpB C-terminal tail, closing off access to the mRNA channel. The positions of both the C-terminal tail and the stem-loop are incompatible with a ribosome undergoing canonical translation, because a steric clash would occur with a non-truncated mRNA (Fig. 9).”

Line 100. The authors claim that tmRNA resembles aminoacyl-tRNA in the PRE state. Please support this claim with data visualization, possibly by showing the reader an overlay of tmRNA-SmpB and tRNA in the PRE states.

Good point. A structural comparison between the tmRNA-SmpB-EF-Tu-GDP quaternary complex and a canonical tRNA-EF-Tu-GDP ternary complex is now available in the new Supplementary Fig. 7.

Line 102. Extended data figure 4 has a labeling inconsistency. Middle panel Hing1 should be labeled Switch 1 as in extended data figure 3.

Correct. However that figure has been discarded, and the interaction between EF-TU and TLD is now available in the new Supplementary Fig. 6.

Line 110. The MLD has never been visualized outside of the ribosome and a putative hairpin may be present. If indeed the MLD is partly folded and density can be seen, this should be shown in a figure and described in more detail. This could confirm previous secondary structure analysis (as in Felden et al. 1997 RNA).

Thanks, this is a very insightful comment. We could indeed observe a single dense region that is compatible with the presence of this previously described hairpin in the ACC state (see below and Supplementary Fig. 14). However, the local resolution in the PRE-ACC state is not as good as in ACC (see new Supplementary Fig. 3), and we could not make an unambiguous conclusion about the presence of a hairpin.

Line 112. Typo, stacked
Fixed.

Line 113. In extended data fig 7c, A1493 doesn't appear to be "flipped" out in the same sense as in extended data figure 7f for the accommodated state, although a similar description is used. In 7c A1493 appears to be in some sort of intermediate shifted state but not flipped out as in the field's traditional understanding of that description. Either the language should be edited to reflect this nuanced difference or the figure should be aligned to visually clarify the similarity.

A comparison between decoding bases of the PRE tmRNA-SmpB versus PRE canonical tRNA would benefit the reader as well as a comparison to the previous crystal structure of PRE tmRNA-SmpB. It is remarkable that a protein can satisfy the demands usually required of an RNA codon-anticodon interaction.

Indeed, the Extended Data Fig. 7 did not present the best angle for seeing the conformational changes. We therefore revised the views for the PRE-ACC, ACC, and TRANS states in the new main Fig. 3, where the conformations of A1492 and A1493 are now clearer. We also added a new Supplementary Fig. 10 to compare the decoding center of our structures to those of translating *E. coli* ribosomes and to other *trans*-translating ribosomes structures. We updated the text to reflect this (new lines 154-161):

"In the DC, A1492 is stacked in the 16S rRNA H44 helix, while A1493 is in an intermediate state, shifted toward the major groove because of its interaction with SmpB's His22 (Fig. 3a). This conformation is different from that of canonical decoding^{26,27}, but it is similar to the published high-resolution structure of an empty *E. coli* ribosome²⁸ (Supplementary Fig. 10a and 10d). Our structure also differs from the crystal structure of a tmRNA fragment, SmpB and EF-Tu of *Thermus Thermophilus* bound to a ribosome, where the A1493 is only partly shifted¹⁵ (Supplementary Fig. 10a and 10g)."

Is it possible that tmRNA binding to the small subunit has any effect on 30S subunit closure? In that context, can the structure provide any insight into how GTP hydrolysis is triggered in EF-Tu even without a cognate codon-anticodon? If so this would constitute a major advancement in our understanding of the initial stage of tmRNA-SmpB loading.

Yes indeed, 30S closure is observed upon tmRNA-SmpB binding, as previously described by Neubauer *et al.* (2012), and we added a new Supplementary Fig. 12 to illustrate this point. However, since the structure is already in the post-GTP hydrolysis state, it is difficult to speculate on how exactly the tmRNA-SmpB complex might trigger GTP hydrolysis.

Line 121. As the residues of the previously unseen interaction between the SmpB tail and H5 are described (U120, Lys156, Asn157), the density for this interaction should be shown in extended data figure 8 to convince the reader.

This key interaction between the SmpB C-terminal tail and tmRNA's H5 stem-loop is highlighted in the new main Fig. 5, which shows the cryo-electron density map.

Line 125-126. Based on this structure, the contact between the tail of SmpB and H5 in the PRE state could be instrumental for initial positioning of the tail of SmpB during pre-accommodation, and possibly for the transition from the PRE to ACC state. This is an interesting finding that was not well resolved in previous structures. However, there is no evidence provided that this interaction would have any influence on the later positioning of the MLD for re-registration as claimed. Either additional data is needed to support the claim, or this statement must be edited to accurately reflect only the implications of the H5-SmpB tail interaction as can be suggested by this new structure.

Indeed, this is highly speculative, so lines 123-130 were discarded and replaced by a shorter and more reserved sentence (new lines 194-195):

"This specific interaction is probably instrumental in properly positioning the MLD and SmpB to ensure the correct re-registration later on."

We decided to further comment this point in the discussion as our results are in fact supported by the previously published observations that the modification of the extremity of the SmpB C-terminal tail (in particular Met155 and Ile154) severely reduces tmRNA-SmpB tagging activity (new lines 399-415):

"One significant finding is that the SmpB C-terminal tail is not the only sensor used by *trans*-translation in detecting stalled ribosomes. Indeed [...] ensuring the correct positioning of the resume codon."

Lines 126 - 130. It is unclear what point is trying to be made here. The authors should clarify why the flexibility of L27 is relevant with respect to the referred structure of a non-stop ribosome bound by a *trans*-translation inhibitor. In extended data figure 9, as a separate panel, the authors could compare protein L27 in the PRE and ACC states for both tmRNA and canonical tRNA bound ribosomes and reference that here to illustrate the differences.

Right, we wanted to refer to bL27 as it was recently suggested that the conformation of the bL27 N-terminal arm is altered in stalled ribosomes bound to acylaminooxadiazoles (a *trans*-translation inhibitor). A new Supplementary Fig. 15 was added, with the following comment (new lines 492-500):

"Finally, *trans*-translation is an appealing antibiotic target since it is essential for fitness and vital to many pathogens, yet absent in eukaryotes. [...] This may imply that if the binding of such molecules is specific to stalled ribosomes, it would impede the accommodation of tmRNA-SmpB."

ACC

Line 152, suggestion. The MLD is mentioned to be better resolved outside the ribosome, which again could be interesting, as this has never been seen. This could be shown in a figure and compared to the MLD of PRE.

We supported the statement that the MLD is better resolved by providing a new Supplementary Fig. 3 which illustrates the local resolutions of each of the individual components of the *trans*-translation complex. As the resolution of the MLD in the PRE-ACC state is quite poor, it is hard to discuss it. However as already stated in the "PRE" section of the comments, we observed a single dense region compatible with the presence of a previously described hairpin in the ACC state. We illustrate this density in the new Supplementary Fig. 14 and discuss it in the paper (new lines 218-221):

“The MLD is mostly stretched, and presents a single dense region at its centre which is compatible with the presence of a previously described hairpin between nucleotides U88 and A100³⁰, a pairing which may protect the resume codon until it is used (Supplementary Fig. 14).”

Line 163. Again the description of the decoding bases is unclear. Whereas, in extended data 7c A1492 was described as stacking within 16S rRNA, the same is said about A1492 in 7f yet the figure does not appear to show this similarity. If they are indeed the same the figures should be edited to reflect this, otherwise the text must be revised.

As mentioned above, we added a new main Fig. 3 and a new Supplementary Fig. 10 to clarify what happens in the decoding center in the PRE-ACC, ACC, and TRANS states. This was discussed in the text (new lines 234-242):

“In the decoding centre, interactions with the conserved 16S rRNA nucleotides G530 and C1054 are also maintained and even reinforced, with Lys138 binding with G530, and H136 interacting with both C1397 and G530 (Fig. 3b and Supplementary Fig. 11). [...] as well as what is observed in the previous structure from Rae *et al.*¹⁷, although the His22 position differs (Supplementary Fig. 10b and 10h).”

Line 168-170. A direct overlay showing the changes or similarities between the position of H5 and the tail of SmpB in the PRE vs ACC state is needed to illustrate this point. By comparing extended data figure 8a and 8b there seem to be numerous differences and similarities but in its current representation it is difficult for the reader to confirm the authors descriptions.

Supplementary Fig. 13 was added to illustrate the movement of tmRNA's PK ring and H5 helix from the PRE-ACC through to the TRANS state. The different structural conformations are also visible in the main Fig. 2 (panels a, d, and g), which presents an overview of the interactions between tmRNA and the 30S small ribosomal subunit. The text was updated (new lines 210-213):

“When compared to the PRE-ACC state, the PK ring is in the same position around the beak, but is now larger (96 x 128 Å). Thanks to the movement of the H5 stem-loop towards the ribosome (Fig. 2d and Supplementary Fig. 13), PK2 interacts even more tightly with the KH2 RNA-binding domain of uS3 (Fig. 2e).”

TRANS

Line 202. A longstanding structural question is how the nucleotides upstream of the resume codon of the MLD interact with SmpB to set the reading frame. This newly visualized interaction, which was not sufficiently resolved in Rae et al. 2019, is an important advancement in our structural understanding of trans-translation. Extended data fig 10 should show the density for this crucial interaction to convince the reader, and this figure more likely deserves a place in the main manuscript. Additionally, this interaction deserves a more detailed rationalization in the text. For instance, please elaborate on how the mutations in SmpB can overcome the A86C mutation in tmRNA. Early work (Lee et al. 2001 RNA) describing the importance of the nucleotides upstream of the resume codon of tmRNA in setting the reading frame should also be rationalized in the context of this high-resolution structure.

We improved this figure by adding the densities that correspond to these crucial nucleotides and it became the new Fig. 6, which we discuss in the main text (new lines 272-279):

“This precise positioning is made possible by direct interactions between the four nucleotides just upstream from the MLD resume codon and SmpB (Fig. 6a). Indeed, SmpB's Tyr55 and Tyr24 residues are instrumental in correctly positioning the tmRNA resume codon in the A site. Tyr55 stacks with tmRNA's A86 and serves as the foundation for the stacking of the four bases upstream from the resume codon. As for

Tyr24, it lies between tmRNA's A86 and G87, and by forming weak hydrogen bonds with both it serves as a secondary checkpoint, ensuring a finer control of the sequence and limiting the risk of frameshifting."

Moreover, we also explain how SmpB mutations can overcome the A86C mutation in tmRNA, and we discuss the importance of the nucleotides just upstream of the resume codon. This is illustrated by the new Fig. 6, with the effects of different mutations on this region shown in panels b, c, and d. Our discussion was modified as follows (new lines 429-452):

"During translocation, SmpB also plays a major role in ensuring that the resume codon is correctly positioned within the decoding centre. Our TRANS structure unambiguously shows how SmpB binds to the four nucleotides just upstream of the MLD resume codon (Fig. 6). [...] Indeed, by simultaneously altering the sequence checkpoint (mutation Y24C), the shape of the binding pocket (mutation V129A), and the positioning of the tmRNA backbone (mutation E107V), this triple mutation facilitates the shifting of the MLD at the surface of SmpB, allowing for both in-frame and +1 frame re-registration³⁸."

Line 213. In the text His136 is said to stack with C1400 but in extended data figure 8c/d the nucleotide is labeled G1400. This discrepancy should be corrected. It is interesting that the conserved His residues important for interaction between SmpB and the decoding center show a dual purpose and make interactions again with the 16S rRNA, but this time to position tmRNA-SmpB in the P site. These interactions were not previously described in detail and this elegant mechanism should be highlighted as such to make clear to the reader.

We redid Extended Data Fig. 8, and it is now the main Fig. 4. The densities have been added, and we corrected the labeling of the C1400 residue (Fig. 4b). In the new Figure 3, we describe the important roles of His22 and His136 in the DS in the PRE-ACC and ACC states, and we added a new main Fig. 7 to highlight their dual purpose as they are also instrumental in positioning SmpB during the TRANS state. This mechanism is now discussed in the text (lines 302-309):

"It is noteworthy that the two conserved histidines His22 and His136, which were previously involved in the decoding centre during the PRE-ACC and ACC states, have in fact a dual purpose, as they now help position the tmRNA-SmpB complex in the P site. Indeed, His22 now stacks with 16S rRNA's nucleotide A790, while SmpB's His136 is instrumental in positioning the tail through new stacking interactions with 16S rRNA C1400 (Fig. 4c). In doing so, SmpB mimics the way in which a P-site t-RNA anticodon loop interacts with both mRNA and 16S rRNA (Fig. 7)."

TRANS*

Line 261. Use of the term refolding is unclear. During this speculation, it should be clarified whether the authors think tmRNA must unfold to transit the subunit interface of the ribosome or whether conformational changes between the domains are sufficient to permit this movement.

Good point, this is obviously inaccurate. The term "refold" was discarded, and the sentence shortened as follows (lines 388-389):

"The complex would then reach the post-E conformation described by Rae *et al.*¹⁷."

Suggestion. In general the main value of this new structure seems to come from its novel head movement. The authors may consider visualizing, possibly with a structural heat map, the range of known head movements for tmRNA-bound ribosomes and show where this new structure lies on that spectrum.

Good suggestion. To better visualize the head movement and the transit of tmRNA through the B1a, B1b, and B1c bridges, we included a new main Fig. 8, as well as Supplementary Movie 1 and Supplementary Table 2.

CONCLUSION

Line 267. Like in the abstract, in its current form this manuscript has not showcased the differences of trans-translation from canonical translation because the authors have not given the reader a comparison to canonical structures. If this is to be a major conclusion of this work, then that analysis must be done and represented, as suggested above.

The comparison of *trans*-translation with canonical translation has been covered much more fully in this new version of the manuscript. We deeply thank the reviewer for this insightful remark.

Lines 268-279. Structures of trans-translation intermediates may indeed be important for one day understanding the mechanism of action of trans-translation inhibitors. However, as this study did not analyze or characterize antibiotics in the context of trans-translation it does not seem appropriate that the majority of the conclusion is dedicated to this pondering. The conclusion should be written to recapitulate the important new findings of this work.

The concluding paragraphs were greatly expanded accordingly.

METHODS

Line 372. Typo. Complexe

Fixed.

Line 393. The method for focused classification with signal subtraction must be clarified so as to convince the reader that homogeneous classes were indeed obtained. For instance, was focused classification run on every individual factor to prevent a reconstruction of a mixture of ribosomes? Please clarify how this was done and explain how this resulted in four distinct conformational states as described.

This section was thoroughly rewritten, see new lines 609-624:

“We then subtracted the signal of the ribosome from the datasets, using a soft mask (voxel values of 0 inside, 1 outside, extended by 6 pixels with a soft edge of 6 pixels) generated from the previous refinement run. [...] two distinctive conformations of the tmRNA-SmpB complex translocated into the P site (with 14,192 particles in TRANS and 11,059 particles in TRANS*); stalled ribosomes (1 class); and junk (1 class).”

Reviewer #2 (Remarks to the Author):

Even under optimal growth conditions, but particularly under various stress conditions, mRNAs become damaged which can lead to the loss of the stop codon. Translation of these non-stop mRNAs by ribosomes leads to stalling at the 3' end of the mRNA and eventually sequesters ribosomes from the active translating pool. Rescue of these stalled ribosomes is essential for survival and bacteria have developed many different rescue systems to deal with these stalled ribosomes, the most conserved and important of which is trans-translation mediated by the tmRNA and SmpB. Understanding structurally how tmRNA moves through the ribosome has been a long sought-after goal with the most recent tour-de-force being a Science paper in 2019 from the Ramakrishnan group presenting multiple structures of different states of elongation.

Here the Gillet group take a similar approach to catch tmRNA in action and present four cryo-EM structures, two of which appear more or less identical to those reported by Ramakrishnan and coworkers, namely, the accommodated state and the translocated

state. The novel states include the EF-Tu bound state, which is in effect similar to the previous X-ray structure of the SmpB-tmRNA fragment, and a translocated state that the authors refer to as an intermediate because bridge B1a is further shifted away from the 50S subunit but in the end is very similar to the translocated state. Overall, this is a lot of work, however, lacks novelty because of the previous work of the Ramakrishnan group published one and half years ago (Feb 2019). Moreover, I find many of the statements in this paper with respect to the resolution to be very misleading and one wonders whether the authors really have the local resolution to support their many sidechain interaction claims.

Major comments

1. There are endless claims of high resolution and reports that their structures are at 3.0 to 3.4Å (abstract) or “four new high resolution cryo-EM structures” line 67, or “the very first high resolution 3.2Å” line 92 etc etc...While these statements are presumably correct with respect to the average resolution, one immediately sees in the Figure S2 that this resolution is basically for the large subunit and that the regions of interest and relevance for this study, namely, the small subunit, tmRNA etc appear to be more in the resolution range of 4Å or worse (up to 16Å!). I write appear because the only indication of local resolution is from small overviews (no transverse sections) of the structures. No effort has been made to show the local resolution of the individual factors to strengthen their interpretation. This needs to be done at a minimum for each factor separately and from each state.

We understand the concerns of Referee 2. This is why we improved Figure S2 (now Supplementary Fig. 2), and we now show a view sliced halfway through the maps to better visualize the resolutions, especially on the small subunit. Moreover, we added two new supplementary figures (3 and 4). The first of these illustrates the local resolutions of each of the individual components of the *trans*-translation complex, and the second shows the sharpened density maps of the large ribosomal subunit and the small ribosomal subunit body and head regions according to the local resolutions.

2. The overall quality of many of the images is poor. For the main figures, it appears that the densities were isolated using color zone and not cleaned up to distinguish which density is really belonging to the factor and which comes from the ribosome. This does not inspire confidence in the interpretation of the interactions. Many of the maps have additional density/noise flying around, perhaps the authors should consider to filter the maps based on the local resolution, particularly since the local resolution appears to be so different for different parts of the structure, not just the tmRNA molecule but also the subunits. In fact in this regard, I am surprised that no multi body refinement was performed on the small subunit with and without the tmRNA as well as the tmRNA, tRNAs, SmpB alone. This could improve a lot the local resolution for these regions of the different states.

Actually, the densities were not isolated using color zones and, with the exception of Fig. 1, all the maps were cleaned up. Nevertheless, we understand the reviewer's concern and revised every one of the main figures. As requested, we now show the densities around the important residues/nucleotides so the conclusions are more credible. We did not filter the map based on the local resolution. However, we only discuss interactions at the near-atomic level when the resolution is relevant (i.e. below 5 Å), as shown in Supplementary Fig. 3.

As pointed out by Reviewer 2, the local resolution differs for different regions of the ribosome. In fact, to take this into account, we did perform a multibody refinement approach that is covered in the first version of the manuscript. This used three separated overlapping bodies (50S, 30S head, and 30S body), and including the tmRNA-

SmpB complex, and it did indeed improve the local resolution, especially for the PRE-ACC state. To further improve the overall resolution of the PK ring, we also performed a four-body approach, but this was not helpful (results not shown). The multibody refinement cannot be applied to independent components of the tmRNA-SmpB complex because they are too small. All of this was not clearly explained in the Methods section, and we apologize. It has been corrected in the current version, and we have added the separate body maps to our EMDB/PDB deposition.

3. Figure S3 is simply not acceptable. Nothing can be validated with respect to the Kir or GDP density with this poor presentation. Many of the other sup figures (S4-S8) don't show density, just endless interactions using the model...one wonders whether the authors have the local resolution to describe these interactions unambiguously. The one exception is 7a and b where overviews are shown of the density, which does not look particularly resolved and then little boxes are put on regions that one cannot see properly because they are too small. The density images need to be provided for the corresponding panels b-f at the same size and orientation to be convincing. Figure S9 is not convincing to illustrate that the nascent chain is attached to the A-site and not the P-site.

As requested, we added the cryo-electron density maps in most places. We also reworked many of the figures in other ways, including the following changes:

- Figs. S3 and S4 were removed and replaced by Supplementary Fig. 6, and the kirromycin and GDP densities are now shown.
- The information in Fig. S6 was clarified, and it is now main Fig. 2.
- Fig. S7 has a new view angle and was made clearer, and it is now main Fig 3.
- Fig. S8 was simplified and changed to main Fig. 4. Its new angle also showcases SmpB's C-terminal rotation from the ACC to the TRANS state.
- Fig. S9 was also improved and a comparison with the post-catalysis of canonical translation was added. It is now Supplementary Fig. 15.
- Two new supplementary figures, 16 and 17, were added in response to reviewer comments.

4. The conclusions section I find totally speculative and should be rewritten. The idea that having "high" resolution structures of tmRNA will help drug design is a fantasy. How due the authors envisage being able to design compounds to bind at selective sites on the ribosomes to prevent tmRNA binding or changing conformations?

While screening may be sufficient for finding new molecules – and as a matter of fact, our team has developed and patented two screening methods for evaluating *trans*-translation (Macé *et al.*, JMB 2017; Guyomar *et al.*, NAR 2020) – we are convinced that high-resolution structures are essential for understanding and characterizing the actions of *trans*-translation inhibitors as well as being helpful for improving the activity of new chemical scaffolds. For example, as suggested by our results, such molecules could target the interface between tmRNA and SmpB to inhibit the complex, or they could target the SmpB C-terminal tail or the H5 stem-loop to alter stalled ribosome recognition or even to stabilize the B1b/c bridges to trap the tmRNA. However, we agree with the referee's remark, since this is still speculative. The conclusion was therefore toned down (lines 500-504):

"Our structures give us a better understanding of how *trans*-translation occurs, but they also showcase its differences and similarities with canonical translation, and confirm the fact that it does not interfere with that process. This is of primary importance for developing new antibiotics targeting *trans*-translation, as this depends on understanding how tmRNA and SmpB interact and how they bind to stalled ribosomes."

Minor comments

1. The use of PRE for the EF-Tu delivery state is somehow confusing since actually the accommodated state is a classical PRE state. Maybe the authors want to reconsider their nomenclature.

Thank you for the remark. To avoid any confusion, we replaced the "PRE" nomenclature by "PRE-ACC" throughout the manuscript and in the figures.

Reviewer #3 (Remarks to the Author):

This manuscript describes four structures of tmRNA-SmpB bound to the bacterial ribosome. To my knowledge, three of these structures of tmRNA-SmpB have been solved bound to the ribosome at varying resolutions. I would agree with the authors that one of the most important states- that is, the pre-accommodation state (PRE) of tmRNA-SmpB-EF-Tu - has not been solved to high resolution. This structure is clearly the most important result of this manuscript. Despite the biological importance of this structure, overall the manuscript lacks sufficient information required to understand its significance and to make these results broadly accessible to a diverse scientific audience as outlined in more detail below. Also most, if not all, the figures are not presented to a high standard or high resolution, further preventing the ability to properly evaluate the work. These two items are major weaknesses.

In the future, please include line numbers to allow reviewers the ability to easily refer to different sections of the manuscript.

We apologize, line numbers appear in our PDF version, so we will check with the editors.

MAJOR ISSUES

1. Manuscript clarity. There are a number of details that are not explained in sufficient detail to understand the work presented. Outlined below are some examples but overall, the entire manuscript is difficult to read if one is outside the field.

The manuscript has been entirely rewritten. We expect this new version to be much clearer.

-The first paragraph explains broadly what is known about trans-translation. Paragraph 2 goes into more detail regarding which structures are known of the ribosome bound to tmRNA-SmpB. When the authors describe the PRE, ACC and TRANS steps, to a non-expert, these states are unclear. Before describing these structures, a better description of what these steps are should be provided.

Good point. We added a short description of each state (new lines 52-60):

"The early steps of *trans*-translation can be subdivided into three main processes. The first is the pre-accommodation step (PRE-ACC), when the quaternary complex made by alanylated tmRNA, SmpB, EF-Tu, and GTP binds to the A site of stalled ribosomes. In the accommodation step (ACC), EF-Tu hydrolyses GTP and disassociates from the complex, causing the tmRNA aminoacyl end to swing into the peptidyl transferase centre. The stalled peptide is then transferred to the alanine residue on tmRNA, and in the third translocation step (TRANS), EF-G catalyses the shifting of the tmRNA-SmpB complex from the A to the P site. The problematic mRNA is released, and the tmRNA resume codon enters the A site to be decoded."

-What is the significance of the 3.2A structure not including the PK ring? (Intro)

In an earlier work we had to design a truncated tmRNA, where all the pseudoknots and the MLD were deleted, in order to obtain the crystal structure (Neubauer *et al.*, Science 2012). The present work therefore showcases the first high-resolution structure of the entire tmRNA in the PRE-ACC state. We also replaced “entire PK ring” by “PK ring.”

-The authors previously published a PRE state in 2006 with Frank. How does this fit into the structures presented here?

In the previous low-resolution PRE-ACC structure, we could fit two SmpB proteins. However, this was a misinterpretation due to the low resolution (the density attributed to the second SmpB actually corresponds to the helix H63 of the 23S rRNA which interacts with the tmRNA T-arm), and it was definitely contradicted by the work presented here. A sentence was added (new lines 114-115) to make this clearer:

“It also confirms that only one SmpB protein is bound to the TLD during pre-translocation¹⁸.”

-The authors should explain in the Intro why the PRE structure is important and why tmRNA pre-accommodation is likely to be very different from normal tRNA accommodation (if that is their point). Related to this, why is this structure important to understand the mechanism of the transition of the TRANS state?

Thank you, we have added the following sentence (new lines 76-79):

“This step would be particularly interesting to elucidate since, unlike in canonical translation, ribosome recognition only occurs in the absence of codon-anticodon base pairing. The major role played by SmpB and tmRNA interactions during this first step and how this affects the subsequent transition to the ACC and TRANS states has remained unclear.”

-To my knowledge, a recent paper (ref 15) did show the ACC and TRANS states, correct? This needs to be described (end of third Intro paragraph).

Correct. The following sentence was added (new lines 71-73):

“More recently, three high-resolution structures of *trans*-translation intermediates were also published¹⁷ and these provide more details on how the circularized tmRNA-SmpB complex moves through the ribosome.”

-When describing the structures the authors have trapped at the end of the Intro, they state they have trapped a TRANS* state but this is the first time this terminology has been used. It is strongly suggested that the authors clearly describe the different states in the second Intro paragraph to help place the importance of these structures in the context of different functional states.

Yes indeed, we added the following sentence at the end of the introduction (new lines 87-89):

“The first ‘TRANS’ state is just after the translocation of tmRNA-SmpB from the A to the P site (Fig. 1c), while the ‘TRANS*’ state (Fig. 1d) occurs after TRANS but just before the tmRNA-SmpB complex exits the P site.”

2. PRE state structure. I agree with the authors that the lack of a PRE state to high resolution prevents a complete description of the molecular details of this mechanism. The text describes some new interactions seen that may be important but Figure 2 is

messy and unclear so none of these details can be seen. Since this is the most important new information in this manuscript, without clear figures, I cannot evaluate the work. More details are below:

-EF-Tu switch 1 is flexible and the authors are unable to model it in the PRE state. Both GDP and kirromycin are bound- how does this compare to other EF-Tu states already solved? (likely this analysis should be discussed in the Discussion section). A comparison should be provided to allow the reader a comparison to how tRNAs are delivered to the ribosome.

We removed Supplementary Fig. 3 which had close-up views of EF-Tu during the PRE-ACC step. Instead, Supplementary Fig. 6 allows for a better visualization of GDP and kirromycin. We also added Supplementary Fig. 7, which shows an overlay between our tmRNA-SmpB-EF-Tu-GDP quaternary complex and a canonical tRNA-EF-Tu-GDP ternary complex. That figure illustrates how the SmpB-bound tmRNA functionally mimics a canonical tRNA as well as how, similarly to a canonical tRNA, the SmpB-tmRNA complex is delivered to the stalled ribosome. Both the switch 1 disorder and the GDP/kirromycin conformation are similar to what is observed in canonical translation (Loveland *et al.*, Nature, 2020).

-In Figures 1a and 2, the structure that was captured was EF-Tu bound in a GDP state so after GTP hydrolysis but before released. It is mention that the “The TLD is partly positioned in the A-site in a pseudo-A/T state, except that it is in an open L-shape^{17,18}”. It is not clear what a pseudo-A/T state or an open L-shapes are and why these are important. This needs a better introduction.

Good point, we added the following (new lines 117-121):

“The TLD is partly positioned in what is known as the ‘A/T’ state, which allows the simultaneous interactions of tmRNA-SmpB with the decoding centre and EF-Tu with the 50S subunit²¹. As expected^{22,23}, the large open L-shaped TLD forms an angle of $\sim 120^\circ$, allowing the SmpB-bound tmRNA to mimic the functioning of a canonical tRNA.”

-In this new PRE state structure, the authors’ describe how conserved SmpB residue Trp122 is inserted into a hydrophobic pocket. Does A8, A15, U17, A334, and C335 refereeing to 16S rRNA and if so, which region/ helix? Figure 2c does not show these interactions in any detail.

A8, A15, U17, A334, and C335 are all nucleotides belonging to the TLD domain of tmRNA. This interaction has been detailed in the PRE-ACC, ACC, and TRANS states in the new Supplementary Fig. 9a.

-The interactions between SmpB and EF-Tu shown in Figure 2b-d are not of sufficient quality to ascertain the major points that the authors attempt to make in the text.

We removed Fig. 2. Instead, we added Supplementary Fig. 5, which provides a structural comparison between the Ala-tmRNA-SmpB-EF-Tu-GDP quaternary complex observed in the PRE-ACC and the crystal structure of Ala-tmRNA Δ^m -SmpB-EF-Tu-GDP (Neubauer *et al.*, Science 2012).

-It is described that PK2 interacts with uS3 and that H7 interacts with uS3/4/5- any functional significance to this considering that these are helicase proteins? (also- are all these proteins universal (“u”)?)

Very good point, we thank the reviewer for this remark. Yes, all of these proteins are universal ones, and it is clear that they are of importance as helicases. We hypothesize that they may play a role in the unwinding of the H5 stem-loop during the translation of the MLD, as was suggested to occur for hairpin-containing mRNA (Amiri & Noller, RNA

2019). We added the Supplementary Fig. 17 to illustrate this point, and a new paragraph in the Discussion (new lines 416-428):

“The contacts of the SmpB C-terminal tail with the tmRNA step-loop that are observed during accommodation and pre-accommodation are made possible by the tight interactions between H5 and the universal ribosomal proteins uS3, uS4, and uS5 at the entrance of the mRNA channel (Supplementary Fig. 17). [...] This will allow for the unfolding of H5 during the following steps.”

-None of the interactions between SmpB and the decoding can be observed in Figure 2d. Why not have a close up of an updated Extended Figure 7 in the main text?

Done. We added a new main Fig. 3 to replace the previous Fig. 2d. This new figure is taken from a better angle, showcasing the interactions between the conserved residues in the decoding site and SmpB or the tmRNA's MLD in the TRANS state.

-The authors state that “In a previously unseen interaction, the SmpB C-terminal tail engages with the tmRNA H5 helix (Fig. 2e), with U120 from the first tmRNA stop codon interacting with SmpB residues Lys156 and Asn157 (Extended Data Fig. 8a).”- Figure 2e does not allow one to see these interactions while Extended Data Fig. 8a shows too many details that obscures what the main interactions the reader needs to view. If this is the first time this interaction has been observed, it should be shown in detail in the main text. However, Extended Data Fig. 8 needs extensive updating for this point to be taken. It is messy and unclear.

We agree with Referee 3. That figure has been divided into two main ones to show the most important residues involved in the interactions. The new Fig. 4 shows the residues involved in the interactions between SmpB's C-terminal tail and the 30S small ribosomal subunit, while the new Fig. 5 shows the interactions between the C-terminal tail of SmpB and tmRNA's H5 helix.

All of these interactions are mentioned in the main text (new lines 177-195) (in PRE-ACC):

“The rest of the SmpB C-terminal tail is folded into a α -helix occupying the mRNA path downstream from the stalled mRNA (Fig. 1e), and it is stabilized there by its interaction with 16S rRNA. [...] This specific interaction is probably instrumental in properly positioning the MLD and SmpB to ensure the correct re-registration later on.”

And lines (243-250) (in ACC):

“SmpB's C-terminal tail has the same position and folding as are observed during the PRE-ACC state (Fig. 1f), [...] while Ile154 and Ala158 join with uS5 to form a hydrophobic pocket in which U120 resides (Fig. 6b).”

-The authors makes the point that the SmpB C-terminal tail interactions with H5 may be species specific because some bacteria have a truncated C terminal tail and H5. How then can they make the argument that these elements are important for positioning the MLD? Have any biochemical studies been performed that could provide these insights? As written, it is unclear and doesn't really make biological sense.

We thank reviewer 3 for this remark. We removed the statement that some bacteria have a truncated C-terminal tail and a shortened or no H5, since while the H5 helix is not very well-conserved, the part of the stem-loop that we see interacting with the SmpB C-terminal tail is highly conserved. This interaction between H5 and the C-terminal tail of SmpB is clearly observed in both the PRE-ACC and ACC states, and certainly explains the previously published observations that the modification of the extremity of the SmpB C-terminal tail (in particular Met155 and Ile154) severely reduces tmRNA-SmpB tagging activity. Since the C-terminal tail is not directly involved in *trans*-translational tagging, we hypothesize that this interaction between the tmRNA H5 stem-loop and the end of

the SmpB C-terminal serves as an early mechanical sensor, ensuring the correct positioning of the resume codon. We therefore adapted the text and moved it to the Discussion, lines 399 to 415:

"One significant finding is that the SmpB C-terminal tail is not the only sensor used by *trans*-translation in detecting stalled ribosomes. [...] ensuring the correct positioning of the resume codon."

-Immediately after the statement about the C-terminal SmpB tail and H5 interaction, the following statement is made: "It was recently suggested that the region encompassing the first seven residues of bL27 is rotated in non-stop ribosomes, and that this plays an important role at the start of *trans*-translation¹⁹". While I agree that this new structure could be described in this manuscript, at this point in the paragraph is completely out of place and there is no connection to the surrounding text. Also from my quick read of the bioRxiv manuscript, the main conclusion is not that bL27 is rotated in non-stop ribosomes otherwise wouldn't all the other non-stop ribosomes contain a bL27 that is rotated? To my knowledge, these other non-stop structures do not contain a rotated bL27. The reason for the bL27 rotation appears to be both a non-stop ribosome and the presence of an inhibitor that specifically recognizes *trans*-translation. Regardless, this analysis and comparison needs expanding which would be more appropriate for the Discussion section.

Agreed. We moved this paragraph to the end, in the conclusions, and modified it as follows (new lines 492-500):

"Finally, *trans*-translation is an appealing antibiotic target since it is essential for fitness and vital to many pathogens, yet absent in eukaryotes. It was, for example, recently suggested that a rotated conformation of bL27 is preferred when ribosomes are bound by acylaminooxadiazole ribosome-rescue inhibitors⁵⁵. While we could not unambiguously reconstruct the first five residues of bL27 in the PRE-ACC and TRANS states, the N-terminal arm is particularly well resolved in the ACC conformation (Supplementary Fig. 14), with the PTC perfectly mimicking a canonical tRNA. This may imply that if the binding of such molecules is specific to stalled ribosomes, it would impede the accommodation of tmRNA-SmpB."

We also added a new Supplementary Fig. 15. It shows the interactions between the ribosomal protein bL27, the P-site tRNAPhe, and the tmRNA, and compares the PTC in the post-catalysis state of canonical translation with what we observed during *trans*-translation.

-The authors state that "As is frequently the case for structures lacking A-site tRNA, we could not unambiguously reconstruct the first five residues, which confirms that this region is highly flexible." Which structures? Ones containing normal tRNA or a non-stop ribosome? This statement is ambiguous and needs a further explanation.

To simplify the text, we removed that statement and replaced it by the text copied just above.

3. ACC state. The authors also capture a accommodated state where EF-Tu has been released and the TLD moves into the PTC. Although the data presented here are emphasized by the authors to be higher resolution than the Ramakrishnan paper published in Science in 2019 (3.1 vs 3.9A), it is hard to see how this increase in resolution has improved our understanding mainly because the figures are not clear.

We all agree that the original figures were poor. We hope that the new set of (main and supplementary) figures will help.

-The authors state that “After analyzing the PTC density, we concluded that the transfer of the P-site tRNA phenylalanine to the incoming tmRNA alanine had already occurred.” Which figure shows this? These details cannot be seen. Later on in the text, Figure 3c is called out but none of these details can be seen in Figure 3.

To clarify this point, we added Supplementary Fig. 15, which focuses on the peptidyl transfer center. It has a close-up view of the PTC in the post-catalysis state of canonical translation which reinforces the assertion in question.

-The authors state that “The tmRNA T-arm interacts with uL16 and the 23S rRNA helices H38, and H89 (Fig. 3c).” I can’t see these details. Are they important?

Yes, these details are interesting since they show that the TLD T-arm perfectly mimics a canonical tRNA in its interactions with the 50S subunit. To illustrate that, we added a new Supplementary Fig. 16.

-The authors state that “A1493 is now completely flipped outside and interacting with the SmpB His22 residue (Extended Data Fig. 7f)”. This stacking is hard to see in this figure and maybe more importantly, the question is- is this a new interaction?

As stated above, we added a new main Fig. 3 to compare the structural conformations of the 16S rRNA residues G530, A1492, and A1493 in all three structures presented here. We also added the new Supplementary Fig. 10, which compare this conformation with the ACC state in canonical translation and per Rae *et al.* All of these interactions as well as the comparisons with other structures have been added to the main text in the PRE-ACC, ACC, and TRANS sections.

4. Translocation step and intermediate translation state structure.

- the authors needs to describe what a TRANS and TRANS* conformations are.

Done (new lines 349-359):

“During canonical translation, tRNAs move into hybrid A/P and P/E sites, coupled to the rotation of the platform and body domains of the small ribosomal subunit, which occurs as a 8-10° ratchet-like movement⁴¹ (Supplementary Table 2). [...] The presence of the TLD in the P/P state clearly shows that TRANS* is a post-translocational intermediate state.”

We also added a statement at the end of the introduction (new lines 87-89):

“The first ‘TRANS’ state is just after the translocation of tmRNA-SmpB from the A to the P site (Fig. 1c), while the ‘TRANS*’ state (Fig. 1d) occurs after TRANS but just before the tmRNA-SmpB complex exits the P site.”

-The authors state “The map’s high-resolution allowed us to build a robust and detailed atomic model, including crucial portions of the MLD (Fig. 4a).” This figure does not show this.

The original Fig. 4 has been removed. The different insets were clarified in various new figures. In particular, Fig. 3c shows the MLD’s resume codon positioning in the decoding site.

-The authors state that “In the PRE state, in accordance with the open conformation of bridge B1a, the tip of the 50S H38 helix is unstructured, while the B1b/c bridges are closed (Fig. 1e).” Figure 1e does not show this. What am I supposed to be looking at?

To clarify this point, we added main Fig. 8. This provides close-up views of the transit of tmRNA’s H2 helix through the B1a, B1b, and B1c bridges during *trans*-translation.

- The authors state that "... the tmRNA H2 helix is slightly bent, and the entire PK ring has shifted (RMSD = 6.5 Å²).” RMSD of what? And how aligned?

We provided the RMSD between the PK ring (i.e., everything but the TLD and the H2 helix) in the TRANS and TRANS* states after aligning the two structures on the 30S body. However, since this information is not particularly relevant, we decided to omit it, and changed the text to indicate that “and the entire PK ring has rotated along with the head” (lines 370-371).

-The authors state that “Compared to the post-translocational intermediate state observed by Ramrath et al.¹⁴, here the swivel of the 30S head is somewhat less noticeable (17.8° vs 20.7°), but EF-G has already been released, the TLD-SmpB module is clearly in a P/P state, and there are no tRNAs in the E-site.” When EF-G is bound in intermediate states (the authors need to cite the Noller papers here), the 30S head both swivels and tilts and both movements should be described. Further there are some structures that also have the 30S head swivel/tilting in the absence of EFG (Noller papers and Dunham papers of tRNAs that frameshift). How does the structures presented here compare?

We thank the reviewer for this very interesting remark. To answer the question, we re-analyzed our structures using the method of Nguyen and Whitford, and this allowed us to breakdown the swivel into head rotation and tilt. This resulted in a new paragraph in intermediate translocation section (new lines 354-361):

“Here, the swivel of the 30S head is rather large (13.9°), but it is still 3-6° less than that observed in the presence of EF-G-GDP and either tRNA^{44,45} or tmRNA¹⁶, or in the presence of frameshift-prone tRNA⁴⁶. [...] This demonstrates that the presence of tmRNA-SmpB in the P site is sufficient to tilt the head.”

We cited Noller’s paper and compared all of our structures with Dunham’s publications, and Supplementary Table 2 was reshaped accordingly.

-Although the authors state both they and the Ramakrishnan group could not see a tmRNA in the E site, I think this might be incorrect. Didn’t the Ramakrishnan group also visualize a state past the E site? In this case, isn’t the authors’ suggestion then that the tmRNA transits quickly through the E site without merit? (or without any biochemical data that supports this statement?)

Correct, so we changed the sentence as follows (new lines 376-380):

“The large tilt motion of the 30S head observed in this new intermediate TRANS* state (Supplementary Movie 1) sheds light on how tmRNA can pass through the B1b-c bridges, but we could not detect structures showing SmpB and/or the TLD in the E site. In fact, Rae *et al.*¹⁷ recently suggested that a stable E-site intermediate was unlikely, due to induced clashes with the ribosome.”

5. Figures. Most if not all the figures needs a substantial overhaul in order to see the points that authors assert in the text. In addition to new figures, better organization rather than including all four structures together in a single figure (Figure 1).

The figures have undergone extensive changes. In addition, most of the details previously presented in the extended data section are now included in new main figures to better support the points covered in the text.

-Overall the details described in the text and shown in Figure 1e-h are not observable given the current figures. For example, it is mentioned that Helix 2 points out of the ribosome but this cannot be seen in any detail.

The insets from the original Fig. 1 have been removed and the information is included in a new main Fig. 8 dedicated to the transit of tmRNA's H2 helix through the B1a, B1b, and B1c bridges during *trans*-translation. In this new figure, the overviews and close-ups combine to allow for better visualization of the position of H2.

-Extended Data Figure 3 needs to be updated as it is not clear what the point of this figure is. If the point is to show the map quality, this depiction does not provide any confidence with the high-resolution map. If the point is to show the details of GDP, switch 1 and KIR are located on EF-Tu, this does not come across. These two important points need to be separately evaluated.

-Extended Data Figure 4- again what is the purpose of this figure? These molecular interactions are difficult to discern.

We removed the Extended Data Fig. 3 and 4, and replaced them with Supplementary Fig. 6. This new figure details the interactions between the EF-Tu and the tmRNA TLD, with a focus on the GDP and kirromycin densities. We use a new viewing angle in order to better discern the residues and nucleotides involved in the interactions between the TLD and EF-Tu.

-The same is true for Extended Data Figure 5, 6, 8

These were all reworked. Fig. S5 (now Supplementary Fig. 8) was reshaped to remove the TRANS* state, as no major changes were seen between it and the TRANS state. Fig. S6 was made clearer and improved, and it is now the main Fig. 2. Fig. S8 has been simplified and replaced with the new Fig. 4, where the information is shown from a new angle which enables a better view of the rotation of SmpB's C-terminal from the ACC to the TRANS state. In each case, we also displayed the corresponding part of the electron density map to show the quality of the atomic model.

-Although the authors include the map resolutions of the four structures in Extended Data Figure 2, the authors should also include SmpB and tmRNA with their corresponding resolutions as this part of the manuscript is the most important.

We added a new Supplementary Fig. 3 which shows the local resolutions of each of the individual components of the *trans*-translation complex. As suggested, these resolutions are discussed in the text for each state.

MINOR-SmpB should be labeled in Figures 1 a-d.

Done.

6. The Conclusion/Discussion section is too brief. Although the authors state that these structures "showcase its differences from canonical translation and the fact that it does not interfere with that process", they never describe in detail what these differences are. This needs to be done in the Discussion section. Comparison with EFTu structures and with structures showing altered head domains should be performed.

This section was greatly expanded according to the comments of the reviewers, and the requested comparisons were included (see Supplementary Table 2).

Some of the phrasing is odd and should be updated (MINOR):

Introduction

-"...thanks to its third base pair," "thanks to the stop codon", "...understanding of re-registration on the MLD..."

Thanks for the comments. The paper was reworked by a native English speaker, and we hope that the style has been improved.

-“The problematic mRNA is taken up by RNase R6,....” – a better way to state this is- The non-stop mRNA is degraded by RNase R

Corrected.

-To my knowledge, uL27 is incorrect and should be relabeled bL27 (it is specific to bacteria, correct?).

Corrected.

-Second sentence of the second paragraph of Introduction - the authors should cite a review that describes how trans-translation is an attractive target for new antibiotics. A quick search brought up one review by the Keiler lab.

Good point, we added two citations: Buskirk and Green (2017), and Feaga and Keiler (2014).

- “As for SmpB, it lies on the decoding site (DS),...”

The sentence was simplified.

- “Thanks to another population of 11,059 particles obtained from the EF-G experiment,....”

-“For instance, the binding of tmRNA to a stalled ribosome is based on SmpB recognition of the problematic situation,...”

These sentences were deleted.

Reviewers' comments:

Reviewer #1 (Remarks to the Author):

Guyomar, d'Urso and colleagues have satisfactorily responded to each of my concerns about the original manuscript and have substantially improved the manuscript to showcase the important findings of their work. The authors have thoroughly described the new trans-translation structures in the context of previous structural data and made useful comparisons to canonical translation. This manuscript may become a valuable resource for future structural analysis in the context of drug binding studies and is worthy of publication.

Christopher D. Rae

Reviewer #2 (Remarks to the Author):

Major comments

1. There are ends claims of high resolution and reports of their structures at 3.0 to 3.4Å (abstract) or "four new high resolution cryo-EM structures" line 67, or "the very first high resolution 3.2Å" line 92 etc etc...While these statements are correct with respect to the average resolution, one immediately sees in the Figure S2 that this resolution is basically for the large subunit and that the small subunit, tmRNA etc appear to be more in the resolution range of 4Å or worse. I write appear because the only indication of local resolution is from small overviews of the structures. No effort has been made to show the local resolution of the individual factors to strengthen their interpretation. This needs to be done at a minimum.

We understand the concerns of Referee 2. This is why we improved Figure S2 (now Supplementary Fig. 2), and we now show a view sliced halfway through the maps to better visualize the resolutions, especially on the small subunit. Moreover, we added two new supplementary figures (3 and 4). The first of these illustrates the local resolutions of each of the individual components of the trans-translation complex, and the second shows the sharpened density maps of the large ribosomal subunit and the small ribosomal subunit body and head regions according to the local resolutions.

The authors have now added the appropriate local resolution figures, especially Sup Figure 3 showing the local resolution of tmRNA, SmpB and EFTu in the different states. Unfortunately, they did not provide any conclusion from the figures...however, as I see, the tmRNA appears to be basically 4-5Å or worse with the exception of the TLD. EF-Tu is 5-10Å and the only resolved factor is SmpB in the accommodated and translocation state that has already been described previously. SmpB appears to be partially resolved in the new pre-accommodation and translocation state with resolution at best around 4Å+.*

2. The overall quality of many of the images is poor. For the main figures, it appears that the densities were isolated using color zone and not cleaned up to distinguish what density is really belonging to the factor and what comes from the ribosome. This does not inspire confidence in the interpretation of the interactions. Many of the maps have additional density/noise flying around, perhaps the authors should consider to filter the maps based on the local resolution, particularly since the local resolution appears to be so different for different parts of the tmRNA molecule.

Actually, the densities were not isolated using color zones and, with the exception of Fig. 1, all the maps were cleaned up. Nevertheless, we understand the reviewer's concern and revised every one of the main figures. As requested, we now show the densities around the important residues/nucleotides so the conclusions are more credible. We did not filter the map based on the local resolution. However, we only

discuss interactions at the near-atomic level when the resolution is relevant (i.e. below 5 Å), as shown in Supplementary Fig. 3.

As pointed out by Reviewer 2, the local resolution differs for different regions of the ribosome. In fact, to take this into account, we did perform a multibody refinement approach that is covered in the first version of the manuscript. This used three separated overlapping bodies (50S, 30S head, and 30S body), and including the tmRNASmpB complex, and it did indeed improve the local resolution, especially for the PREACC state. To further improve the overall resolution of the PK ring, we also performed a four-body approach, but this was not helpful (results not shown). The multibody refinement cannot be applied to independent components of the tmRNA-SmpB complex because they are too small. All of this was not clearly explained in the Methods section, and we apologize. It has been corrected in the current version, and we have added the separate body maps to our EMDB/PDB deposition.

Whether or not the authors used color zone before or not, the figures were of poor quality as also pointed out by the other reviewers. Now they are considerably better. I am little concerned by the statement "we discuss interactions at the near-atomic level when the resolution is relevant (i.e. below 5Å)". 5Å is not near-atomic and this resolution does not allow sidechains to be seen or provide a molecular interpretation. Although the additional density images help partially, it's still hard to tell whether the authors have the resolution for some of the molecular descriptions. An example is that in Figure 6, Tyr24 and Tyr55 seem quite convincing based on the density, however, Val and Glu107 not so. Figure 7, His22 seems good, but His126 not so...especially since the authors show only density for the sidechain they are discussing therefore, it's hard to know if the register is correct since the density either side of this sidechain does not appear to display features. This is also seen in the supplementary images, for example Figure S9, U17 doesn't appear to have density. The density for Trp122 appears to be a huge blob...yet the authors model different rotamers. One wonders if they have the resolution for such a detailed interpretation. In Sup Fig 11, the conformation of Lys134 is not convincing as is the density for Lys138 or Gln135. Maybe His136 does stack on C1400, this would make sense, but does the map really have the resolution to allow this molecular interpretation? Everything is always stated so matter-of-factly.

3. Figure S3 is simply not acceptable. Nothing can be validated with respect to the Kir or GDP density with this poor presentation. Many of the other sup figures (S4-S8) don't show density, just endless interactions...one wonders whether the authors have the local resolution to describe these interactions. The one exception is 7a and b where overviews are shown of the density, which does not look particularly resolved and then little boxes are put on regions that one cannot see properly. The density images need to be provided for the corresponding panels b-f at the same size and orientation to be convincing. Figure S9 is not convincing to show that the nascent chain is attached to the A-site and not the P-site!

As requested, we added the cryo-electron density maps in most places. We also reworked many of the figures in other ways, including the following changes:

- Figs. S3 and S4 were removed and replaced by Supplementary Fig. 6, and the kirromycin and GDP densities are now shown.
- The information in Fig. S6 was clarified, and it is now main Fig. 2.
- Fig. S7 has a new view angle and was made clearer, and it is now main Fig 3.
- Fig. S8 was simplified and changed to main Fig. 4. Its new angle also showcases SmpB's C-terminal rotation from the ACC to the TRANS state.
- Fig. S9 was also improved and a comparison with the post-catalysis of canonical translation was added. It is now Supplementary Fig. 15.
- Two new supplementary figures, 16 and 17, were added in response to reviewer comments.

I don't want to be pedantic but do the authors seriously consider that the density images in Sup

Fig 15 support their model...I see a blob of density connecting the A and P-tRNA and then a blob of density flying in space where they have modelled the Phe residue. Its not that I don't believe that it's a post-peptide bond formation state and I am sure that the dipeptide is on the A-tRNA, but again I have the feeling that the authors over-interpret their data.

4. The conclusions section I find totally speculative and unrealistic. The idea that having "high" resolution structures of tmRNA will help drug design is a fantasy. However due the authors envisage being able to design compounds to bind at selective sites on the ribosomes to prevent tmRNA binding or changing conformations?

While screening may be sufficient for finding new molecules – and as a matter of fact, our team has developed and patented two screening methods for evaluating transtranslation (Macé et al., JMB 2017; Guyomar et al., NAR 2020) – we are convinced that high-resolution structures are essential for understanding and characterizing the actions of trans-translation inhibitors as well as being helpful for improving the activity of new chemical scaffolds. For example, as suggested by our results, such molecules could target the interface between tmRNA and SmpB to inhibit the complex, or they could target the SmpB C-terminal tail or the H5 stem-loop to alter stalled ribosome recognition or even to stabilize the B1b/c bridges to trap the tmRNA. However, we agree with the referee's remark, since this is still speculative. The conclusion was therefore toned down (lines 500-504):

"Our structures give us a better understanding of how trans-translation occurs, but they also showcase its differences and similarities with canonical translation, and confirm the fact that it does not interfere with that process. This is of primary importance for developing new antibiotics targeting trans-translation, as this depends on understanding how tmRNA and SmpB interact and how they bind to stalled ribosomes."

Yes, I find the results of the screening efforts of the authors very exciting and I also believe that the authors can use their cryoEM to provide insights into the mechanism by which such identified factors work, however, I am skeptical that the "high" resolution structures presented here are of any use to designing new compounds to target this process.

Minor comments

1. The use of PRE for the EF-Tu delivery state is somehow confusing since actually the accommodated state is a classical PRE state. Maybe the authors want to reconsider their nomenclature.

Thank you for the remark. To avoid any confusion, we replaced the "PRE" nomenclature by "PRE-ACC" throughout the manuscript and in the figures.

This seems like a good solution

Reviewer #3 (Remarks to the Author):

Thanks to the authors for their thorough explanation of the questions presented in the first review. The authors confirmed that 3 out of the 4 structures have already been published, suggesting that this manuscript presents little new information. Or one way to look at it, more effort should be included on telling the reader what new insights these structures provides. Sometimes this message gets lost (Abstract, Results)...

In general, the new manuscript is presented to a higher standard with better figures and updated text (in particular the Intro is substantially improved). However, there are still substantial issues with the

clarity of the figures and the text as noted below.

MAJOR ISSUES

Clarity in the text:

There are sections where the wording is still not clear, is ambiguous, or does not accurately reflect the field. While it is noted that the authors made substantial changes to the text, this lack of clarity prevents the important results of this manuscript to be easily grasped. See below for examples of this:

-Abstract: In the Abstract, the authors state that: "However, the precise molecular details of this process have remained mostly elusive." This is not true given all the previously solved structures of tmRNA-SmpB bound to the ribosome and in particular, the recent Science paper. This needs to be updated to fairly reflect what this particular manuscript adds to the field, namely the structure of the PRE-ACC state.

-Abstract: " Several previously unseen interactions between tmRNA-SmpB and the ribosome explain why the process does not interfere with canonical translation, and suggest how this process, which is unique to bacteria, could be targeted by new antibiotics." While the PRE-ACC state is new, what specifically does the structure show that provides information on how this state can be targeted by new antibiotics? This should be noted briefly in the Abstract.

Pre-accommodation step (starts at line 107)

-This new structure provides insights into tmRNA's PK ring, H2 and H5 and their interactions are nicely described (lines 108-140)

-Lines 136-140: The description of how H5 and PK2 interacts with of uS3 and uS5 is a new observation but the details of these interactions are minimal. Both ribosomal proteins have been studied functionally (PMID 15652481) and structurally (PMIDs 30552154, 29456023, 32427100) but the details provided here are only descriptive. It seems like a great opportunity to understand how complex RNAs engage the ribosomal helicase proteins and the authors miss out on this.

-Lines 158-163: The authors state that their structure looks different from the Thermus structure but all that is understood from this paragraph is that 16S rRNA A1493 is slightly different. So what exactly then looks different and how is this functionally relevant? Looking at Supp Fig 10, it looks like SmpB residues are different between the two bacterial species- could this be the reason for these differences? If not, what are the reasons? - Also this paragraph ends with a description of the interactions of the C-terminal tail of SmpB with G530 but is this interaction mentioned because it has not been seen before, is difference form the Thermus interactions, or what? It seems misplaced...

-Lines 192-195: How the C-terminal tail of SmpB interacts with H5 is a new observation but little weight is given to why this is important. The authors make the statement that "This specific interaction is probably instrumental in properly positioning the MLD and SmpB to ensure the correct re-registration later on." What does this mean? This is a very open ended statement...

Accommodation step: (starts at line 204)

-Lines 207-9): "In contrast with the structure recently described by Rae et al.17, our structure's improved resolution of 3.1 Å (Supplementary Fig. 2) allows for better analysis of the details of the interactions between tmRNA, SmpB, and the ribosome.". Supp Fig 2 does not show this. Another question is- is this statement really needed? How much better is the resolution reported here of these features and what new interactions are you able to discern? Once statements like this are made, there needs to be some backup of what improvements your structures provide.

-Lines 225-7: Is the N terminus of bL27 built? There information should be provided in the Results (it does come up but much later in the Discussion). Related to this, although in the Supp Fig S15 legend, the authors state: "The TLD closely resembles an aminoacylated tRNA, and the flexible N-terminal arm of bL27 is in the same conformation as during canonical translation.", most, if not all, structures do not contain the first 15 residues of bL27 because of poor map quality. So what are the authors comparing their structures to?

-Line 242-3: "The tmRNA H5 stem-loop still interacts with uS4 Arg47, and it is more tightly packed on

the C-terminal domain of uS3 and on uS5 (Fig. 2f).” See comment about these ribosomal helicase proteins- what is the significance of these interactions?

TRANS structure (starts at line 252)

-Lines 258-60: “The tmRNA pseudoknot ring is now fully distorted but remains well-outlined and stable (Supplementary Fig. 3)”. How can the PK ring be fully distorted but at the same time, stable?

-Lines 269-70: “As for G530, it interacts with the third resume’s codon A92” What is the ‘third resume codon’? This needs defining.

-Line 277-9: “As for Tyr24, it lies between tmRNA’s A86 and G87, and by forming weak hydrogen bonds with both it serves as a secondary checkpoint, ensuring a finer control of the sequence and limiting the risk of frameshifting.” How? Clearly other experiments tested this?

TRANS* structure (starts at line 342)

While the TRANS* section better reflects the new insights from this structure, these paragraphs should be carefully rewritten to ensure clarity but also to remove grammatical issues.

-Lines 347-8: In this sentence, the authors propose that this structure improves how we understand the exit of tmRNA-SmpB from the P site and how this opens B1 bridges. This seems like an important point and the authors come back to this in one paragraph later. Are these the same bridges that EF-G has to open to move normal tRNA through during translocation? Seems like an interesting discussion point given that tmRNA is translocated much more slowly by EF-G than tRNAs....

-Lines 349-351: The authors should describe ratcheting within the context of intersubunit movement relative to each other which is distinct from swiveling.

Discussion

-Lines 394-398: While again I agree the new state seen here is interesting, the authors oversell the other structures that have already been published by others.

-Lines 405-415: The discussion section of how both the SmpB C-terminal and H5 SL may close the mRNA channel is intriguing. However, the extrapolation of the mutational data of Met155 and Ile154 is not clear. How are these residues, together with H5, serving as a mechanical sensor (I also don’t know what ‘early’ refers to either)? How does this lead to the correct positioning of the codon?

-Line 427: The authors mention a Noller paper that shows uS3/4 positioned closely to a SL. Do the other papers (PMIDs 29456023, 32427100) show the same result as the Noller paper (PMID 30552154)? Seems like an obvious discussion point. -Also what uS3/S4 residues engage H5 and are these the same residues that engage other structured RNAs as shown in these two papers or are they different?

-Lines 429-452: The description of the TRANS structure and its functional implications is much improved.

-Lines 492-503: The recent structure of a trans-translation inhibitor bound to the ribosome interestingly shows a change in the placement of the L27 tail. The authors describe these interactions briefly but their overall points aren’t well articulated. While the N terminus of L27 in their ACC structure is similar to when a normal aminoacylated tRNA is bound at the A site, how then can the inhibitor be thought to impede accommodation of tmRNA-SmpB? Since the N term of L27 looks the same in both structures wouldn’t it make more sense for it to prevent a downstream event of the ACC step? The statement in lines 500-2 seems to then discuss their results in general terms but then the last part comes back to the point of the trans-translation inhibitor: “Our structures give us a better understanding of how trans-translation occurs, but they also showcase its differences and similarities with canonical translation, and confirm the fact that it does not interfere with that process.” This section needs clarifying.

Figures:

-The figures containing maps need to be remade using a different way to show density (ie use transparencies) otherwise they make the figures worse. These include Fig 2b,c,e,f,h,I (and some of these maps do not appear to be aligned with the models!), Fig 3, Fig 4 (do you need to see the map of 16S?), Fig 5 (do you need to see the map of tmRNA?), Fig 6, Fig 8, and Supp Fig 9 (hard to focus on

what is important). In contrast, the maps shown in Fig 7 and Supp Fig 11 look good but it appears this is using a different program to generate the figures?

-The supplementary figure numbering is out of order and is incredibly confusing. First supplementary figure to be called out in the manuscript is S5 (line 114). This needs updating.

-Figure 9 should be labeled.

-The phosphate groups of most of RNA molecules including tmRNA and rRNA are above the ribbon denoting the backbone. This can be easily fixed by either having the ribbon go through the phosphate or turning off the phosphate group. Either way, it should be fixed because in some cases there is no connectivity in the RNA.

MINOR ISSUES

Text:

-Line 37: tmRNA is not completely ssRNA (has a tRNA domain!) so this should be updated.

-Line 50, 211, 344: "thanks". A better word should be used as noted previously

-Line 110: Issues with again state "very first high resolution"

-Lines 135-7: "PK2's nucleotides C183 to A185 interact with residues Arg72, Pro73, and Ile77, which belong to the type II K-homology (KH2) RNA-binding domain of uS3 (Fig. 2b)25." Is it appropriate to cite Noller's paper here?

-Line 213: If an interaction becomes closer, that is fine to describe as "closer" but the term "tighter" should not be used as this implies some sort of binding thermodynamics.

-Line 234: The text jumps from Fig 5 to Fig 9 in the text

-"16S rRNA H44 helix"- helices of the small subunit are lowercase

-Lines 247-9: "Arg153 and His159 are bound ionically to the phosphate groups between U119, U120, and A121,"... This is awkward and it is not clear this is scientifically correct. Why not just say "interact".

-Line 349: need to define what A/P refers to.

-Lines 453-5: This sentence should be rewritten for clarity

-Line 463-4: What do the authors mean by "we only observed a moderate interaction between His22 and A1493"? Do they mean stacking interaction? If so, they should use this terminology.

Figures:

-Supplementary Fig 6: panel A has the TLD residues labeled but the EF-Tu residues are not (although it says they are in the legend). Please label.

-Supplementary Fig 5- overlay between previously determined structure and the structure presented here. The authors should include how these structures were aligned and provide an rmsd between the two to quantify the differences

-Supplementary Fig 7: silver and gray are the same colors so please update the legend.

-Grammatical errors in the legend for Supplementary Fig 9

Reviewer #1 (Remarks to the Author):

Guyomar, d'Urso and colleagues have satisfactorily responded to each of my concerns about the original manuscript and have substantially improved the manuscript to showcase the important findings of their work. The authors have thoroughly described the new trans-translation structures in the context of previous structural data and made useful comparisons to canonical translation. This manuscript may become a valuable resource for future structural analysis in the context of drug binding studies and is worthy of publication.

Christopher D. Rae

Reviewer #2 (Remarks to the Author):

Major comments

1. There are ends claims of high resolution and reports of their structures at 3.0 to 3.4Å (abstract) or “four new high resolution cryo-EM structures” line 67, or “the very first high resolution 3.2Å” line 92 etc etc...While these statements are correct with respect to the average resolution, one immediately sees in the Figure S2 that this resolution is basically for the large subunit and that the small subunit, tmRNA etc appear to be more in the resolution range of 4Å or worse. I write appear because the only indication of local resolution is from small overviews of the structures. No effort has been made to show the local resolution of the individual factors to strengthen their interpretation. This needs to be done at a minimum.

We understand the concerns of Referee 2. This is why we improved Figure S2 (now Supplementary Fig. 2), and we now show a view sliced halfway through the maps to better visualize the resolutions, especially on the small subunit. Moreover, we added two new supplementary figures (3 and 4). The first of these illustrates the local resolutions of each of the individual components of the trans-translation complex, and the second shows the sharpened density maps of the large ribosomal subunit and the small ribosomal subunit body and head regions according to the local resolutions.

The authors have now added the appropriate local resolution figures, especially Sup Figure 3 showing the local resolution of tmRNA, SmpB and EFTu in the different states. Unfortunately, they did not provide any conclusion from the figures...however, as I see, the tmRNA appears to be basically 4-5Å or worse with the exception of the TLD. EF-Tu is 5-10Å and the only resolved factor is SmpB in the accommodated and translocation state that has already been described previously. SmpB appears to be partially resolved in the new pre-accommodation and translocation state with resolution at best around 4Å+.*

As suggested by reviewer 2 we added new conclusions on each of the steps, stating the various local resolutions and subsequently describing the flexible vs more stable regions:

In the PRE-ACC step (lines 118 to 122): *“The complex is quite dynamic and the PK ring is flexible, resulting in a local resolution that fluctuates between 3.5 and 10 Å. However, the tips of the H5 stem loop and SmpB C-terminal tail are seen ~3.5Å, PK2 is ~4.5Å, and the interfaces between tmRNA, SmpB and EF-Tu are ~5.5Å, which allows for the molecular description of specific interactions.”*

In the ACC step (lines 236 to 242): *“The structure resolution is 3.1 Å (Supplementary Fig. 2), with local resolutions fluctuating between 3 and 10 Å. Unsurprisingly, the PK ring is the least well-resolved part of the complex. The H5 stem-loop, TLD, SmpB, and some parts of PK2 are observed at resolutions better than 3.5 Å, allowing for detailed molecular description of the interactions between tmRNA, SmpB and the ribosome. In contrast with the structure recently described by Rae et al.¹⁷, our structure also includes a tRNA^{phe} in the P site, and no tRNA in the E site.”*

And in the TRANS step (lines 300 to 302): *“The local resolution of the TLD and SmpB C-terminal tail is ~3.25 Å, the rest of SmpB and the MLD are ~3.5 Å, and the resolution of the H5 stem-loop and PK2 fluctuates between 3.75 and 5 Å.”*

Note that we were very careful to only discuss interactions for the most well resolved parts of the system ie C-terminal tail of SmpB, TLD, tip of H5 and MLD in the TRANS state parts of the system. However, it seems to us very important to remind that heterogenous resolution is always observed in cryo-EM data and that maps are usually compared based on the global resolution. In that sense we are one Angstrom below than the previously published data from Rae et al., which allow us to describe side chains with more accuracy.

2. The overall quality of many of the images is poor. For the main figures, it appears that the densities were isolated using color zone and not cleaned up to distinguish what density is really belonging to the factor and what comes from the ribosome. This does not inspire confidence in the interpretation of the interactions. Many of the maps have additional density/noise flying around, perhaps the authors should consider to filter the maps based on

the local resolution, particularly since the local resolution appears to be so different for different parts of the tmRNA molecule.

Actually, the densities were not isolated using color zones and, with the exception of Fig. 1, all the maps were cleaned up. Nevertheless, we understand the reviewer's concern and revised every one of the main figures. As requested, we now show the densities around the important residues/nucleotides so the conclusions are more credible. We did not filter the map based on the local resolution. However, we only discuss interactions at the near-atomic level when the resolution is relevant (i.e. below 5 Å), as shown in Supplementary Fig. 3.

As pointed out by Reviewer 2, the local resolution differs for different regions of the ribosome. In fact, to take this into account, we did perform a multibody refinement approach that is covered in the first version of the manuscript. This used three separated overlapping bodies (50S, 30S head, and 30S body), and including the tmRNASmpB complex, and it did indeed improve the local resolution, especially for the PREACC state. To further improve the overall resolution of the PK ring, we also performed a four-body approach, but this was not helpful (results not shown). The multibody refinement cannot be applied to independent components of the tmRNA-SmpB complex because they are too small. All of this was not clearly explained in the Methods section, and we apologize. It has been corrected in the current version, and we have added the separate body maps to our EMDB/PDB deposition.

Whether or not the authors used color zone before or not, the figures were of poor quality as also pointed out by the other reviewers. Now they are considerably better. I am little concerned by the statement "we discuss interactions at the near-atomic level when the resolution is relevant (i.e. below 5Å)". 5Å is not near-atomic and this resolution does not allow sidechains to be seen or provide a molecular interpretation.

The "near-atomic" statement is now absent of the paper and we limit the use of "high resolution" to the minimum.

What we wanted to say is that we only considered the most resolved regions to discuss the interactions. To do so we relied on the multibody refinement results and while the resolution fluctuates, it is locally good enough to describe molecular interactions. We modified the text to make it clear (see above) and state the local resolution of each considered regions. To reassure the reviewers and the readers of the quality of the maps and models we also modified Supplementary Fig. 3 by adding close-ups of the TLD, SmpB and MLD with all the side chains and the electron density map. We also modified Figure 6, Figure 7 and Supplementary Fig. 15 (see below).

Although the additional density images help partially, it's still hard to tell whether the authors have the resolution for some of the molecular descriptions. An example is that in Figure 6, Tyr24 and Tyr55 seem quite convincing based on the density, however, Val and Glu107 not so. Figure 7, His22 seems good, but His126 not so...especially since the authors show only density for the sidechain they are discussing therefore, it's hard to know if the register is correct since the density either side of this sidechain does not appear to display features. This is also seen in the supplementary images, for example Figure S9, U17 doesn't appear to have density. The density for Trp122 appears to be a huge blob...yet the authors model different rotamers. One wonders if they have the resolution for such a detailed interpretation. In Sup Fig 11, the conformation of Lys134 is not convincing as is the density for Lys138 or Gln135.

Maybe His136 does stack on C1400, this would make sense, but does the map really have the resolution to allow this molecular interpretation? Everything is always stated so matter-of-factly.

Local resolutions for the regions pointed out by referee 2 are better than 4 Å but we agree that the orientation of the view can be sometimes misleading. Moreover, it is known that for aspartate and glutamate, atoms beyond C-β are less well defined in cryo-EM (*Hryc et al., PNAS 2017*), which is obviously the case for Glu107.

Certainly it is hard to assess the model quality without all of the side-chains, but for clarity reasons, we thought it was better to only focus on the important residues. Nevertheless, to prove the accuracy of our observations, we modified Figure 6, Figure 7 and Supplementary Fig. 15, by adding a new panel displaying all the side chains as well as the electron density map colored to reflect the local resolution. We did not modify the Supplementary Figure 11c as it depicts the same region illustrated in Figure 7. As commented by the reviewers, it is hard to interpret Supplementary Fig. 9, we therefore decided to modify it by focusing on the most important residues and by removing the density to increase clarity.

3. Figure S3 is simply not acceptable. Nothing can be validated with respect to the Kir or GDP density with this poor presentation. Many of the other sup figures (S4-S8) don't show density, just endless interactions...one wonders whether the authors have the local resolution to describe these interactions. The one exception is 7a and b where overviews are shown of the density, which does not look particularly resolved and then little boxes are put on regions that one cannot see properly. The density images need to be provided for the corresponding panels b-f at the same size and orientation to be convincing. Figure S9 is not convincing to show that the nascent chain is attached to the A-site and not the P-site!

As requested, we added the cryo-electron density maps in most places. We also reworked many of the figures in other ways, including the following changes:

- Figs. S3 and S4 were removed and replaced by Supplementary Fig. 6, and the kirromycin and GDP densities are now shown.

- The information in Fig. S6 was clarified, and it is now main Fig. 2.

- Fig. S7 has a new view angle and was made clearer, and it is now main Fig 3.

- Fig. S8 was simplified and changed to main Fig. 4. Its new angle also showcases SmpB's C-terminal rotation from the ACC to the TRANS state.

- Fig. S9 was also improved and a comparison with the post-catalysis of canonical translation was added. It is now Supplementary Fig. 15.

- Two new supplementary figures, 16 and 17, were added in response to reviewer comments.

I don't want to be pedantic but do the authors seriously consider that the density images in Sup Fig 15 support their model...I see a blob of density connecting the A and P-tRNA and then a blob of density flying in space where they have modelled the Phe residue. Its not that I

don't believe that it's a post-peptide bond formation state and I am sure that the dipeptide is on the A-tRNA, but again I have the feeling that the authors over-interpret their data.

We built the model including the dipeptide as it fits well with what is expected from the biology of the complex. We compared our structure with the 2.55 Å Crystal structure of the *Thermus thermophilus* 70S ribosome in the post-catalysis state of peptide bond formation containing dipeptidyl-tRNA (PDB ID 1VY5). After rigid body fitting the whole *Thermus thermophilus* 70S ribosome in our map, the structural similarity of the PTC was evident and we modeled the dipeptide in the unattributed “blob” of density between the P-site tRNA and the TLD.

However, we certainly do not want to give the feeling that we over-interpret the data. We therefore removed the statement “*since the phenylalanine is already bound to Ala tmRNA*” from the text (Line 256-257). We also modified Supplementary Fig. 15 to better show the structural similarity between our data and the post-catalysis state of peptide bond. We also displayed the local resolution to convince the reader of the quality of the data, and we didn't modeled the dipeptide in the remaining density between the P-site tRNA and the TLD.

4. The conclusions section I find totally speculative and unrealistic. The idea that having “high” resolution structures of tmRNA will help drug design is a fantasy. However due the authors envisage being able to design compounds to bind at selective sites on the ribosomes to prevent tmRNA binding or changing conformations?

While screening may be sufficient for finding new molecules – and as a matter of fact, our team has developed and patented two screening methods for evaluating transtranslation (Macé et al., JMB 2017; Guyomar et al., NAR 2020) – we are convinced that high-resolution structures are essential for understanding and characterizing the actions of trans-translation inhibitors as well as being helpful for improving the activity of new chemical scaffolds. For example, as suggested by our results, such molecules could target the interface between tmRNA and SmpB to inhibit the complex, or they could target the SmpB C-terminal tail or the H5 stem-loop to alter stalled ribosome recognition or even to stabilize the B1b/c bridges to trap the tmRNA. However, we agree with the referee's remark, since this is still speculative. The conclusion was therefore toned down (lines 500-504):

“Our structures give us a better understanding of how trans-translation occurs, but they also showcase its differences and similarities with canonical translation, and confirm the fact that it does not interfere with that process. This is of primary importance for developing new antibiotics targeting trans-translation, as this depends on understanding how tmRNA and SmpB interact and how they bind to stalled ribosomes.”

Yes, I find the results of the screening efforts of the authors very exciting and I also believe that the authors can use their cryoEM to provide insights into the mechanism by which such identified factors work, however, I am skeptical that the “high” resolution structures presented here are of any use to designing new compounds to target this process.

We believe that the new interface between the MLD and SmpB in the TRANS conformation or the details of the interface between the accommodated tmRNA-SmpB and the ribosome (both around 3.5 Å resolution) could be of use for rational drug design. In fact, cryo-EM has already

made contributions to advancing small molecules into human trials (see Wigge *et al.* Drug Discovery Today: Technologies DOI: 10.1016/j.ddtec.2020.12.003 for a review) even with data at global overall resolution of more than 3 Å (see Wei *et al.* (2019) Nature **568**: 566-570 for example). We modified the last paragraph of the manuscript to clarify this point (Lines 616- 620).

Minor comments

1. The use of PRE for the EF-Tu delivery state is somehow confusing since actually the accommodated state is a classical PRE state. Maybe the authors want to reconsider their nomenclature.

Thank you for the remark. To avoid any confusion, we replaced the “PRE” nomenclature by “PRE-ACC” throughout the manuscript and in the figures.

This seems like a good solution

Thanks again for your suggestion, this indeed makes it clearer.

Reviewer #3 (Remarks to the Author):

Thanks to the authors for their thorough explanation of the questions presented in the first review. The authors confirmed that 3 out of the 4 structures have already been published, suggesting that this manuscript presents little new information. Or one way to look at it, more effort should be included on telling the reader what new insights these structures provides. Sometimes this message gets lost (Abstract, Results)...

In general, the new manuscript is presented to a higher standard with better figures and updated text (in particular the Intro is substantially improved). However, there are still substantial issues with the clarity of the figures and the text as noted below.

MAJOR ISSUES

Clarity in the text: There are sections where the wording is still not clear, is ambiguous, or does not accurately reflect the field. While it is noted that the authors made substantial changes to the text, this lack of clarity prevents the important results of this manuscript to be easily grasped. See below for examples of this:

-Abstract: In the Abstract, the authors state that: “However, the precise molecular details of this process have remained mostly elusive.” This is not true given all the previously solved structures of tmRNA-SmpB bound to the ribosome and in particular, the recent Science paper. This needs to be updated to fairly reflect what this particular manuscript adds to the field, namely the structure of the PRE-ACC state.

We totally agree and our goal is certainly not to underestimate the impact of the previously published structures of tmRNA. We provided a more detailed abstract highlighting the originality of our work (lines 17-27):

“These include the first high-resolution structure of the whole pre-accommodated state, as well as structures of the accommodated state, the translocated state, and a new translocation intermediate. Together, they shed light on the movements of the tmRNA-SmpB complex in the ribosome, from its delivery by the elongation factor EF-Tu to its passage through the ribosomal A and P sites after the opening of the B1 bridges. Notably, we describe previously unseen interactions not just between tmRNA and SmpB, but also between the tmRNA-SmpB complex and the ribosome. These explain why the process does not interfere with canonical translation, and suggest how this mechanism, which is unique to bacteria, could be impaired by new antibiotics targeting these new interfaces.”

-Abstract: “Several previously unseen interactions between tmRNA-SmpB and the ribosome explain why the process does not interfere with canonical translation, and suggest how this process, which is unique to bacteria, could be targeted by new antibiotics.” While the PRE-ACC state is new, what specifically does the structure show that provides information on how this state can be targeted by new antibiotics? This should be noted briefly in the Abstract.

We describe yet unseen interactions such as Trp122 and tmRNA (throughout the 4 states), the H5 stem loop and SmpB C-terminal tail (in the PRE-ACC and ACC) or the MLD and SmpB (TRANS and TRANS*). We think that, for example, the new interface between the MLD and SmpB in the TRANS conformation or the details of the interface between the accommodated tmRNA-SmpB and the ribosome (Both around 3.5 Å resolution) could be of use for rational drug design and we modified the abstract accordingly. However, this cannot be extensively described in the abstract, so it was done in the discussion section:

*Abstract: “These explain why the process does not interfere with canonical translation, and suggest how this mechanism, which is unique to bacteria, could be impaired by new antibiotics **targeting these new interfaces.**”*

Discussion (lines 616-620): “We think that, for example, our detailed description of the interactions between the accommodated tmRNA-SmpB and the ribosome, as well as the newly described interface between the MLD and SmpB in the TRANS conformation, should be useful for the recently proposed anti-trans-translation strategies^{38,61-63} and even in rational drug design.”

Pre-accommodation step (starts at line 107)

-This new structure provides insights into tmRNA’s PK ring, H2 and H5 and their interactions are nicely described (lines 108-140)

-Lines 136-140: The description of how H5 and PK2 interacts with of uS3 and uS5 is a new observation but the details of these interactions are minimal. Both ribosomal proteins have been studied functionally (PMID 15652481) and structurally (PMIDs 30552154, 29456023, 32427100) but the details provided here are only descriptive. It seems like a great opportunity to understand how complex RNAs engage the ribosomal helicase proteins and the authors miss out on this.

Yes indeed, as such we chose to discuss these interactions in the discussion section (as they are observed in the four states). PMIDs 15652481 and 30552154 were already quoted (refs 49

and 25, respectively). We nevertheless modified the text to introduce the question of the ribosome helicase activity earlier (lines 149-158):

“The H5 stem-loop is well-defined and interacts with the uS3, uS4 and uS5 proteins involved in the helicase activity of the ribosome^{25,26} (Fig. 2a). Its nucleotides G114 and C115 are stabilized by residues Arg132 and Arg136 of the uS3 C-terminal domain. U119 interacts with uS4 residue Arg47. U120, the stop codon’s first nucleotide, lies on top of uS5 residue Ile60, while its second, A121, is at the interface between uS3 and uS5 (Fig. 2c). Interestingly, most of these interactions are similar to those previously described for structured mRNAs^{27,28}. However, compared to those mRNAs, the H5 stem-loop mostly differs in how it interacts with uS5. Indeed, as the tmRNA is not yet engaged in the mRNA channel, there are no interactions with Arg20, Phe33, or Val56, and instead the tip of the stem-loop rests on top of the α 1 helix residues Ile60 and Gln61.”

We also added sentence in the “Accommodation step” section (lines 281-287):

“The tmRNA H5 stem-loop still interacts with helicases uS4 and uS5, and it is even more tightly packed on the C-terminal domain of uS3 (Fig. 2f). The interactions with uS3 residues Arg132, Lys135, and Arg136 and uS4 residue Arg47 are maintained, while new interactions are observed with uS3 Arg72, Asn140 and Arg143. However, both uS3 Arg131 and uS4 Arg44, which were previously shown to be critical for helicase activity²⁵, point away from the helix.”

and in the “Translocation step” section (lines 311-319):

“The MLD is stretched and inserted into the mRNA channel (Fig. 1g). It interacts with the helicases uS3, uS4 and uS5 in a way that resembles the binding of structured mRNAs (Supplementary Fig. 17)^{27,28}. The contacts observed during the PRE-ACC and ACC steps are maintained. However, the MLD also interacts with the uS3 α 5 helix (Gln123, Arg126 and Arg127) and β 5 sheet (Ile162, Arg164, Glu166) and lies on the surface of uS5 (Phe31, Glu55 and Val56). The tmRNA nucleotide A97 interacts with Arg131 and is stacked on uS5 Val56, which puts it right at the centre of the proximal helicase active site⁴⁰.”

We also largely extended our discussion (lines 495-526) added a new Supplementary Fig. 18, and cite PMID 29456023 (ref 27). We chose not to talk about 32427100 since in that case the mRNA stem-loop binds directly to the A-site, without going through the mRNA channel.

-Lines 158-163: The authors state that their structure looks different from the Thermus structure but all that is understood from this paragraph is that 16S rRNA A1493 is slightly different. So what exactly then looks different and how is this functionally relevant? Looking at Supp Fig 10, it looks like SmpB residues are different between the two bacterial species- could this be the reason for these differences? If not, what are the reasons? - Also this paragraph ends with a description of the interactions of the C-terminal tail of SmpB with G530 but is this interaction mentioned because it has not been seen before, is difference from the Thermus interactions, or what? It seems misplaced...

Good point. We focused on A1492 and A1493 because they are crucial for canonical decoding and we wondered whether tmRNA-SmpB perform a “pseudo-decoding”. To us, the main difference between the two structures comes from the presence of the pseudoknot ring within our complex. In our structure the H5 stem-loop is stalled at the entrance channel and it

contacts with SmpB C-terminal. This slows down the movement of the complex and, as a result, SmpB is at an earlier stage of pre-accommodation than in the crystal structure. We hypothesize that this is the reason why A1493 is not yet flipped out in our structure compared to the truncated complex depicted in the crystal structure and added a sentence to clarify this point (lines 180-186):

“Since the main difference between the two complexes comes from our inclusion of the pseudoknot ring, we hypothesize that H5 stalling at the entrance channel slows down the movement of the complex. The interaction between H5 and the SmpB C-terminal tail (see below) would therefore result in the stabilization of an earlier stage of pre-accommodation, explaining the difference in SmpB position/conformation (Supplementary Fig. 5) and why A1493 is not yet flipped out”

As for G530-SmpB interactions, they are described here because G530 is part of the decoding center. They are not new (Neubauer et al., 2012).

-Lines 192-195: How the C-terminal tail of SmpB interacts with H5 is a new observation but little weight is given to why this is important. The authors make the statement that “This specific interaction is probably instrumental in properly positioning the MLD and SmpB to ensure the correct re-registration later on.” What does this mean? This is a very open ended statement...

This is described in the discussion section (lines 472-493), therefore we decided to discard this sentence from this early chapter.

Accommodation step: (starts at line 204)

-Lines 207-209: “In contrast with the structure recently described by Rae et al.17, our structure’s improved resolution of 3.1 Å (Supplementary Fig. 2) allows for better analysis of the details of the interactions between tmRNA, SmpB, and the ribosome.”. Supp Fig 2 does not show this. Another question is- is this statement really needed? How much better is the resolution reported here of these features and what new interactions are you able to discern? Once statements like this are made, there needs to be some backup of what improvements your structures provide.

In average we did improve the overall resolution by 1Å compared to Rae et al, which allows for a better atomic model. This is especially evident when looking at the contacts between tmRNA and SmpB (see for example the ACC state in the following figure, which is not for publication)

Comparison of the interface between the TLD and SmpB in the ACC conformation.

a) Our work (global resolution 3.2Å). b) Rae et al. structure (global resolution 4.3Å). (redacted)

Both maps are displayed at Rae's recommended contour level of 3.7σ.

However, and again, we do not intend to underestimate the work by Rae et al. and we did change this sentence (lines 236-242):

“The structure resolution is 3.1 Å (Supplementary Fig. 2), with local resolutions fluctuating between 3 and 10 Å. Unsurprisingly, the PK ring is the least well-resolved part of the complex. The H5 stem-loop, TLD, SmpB, and some parts of PK2 are observed at resolutions better than 3.5 Å, allowing for detailed molecular description of the interactions between tmRNA, SmpB and the ribosome. In contrast with the structure recently described by Rae et al.¹⁷, our structure also includes a tRNA^{phe} in the P site, and no tRNA in the E site.”

-Lines 225-7: Is the N terminus of bL27 built? There information should be provided in the Results (it does come up but much later in the Discussion).

Yes, it is built and we modified the text as follows (Lines 257-259):

“The N-terminal arm of bL27, known to play a critical role in tRNA substrate stabilization during the peptidyl transfer reaction³⁴⁻³⁷, is well resolved, which allowed us to build a complete atomic model.”

Related to this, although in the Supp Fig S15 legend, the authors state: “The TLD closely resembles an aminoacylated tRNA, and the flexible N-terminal arm of bL27 is in the same conformation as during canonical translation.”, most, if not all, structures do not contain the first 15 residues of bL27 because of poor map quality. So what are the authors comparing their structures to?

Indeed, a lot of structures do not contain the first 15 residues (most probably because the N-terminal arm is quite flexible) but it is not the case for PDB structures (4V4H, 4V9O, 4V4Q, 3J9Z, 3JA1, 4V51, 4V5C and 1VY5). We compared our work to 1VY5 as it has the best resolution (2.55 Å).

-Line 242-3: “The tmRNA H5 stem-loop still interacts with uS4 Arg47, and it is more tightly

packed on the C-terminal domain of uS3 and on uS5 (Fig. 2f).” See comment about these ribosomal helicase proteins- what is the significance of these interactions?

Indeed, this is a very important and original point. As stated in the pre-accommodation section, we slightly expanded the explanations in the current version, including the description of the residues that are of importance during the process (ie in the pre-accommodation, accommodation and translocation sections of the manuscript). We also expanded the discussion to better address the helicase question.

TRANS structure (starts at line 252)

-Lines 258-60: “The tmRNA pseudoknot ring is now fully distorted but remains well-outlined and stable (Supplementary Fig. 3)”. How can the PK ring be fully distorted but at the same time, stable?

OK, this is a language mistake. We did remove “*and stable*”

-Lines 269-70: “As for G530, it interacts with the third resume’s codon A92” What is the ‘third resume codon’? This needs defining.

Typo: please read “*the third resume codon’s nucleotide A92*”

-Line 277-9: “As for Tyr24, it lies between tmRNA’s A86 and G87, and by forming weak hydrogen bonds with both it serves as a secondary checkpoint, ensuring a finer control of the sequence and limiting the risk of frameshifting.” How? Clearly other experiments tested this?

This is a hypothesis, the sentence was modified by adding: “*might*”

TRANS* structure (starts at line 342)

While the TRANS* section better reflects the new insights from this structure, these paragraphs should be carefully rewritten to ensure clarity but also to remove grammatical issues.

This section has been heavily reworked, as requested (see below)

-Lines 347-8: In this sentence, the authors propose that this structure improves how we understand the exit of tmRNA-SmpB from the P site and how this opens B1 bridges. This seems like an important point and the authors come back to this in one paragraph later. Are these the same bridges that EF-G has to open to move normal tRNA through during translocation? Seems like an interesting discussion point given that tmRNA is translocated much more slowly by EF-G than tRNAs....

During canonical translocation B1a and B1b are disrupted by EF-G while B1c seems to remain closed (Liu and Fredrick JMB 2016). However, we demonstrate in our work that

B1b,c play a major role during trans-translation. This is discussed in the later paragraph (lines 433-448)

-Lines 349-351: The authors should describe ratcheting within the context of intersubunit movement relative to each other which is distinct from swiveling.

OK, it is now done lines 408-415.

Discussion

-Lines 394-398: While again I agree the new state seen here is interesting, the authors oversell the other structures that have already been published by others.

Sorry we did not mean to be misleading. We changed the sentence as follows (Lines 465-471):

“In this study, we present four cryo-EM structures of the trans-translation machinery, providing unprecedented details of the interactions between tmRNA, SmpB, and the ribosome throughout the process of trans-translation. These shed light on how the tmRNA-SmpB complex recognizes stalled ribosomes, how it selects the codon needed to resume translation of the tag, and how it crosses the various ribosomal bridges without interfering with canonical translation.”

-Lines 405-415: The discussion section of how both the SmpB C-terminal and H5 SL may close the mRNA channel is intriguing. However, the extrapolation of the mutational data of Met155 and Ile154 is not clear. How are these residues, together with H5, serving as a mechanical sensor (I also don't know what 'early' refers to either)? How does this lead to the correct positioning of the codon?

The term “mechanical sensor” was discarded and the discussion was modified as follows (Lines 488-493):

“The fact that Met155 and Ile154 mutations impair tagging but not binding suggest that these residues are mandatory for the correct positioning of the resume codon within the DC. As the H5 stem-loop interacts with SmpB exactly in this region (Figure 5), we hypothesize that the contact between H5 and the C-terminal tail provide a second anchoring point (the first being the SmpB-TLD interaction) and this facilitates the correct positioning of the resume codon during translocation.”

-Line 427: The authors mention a Noller paper that shows uS3/4 positioned closely to a SL. Do the other papers (PMIDs 29456023, 32427100) show the same result as the Noller paper (PMID 30552154)? Seems like an obvious discussion point. -Also what uS3/S4 residues engage H5 and are these the same residues that engage other structured RNAs as shown in these two papers or are they different?

Yes PMID 29456023 show similar results while 32427100 do not display any interaction between uS3/4 (see pre-accommodation section). The detail of the interactions was added to the pre-accommodation, accommodation and translocation sections of the manuscript and we added

a new supplementary figure (Supplementary Fig. 18) and extended the discussion as follows (lines 495-527):

“The tmRNA H5, PK2, and MLD domains also make numerous contacts with the uS3, uS4 and uS5 proteins at the entrance of the mRNA channel. These three proteins are known to be instrumental in ribosome helicase activity²⁵. In the PRE-ACC and ACC states, the H5 stem-loop strongly interacts with residues in the uS3 α 6 helix, uS4 α 1- α 2 linker, and uS5 α 1 helix (Supplementary Fig. 17a-b). Interestingly, this mostly involves the same positively charged residues that have been shown to interact with structured mRNAs^{27,28}. These notably include uS3 residues Arg132 and Lys135, and uS4 Arg47, all of which are known to be critical for helicase activity²⁵. While H5 lies on the distal helical active site, it is not inserted deep enough in the mRNA channel to reach the proximal active site⁴⁰. This may explain why the H5 stem-loop remains highly structured at these early stages. However, because of its strong interaction with the 30S, H5 could play the role of a fulcrum, helping with the mechanical unfolding and correct positioning of the MLD in the mRNA channel during translocation. In the TRANS state, H5 is flipped toward uS2 and replaced by the single-stranded portion of the MLD in a manner resembling the binding of structured mRNAs^{27,28} (Supplementary Fig. 17c-d). The backbone of the single-stranded portion of the MLD is extended and straightened, and SmpB helps properly place the resume codon in the A site. This brings the tmRNA nucleotide A97 (the 11th nucleotide when counting from the P site) to the centre of the helicase proximal active site, while part of the MLD interacts downstream with the distal active site. This confirms that the uS3 protein favors binding of extended single-strand mRNAs at the entrance of the tunnel, and is consistent with the hypothesis that product stabilization plays a role in the unwinding of structured mRNAs by the ribosomal helicase^{28,40}. While the H5 helix is yet to be unfolded, the comparison between the TRANS and TRANS states also provides some information on the helicase mechanism (Supplementary Fig. 18). Indeed, the most apparent difference between these two states is an extra 8° of both rotation and tilt of the 30S head in TRANS* (Supplementary Table 2). Although this state is not as well-resolved as TRANS, the PK2, H5, and MLD domains remain quite well defined, except for a small portion of the single-stranded MLD right in the proximal helicase active site. This suggests that the forward head rotation and tilt lengthen the mRNA channel and destabilize the MLD. The canonical translation could then resume, allowing for the unfolding of H5 (which contains a large portion of the tagging sequence, see Supplementary Fig. 1) during the ensuing steps. Together, our structures confirm the involvement of the positively charged residues of uS3 and uS4 in both the basic mRNA-binding activity of the ribosome as well as its helicase activity, and they are consistent with the recent tandem model for the ribosomal helicase⁴⁰.”*

-Lines 429-452: The description of the TRANS structure and its functional implications is much improved.

Thanks

-Lines 492-503: The recent structure of a trans-translation inhibitor bound to the ribosome interestingly shows a change in the placement of the L27 tail. The authors describe these interactions briefly but their overall points aren't well articulated. While the N terminus of L27 in their ACC structure is similar to when a normal aminoacylated tRNA is bound at the A site, how then can the inhibitor be thought to impede accommodation of tmRNA-SmpB? Since the N term of L27 looks the same in both structures wouldn't it make more sense for it to prevent a downstream event of the ACC step? The statement in lines 500-2 seems to then discuss their results in general terms but then the last part comes back to the point of the trans-translation inhibitor: “Our structures give us a better understanding of how trans-translation

occurs, but they also showcase its differences and similarities with canonical translation, and confirm the fact that it does not interfere with that process.” This section needs clarifying.

We agree. To make our discussion more consistent we decided to move the digression on bL27 and its possible links with KKL-2098 inhibitor to the ACC chapter (lines 260-266):

“Interestingly, it was recently suggested that a rotated conformation of bL27 is preferred when non-stop ribosomes are bound by KKL-2098, a newly proposed trans-translation inhibitor³⁸. The N-terminal arm is usually quite flexible but it is particularly well-resolved in the current ACC conformation. This may imply that trans-translation specifically requires the N terminus of bL27 to be oriented toward the PTC. It would explain why KKL-2098 stabilisation of the rotated orientation of bL27 impedes trans-translation but not canonical translation.”

The last sentence was indeed a bit too generic, however we thought that it was important to conclude with the inhibitors, as we think it is the next step for trans-translation studies. With that in mind we modified the last paragraph as follows (line 604 to 620):

“To conclude, trans-translation is an appealing antibiotic target since it is essential for fitness and vital to many pathogens, yet absent in eukaryotes. Our structures give a better understanding of how trans-translation occurs, but they also showcase its differences and similarities with canonical translation, and confirm the fact that it does not interfere with that process. This is of primary importance for developing new antibiotics targeting trans-translation, as that depends on understanding how tmRNA and SmpB interact and how they bind to stalled ribosomes. We think that, for example, our detailed description of the interactions between the accommodated tmRNA-SmpB and the ribosome, as well as the newly described interface between the MLD and SmpB in the TRANS conformation, should be useful for the recently proposed anti-trans-translation strategies^{38,61-63} and even in rational drug design.”

Figures:

-The figures containing maps need to be remade using a different way to show density (ie use transparencies) otherwise they make the figures worse. These include Fig 2b,c,e,f,h,I (and some of these maps do not appear to be aligned with the models!), Fig 3, Fig 4 (do you need to see the map of 16S?), Fig 5 (do you need to see the map of tmRNA?), Fig 6, Fig 8, and Supp Fig 9 (hard to focus on what is important).

We added densities to most of the figures in response to previous comments of reviewer 2. As requested they were changed from mesh to transparency. We verified that maps and models were properly aligned although we admit that the view and representation can be sometimes misleading. We deleted the densities for the 16S rRNA, uS4 and uS5 in Figures 4 and for the H5 stem-loop in Figure5. We modified Supplementary Fig. 9 to focus on Trp122 and G19 and we removed the densities for more clarity.

In contrast, the maps shown in Fig 7 and Supp Fig 11 look good but it appears this is using a different program to generate the figures?

Thanks, however all the figures were made using UCSF Chimera. Note that Fig. 7 was also modified to answer some concerns about local resolution expressed by reviewer 2.

-The supplementary figure numbering is out of order and is incredibly confusing. First supplementary figure to be called out in the manuscript is S5 (line 114). This needs updating.

Sorry about that, but to us the Supplementary figures are cited in the right order: S1 line 53; S2 to S4 lines 86-87 (end of the introduction); S5 line 123; S6 line 126; S7 line 132; S8a line 134; S9a line 136; S9b line 139 etc.

Your feeling is maybe due to the fact that we often have to come back to figures and Supplementary Fig. already described, since they depict the PRE-ACC, ACC and TRANS states (eg Supp 8, 9, 10, 11)

-Figure 9 should be labeled.

Done

-The phosphate groups of most of RNA molecules including tmRNA and rRNA are above the ribbon denoting the backbone. This can be easily fixed by either having the ribbon go through the phosphate or turning off the phosphate group. Either way, it should be fixed because in some cases there is no connectivity in the RNA.

We use the default settings of UCSF Chimera and it cannot be easily modified and should not as it is a convention. We acknowledge the fact that this is not optimal. Since the figures were heavily reworked and some densities removed we chose to use either the default cartoon representation (in most cases) or the “all atom” representation when necessary.

MINOR ISSUES

Text:

-Line 37: tmRNA is not completely ssRNA (has a tRNA domain!) so this should be updated.

We deleted the word “single-stranded”

-Line 50, 211, 344: “thanks”. A better word should be used as noted previously

Done.

-Line 110: Issues with again state “very first high resolution”

We deleted “very” but kept the notion of “*first high-resolution structure that shows the ... pseudoknot ring*” as it is true. However we removed the “near-atomic” statement and limit the use of “high resolution” to the minimum.

-Lines 135-7: “PK2’s nucleotides C183 to A185 interact with residues Arg72, Pro73, and Ile77, which belong to the type II K-homology (KH2) RNA-binding domain of uS3 (Fig. 2b)25.” Is it appropriate to cite Noller’s paper here?

The citation was removed.

-Line 213: If an interaction becomes closer, that is fine to describe as “closer” but the term “tighter” should not be used as this implies some sort of binding thermodynamics.

The term “tightly” was replaced by “closely”.

-Line 234: The text jumps from Fig 5 to Fig 9 in the text

Sorry about that, but again, to us, the Figures and Supplementary Fig. are in the right order.

-“16S rRNA H44 helix”- helices of the small subunit are lowercase

Thanks. All the small subunit helices are now in lower case.

-Lines 247-9: “Arg153 and His159 are bound ionically to the phosphate groups between U119, U120, and A121,”... This is awkward and it is not clear this is scientifically correct. Why not just say “interact”.

Done

-Line 349: need to define what A/P refers to.

We added the following (Lines 410-411): *“(in these states, the first letter refers to the position of the tRNA on the 30S subunit and the second to its position on the 50S)”*

-Lines 453-5: This sentence should be rewritten for clarity

The sentence was modified as follows (lines 565-567): *“While passing through the ribosome, the SmpB C-terminal tail remains folded in an α -helix. However, it is always tightly fixed inside the mRNA channel thanks to its positively charged residues (Fig. 4)”*

-Line 463-4: What do the authors mean by “we only observed a moderate interaction between His22 and A1493”? Do they mean stacking interaction? If so, they should use this terminology.

Thanks for your remark we added the term “stacking”.

Figures:

-Supplementary Fig 6: panel A has the TLD residues labeled but the EF-Tu residues are not (although it says they are in the legend). Please label.

Done

-Supplementary Fig 5- overlay between previously determined structure and the structure presented here. The authors should include how these structures were aligned and provide an rmsd between the two to quantify the differences

The overlay was obtained by rigid-body, fitting Neubauer *et al.* atomic model (including the ribosome) into the PRE-ACC electron density map. The RMSD between the two complexes

calculated for the P and C α is 3.23 Å, with SmpB (C α -RMSD^{SmpB} = 4.1 Å) accounting for the biggest difference between the two structures. The caption of the Supplementary Fig. 5 was modified in consequence.

-Supplementary Fig 7: silver and gray are the same colors so please update the legend.

The figure was modified. The tRNA is now white and EF-Tu is light blue.

-Grammatical errors in the legend for Supplementary Fig 9

The grammatical errors were corrected.

REVIEWERS' COMMENTS

Reviewer #2 (Remarks to the Author):

The authors have carefully and thoroughly gone through all the remaining criticisms from the last round of review and have updated or provided new figures to illustrate the local resolution of the various regions discussed. I think this has significantly improved the transparency of the manuscript. Overall, I think the results will be of general interest to the translation field, although have suffered somewhat in novelty due to the previous Ramakrishnan publication. However, here the authors present a few new states and in retrospect some validation of the other states by different groups is also not a bad thing.

Reviewer #4 (Remarks to the Author):

Having seen the manuscript and the previous reviewers' comments and response, I believe that this manuscript both confirms the previous results of Rae et al 2019 at improved resolution and also reports an additional 4th structure. Considering the complexity of the process and the general interest in tmRNA, I support its publication in Nature Comm.

I think in their efforts to make the article sound "original", they have not been as forthcoming. In previous rounds, reviewers have pointed out that three of the four structures were reported previously. Although the ms. is much improved, it is still not clear from the manuscript. I suggest that the authors reword the sentence in the abstract to say something like "we describe previously published structures of the accommodated, translocated and doubly-translocated states at about somewhat higher resolution as well as a new structure of the pre-accommodated state."

In the results and discussion, it would also be appropriate to say which parts are confirmatory and which parts are new. This does not stand out clearly.

Finally, I agree with earlier reviews that it is a real reach from this structure to drug discovery and it is still a bit too enthusiastic about what is a far-fetched idea.

Separate point-by-point response to the reviewers' comments:

Reviewer #2 (Remarks to the Author):

The authors have carefully and thoroughly gone through all the remaining criticisms from the last round of review and have updated or provided new figures to illustrate the local resolution of the various regions discussed. I think this has significantly improved the transparency of the manuscript. Overall, I think the results will be of general interest to the translation field, although have suffered somewhat in novelty due to the previous Ramakrishnan publication. However, here the authors present a few new states and in retrospect some validation of the other states by different groups is also not a bad thing.

We thank you very much for your insightful comments. I have to say that the manuscript is not the same now and has been greatly improved thanks to the reviewer's deep remarks.

Reviewer #4 (Remarks to the Author):

Having seen the manuscript and the previous reviewers' comments and response, I believe that this manuscript both confirms the previous results of Rae et al 2019 at improved resolution and also reports an additional 4th structure. Considering the complexity of the process and the general interest in tmRNA, I support its publication in Nature Comm.

I think in their efforts to make the article sound "original", they have not been as forthcoming. In previous rounds, reviewers have pointed out that three of the four structures were reported previously. Although the ms. is much improved, it is still not clear from the manuscript. I suggest that the authors reword the sentence in the abstract to say something like "we describe previously published structures of the accommodated, translocated and doubly-translocated states at about somewhat higher resolution as well as a new structure of the pre-

accommodated state."

OK, that makes a lot of sense and the abstract was modified accordingly, with the help of the editor

In the results and discussion, it would also be appropriate to say which parts are confirmatory and which parts are new. This does not stand out clearly.

OK, we have added the appropriate statements (eg "confirm") where necessary and highlighted in the commentaries the sentences where this was already the case

Finally, I agree with earlier reviews that it is a real reach from this structure to drug discovery and it is still a bit too enthusiastic about what is a far-fetched idea.

OK, this was discarded from the abstract and toned down in the conclusion.